# FB-MOAC: A Reinforcement Learning Algorithm for Forward-Backward Markov Decision Processes

**Mohsen Amidzadeh**                                     *mohsen.amidzade@aalto.fi*
*Department of Computer Science*
*Aalto University, Finland*

**Mario Di Francesco**                                    *mario.di.francesco@aalto.fi*
*Department of Computer Science*
*Aalto University, Finland*

**Reviewed on OpenReview:** *https://openreview.net/forum?id=li5DyC6rfS*

## Abstract

Reinforcement learning (RL) algorithms are effective in solving problems that can be modeled as Markov decision processes (MDPs). These algorithms primarily target forward MDPs whose dynamics evolve over time from an initial state. However, several important problems in different scenarios including stochastic control and network systems exhibit both a forward and a backward dynamics. As a consequence, they cannot be expressed as a standard MDP, thereby calling for a novel theory for RL in this context. Accordingly, this work introduces the concept of Forward-Backward Markov Decision Processes (FB-MDPs) for multi-objective problems, develops a novel theoretical framework to characterize their optimal solutions, and proposes a general forward-backward step-wise template that allows to adapt RL algorithms for FB-MDP problems. A Forward Backward Multi Objective Actor Critic (FB-MOAC) algorithm is introduced accordingly to obtain optimal policies with guaranteed convergence and a competitive rate with respect to standard approaches in RL. FB-MOAC is evaluated on diverse use cases in the context of mathematical finance and mobile resource management. The obtained results show that FB-MOAC outperforms the state of the art across different metrics, highlighting its ability to learn and maximize rewards.

## 1 Introduction

Reinforcement Learning (RL) is a very important field of artificial intelligence, as it enables agents to learn from experience and adapt to complex, dynamic environments (Mnih et al., 2013; Lillicrap et al., 2016; Schulman et al., 2017b). Moreover, recent breakthroughs in deep learning have led to solutions that surpass human performance in a wide variety of challenges. As a result, deep reinforcement learning has lately emerged as a combination of these two fields, with successful applications in different use cases (Mnih et al., 2015; Jaderberg et al., 2018; Rigoli et al., 2021).

Existing RL algorithms mainly address sequential decision-making problems modeled as a forward Markov decision process (MDP) or controlled forward dynamics (Zare et al., 2023). However, there are several sequential tasks whose environment cannot be exclusively captured by this type of dynamics, as they also encompass states evolving backwards in time (Lai et al., 2020; Wang et al., 2021). Such backward dynamics describe a trajectory in a reverse chronological order, wherein the future affects the past. Even further, there are environments exhibiting *both* controlled forward and backward dynamics at the same time (Ji et al., 2022a; Zhang, 2022), namely, as the forward-backward MDP (FB-MDP) illustrated in Figure 1a.

FB-MDPs have wide applications (Section 2.1). For instance, they can be employed to discretize forward-backward stochastic differential equations (SDEs) (see Appendix F for a detailed account), thereby allowing to solve stochastic optimal control problems (Zhang, 2017; Ji et al., 2020). Moreover, they enable accurate

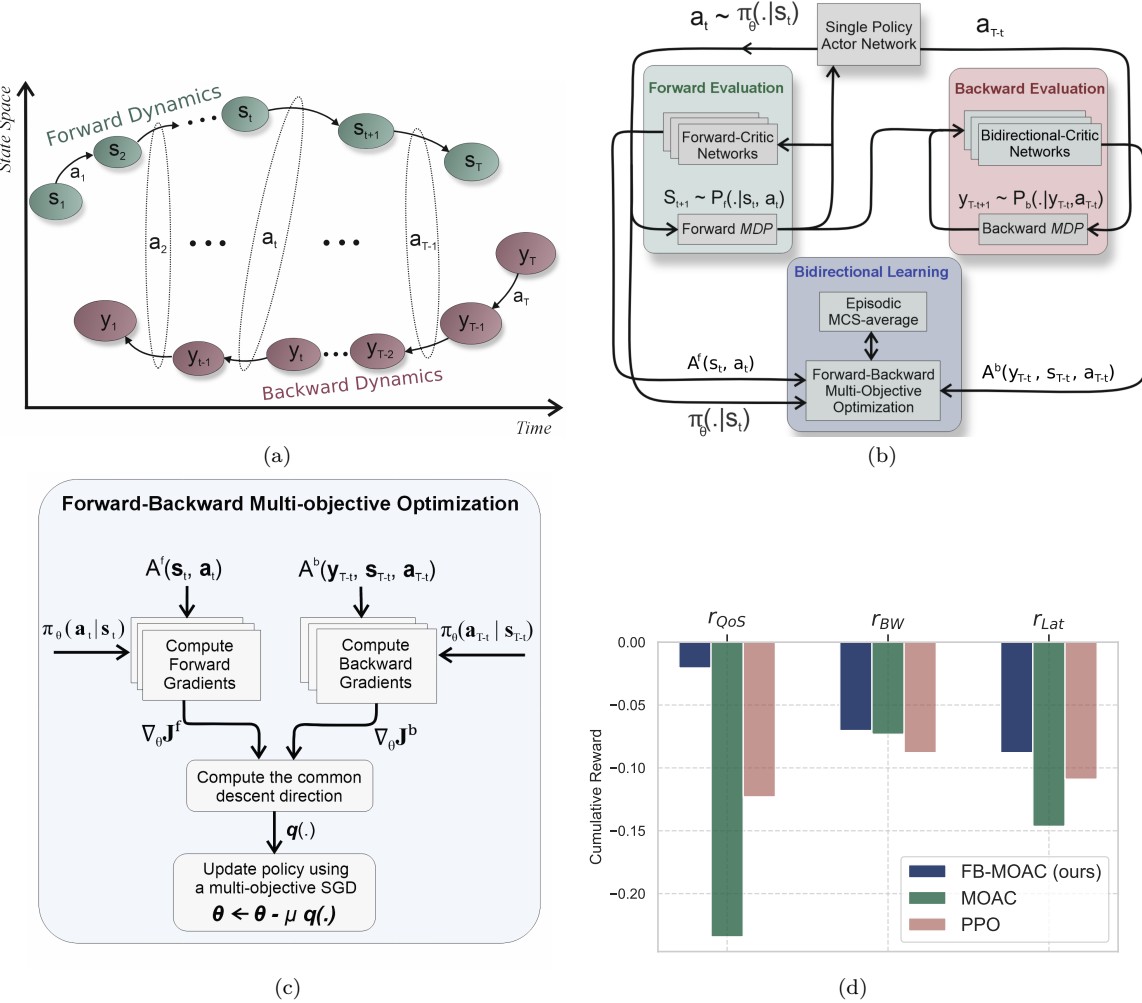

Figure 1: Overview of our approach. (a) A *forward-backward MDP* in which forward states $\{\mathbf{s}_t\}_{t\in\{1,T\}}$ and backward states $\{\mathbf{y}_t\}_{t\in\{1,T\}}$ apply the same actions $\{\mathbf{a}_t\}_{t\in\{1,T\}}$, but with a different ordering in time. (b) The FB-MOAC algorithm comprises three steps: *forward evaluation, backward evaluation* and *bidirectional learning*. During the first two steps, the forward and backward dynamics are evaluated and the resulting experiences are buffered. The policy distribution is optimized in the bidirectional learning step based on the experiences of both forward and backward dynamics. For this purpose, a *forward-backward multi-objective optimization* is employed by following an appropriate chronological order. The *episodic MCS-average* add-on boosts the convergence to Pareto-optimal solutions. (c) The multi-objective optimization module of the FB-MOAC algorithm computes: the vector-valued gradients of forward and backward objectives; the descent direction $\mathbf{q}(\cdot)$ to ensure that all rewards increase simultaneously; and the parameters of the actor network based on $\mathbf{q}(\cdot)$. (d) The cumulative reward of our approach (FB-MOAC) compared to the widely-used proximal policy optimization (PPO) and to MOAC (our multi-objective extension of the advantage actor critic) in the *edge caching* use case for different metrics (see Section 5.2 for a detailed account). FB-MOAC performs better than the other algorithms in terms of the overall reward.

modeling of delay / latency in the context of network communications and of the execution time for task offloading in cloud or edge computing systems (Liu et al., 2019; Wei et al., 2019; Chen et al., 2019b). However, forward-backward dynamics have been marginally addressed in the context of RL and MDPs (Section 2.2). In fact, existing research only formulated a deep learning characterization in terms of forward-backward SDEs (Ji et al., 2020) or considered artificial backward trajectories in forward MDPs to increase the sample efficiency of RL algorithms (Goyal et al., 2019; Wang et al., 2021). In contrast, and to fill this gap, we introduce the concept of FB-MDPs for multi-objective problems entailing both forward and backward rewards that conflict with each other throughout the action space. We then extract the properties of Pareto-optimal

solutions and extend the optimality Bellman equation for this class of MDPs. We provide a general solution mechanism and then develop an RL algorithm specifically suited to FB-MDPs accordingly.

In detail, this work establishes the following contributions.

- We introduce the notion of *multi-objective FB-MDPs* to express sequential multi-task decision-making problems with both forward and backward controlled dynamics, whose rewards are coupled within the action space. We notably show that these MDPs **cannot** be expressed as a standard MDP, and we consequently develop a novel theoretical framework to characterize their optimal solutions (Section 3).

- We propose a step-wise mechanism as a general and versatile template to adapt RL algorithms for FB-MDPs. Accordingly, we devise a multi-objective RL algorithm, called *Forward-Backward Multi-Objective Actor-Critic* (FB-MOAC), the first of its kind, to obtain Pareto-optimal solutions. We further provide a rigorous analysis of FB-MOAC, showing that it reaches convergence with a rate of $\mathcal{O}(1/\sqrt{I})$, where $I$ is the number of policy updates (Section 4).

- We conduct a comprehensive evaluation by considering diverse use cases expressed as FB-MDPs in the context of mathematical finance and mobile resource management. The results establish the necessity of modeling these problems as FB-MDPs, since approximating them with standard MDPs leads to a sub-optimal solution. The evaluation also demonstrates the effectiveness of FB-MOAC compared to RL algorithms in the state of the art (Section 5).

**Notation**: we use lower-case $a$ for scalars, bold-face lower-case $\mathbf{a}$ for vectors, and bold-face uppercase $\mathbf{A}$ for matrices. Moreover, $\mathbf{A}^\top$ is the transpose of $\mathbf{A}$, $\|\mathbf{A}\|$ is the induced matrix norm of $\mathbf{A}$, $\mathbf{I}$ is the identity matrix, $\mathbf{1}$ a vector with all elements equal to one, $\mathbf{0}$ a vector with all elements equal to zero, and $\mathbf{e}_m$ a vector with all elements equal to zero except the $m$-th element which is equal to one. Finally, $\boldsymbol{a} = [a_1, \ldots, a_n]^\top$ are the components of a $n$-dimensional column vector $\boldsymbol{a}$, $|S|$ is the cardinality of the set $S$, and $[\cdot]$ indicates the components of row vectors.

## 2 Background

This section first introduces a few motivating examples of FB-MDPs and then reviews the most relevant works in the existing literature.

### 2.1 Motivating Examples

**Problems Described by Stochastic Differential Equations**

SDEs exhibiting anti-causal dynamics have several applications in the context of differential games (Hamadene & Lepeltier, 1995), diffusion models (Yang et al., 2023), and mathematical finance (Ji et al., 2022a). In particular, problems involving FB-SDEs represent a significant portion of the ongoing research in the field of stochastic control theory (Yong, 2023). Among them, one example is given by an investment-consumption scenario in mathematical finance. Consider a financial market with a single risky asset whose price follows a stochastic process. A trader can invest in this risky asset or engage in risk-free borrowing / lending. The trader's total wealth $Y(t)$ evolves based on their investment in the asset and the risk-free rate. Now, consider a payoff at a future time $T$, which depends on the asset's price. The goal is to determine the minimal initial $Y_0$, required to replicate this payoff. The investment strategy guaranteeing that the final wealth matches the option's payoff is characterized by a backward dynamics (Ji et al., 2022a). In such a context, Section 5.1 presents a use case related to mathematical finance, based on a general method to transform an FB-SDE into an FB-MDP (see Appendix F for more details).

**Network Content Delivery**

Let us consider a scenario in which a content provider (e.g., in a video streaming context) serves users by transmitting $N$ content items with different popularity $\{p_n\}_{n=1}^N$ over a lossy network. Transmissions take

place in time slots indexed by $t$, and the delivery of content $n$ fails with an error probability $e_n(t)$. Upon failure, user requests are re-transmitted until successful delivery. The request probability of content $n$ clearly depends on the success rate of previous requests and the error probability $e_n(t)$. Therefore, it establishes a controlled *forward dynamics* as the content provider affects $e_n(t)$. Now, the average latency $l_n(t)$ experienced by a typical user to successfully receive file $n$ is obtained by: $l_n(t) = d(t)(1 - e_n(t)) + (\tau(t) + l_n(t+1))e_n(t)$, where: $d(t)$ is the transmission delay between the content provider and the user; and $\tau(t)$ is the duration of time slot $t$. This equation is obtained by the law of total expectation and exhibits a controlled *backward dynamics* with $l_n(t)$ as a backward state. As a consequence, minimizing the overall latency $\sum_{n=1}^{N} p_n(t)l_n(t)$ in this context makes the problem an FB-MDP. Existing RL algorithms cannot be applied here since a backward MDP cannot be converted to a standard forward MDP (see Theorem 3.6). Note that this FB-MDP cannot be completely replaced by a standard MDP problem either. In fact, one rewarding policy could be to only track the number of successful transmissions. A higher success rate over $[0, T]$ correlates with a lower latency, however, these two metrics are not interchangeable; one can maintain the same success percentage yet obtain a different overall latency $\sum_{n=1}^{N} p_n(t)l_n(t)$. For instance, both the number of successful transmissions and the average latency $l_n(t)$ become determined by fixing the error probability $e_n(t)$ over $[0, T]$. If we then permute $e_n(t)$ in time, the total count of successful transmissions remains unchanged, but the overall latency differs due to presence of $p_n(t)$. Therefore, the success (or failure) of content transmissions does not completely map to the overall latency. Content delivery is only one instance of network problems that can be modeled as a FB-MDP; Section 5.2 presents a use case on edge caching in wireless networks, whereas Appendix D addresses computation offloading.

## 2.2 Related Work

**Forward-Backward MDPs.** Our work shares similarity with prior research on RL algorithms (Edwards et al., 2018; Goyal et al., 2019; Wang et al., 2021; Lai et al., 2020; Archibald et al., 2023). These studies hypothesize that (creating) a virtual backward trajectory, in relation to a forward dynamics, enhances the sample efficiency of RL algorithms. Specifically, they employ the generated backward trajectories to augment the training dataset for learning forward MDP (SDE) problems. Edwards et al. (2018) train a learnable backwards dynamics that generates fictitious reversal steps from known goal states. These backward paths are then used to augment the replay buffer and contribute to the learning procedure of the considered RL algorithm. Similarly, Goyal et al. (2019) learn an artificial backward model (called backtracking model) trained on agent experiences to predict preceding state-action pairs leading to high-value states. The backtracking model then enriches the training dataset by alternative trajectories, leading to better performance. Lai et al. (2020) construct a backward dynamics model and leverages it alongside the forward model to generate short branched rollouts for policy optimization. This approach theoretically derives a tighter bound of return discrepancy and demonstrates improved performance compared to a forward-only method. Wang et al. (2021) introduce a learnable backward dynamics trained on offline datasets to generate rollouts targeting goal states. They augment the training dataset to cope with the distribution shift between the learning policy and the given offline dataset. Our reference model is characterized based on real backward and forward dynamics, in contrast with the works described above, wherein backward dynamics are artificially constructed based on a forward MDP (dynamics) and it is not independently controlled. Our model also considers rewards jointly competing in both directions of time. Consequently, our investigation is centered around a class of FB-MDPs for multi-task problems and allows for the development of RL algorithms that inherently consider bidirectional dynamics, rather than relying on fictitious backward trajectories. Therefore, it provides a more integrated and principled method for incorporating backward reasoning into RL.

**Multi-objective RL Algorithms.** The majority of Multi-Objective Reinforcement Learning (MO-RL) algorithms has primarily been designed for discrete environments. Mossalam et al. (2016) introduce a MO-RL algorithm that combines deep Q-learning and optimistic linear support learning. Their approach take into account a scalarized vector and potential optima to formulate a convex combination of all objectives. However, they require searching over all potential scalarizing vectors since an a priori knowledge on the importance of distinct objectives is not available. Yang et al. (2019) employ multi-objective Q-learning together with a single-agent framework to acquire a preference-related adjustment that can be generalized across different objectives. Such an approach is computationally efficient, however, it often suffers from sample inefficiency and results

in a sub-optimal policy. MO-RL algorithms have been specifically developed for continuous environments as well. Zhan & Cao (2019) establish reward-specific state-value functions based on a correlation matrix to obtain the relative importance of objectives with respect to each other. However, their approach requires to adjust the weight of such a matrix to determine an appropriate inter-objective relationship. Abdolmaleki et al. (2020) devise a MO-RL approach according to the *maximum a posteriori policy optimization* algorithm. They learn objective-specific policy distributions to identify Pareto-optimal solutions in a scale-independent manner. However, objective-specific coefficients must be adjusted to control the impact on the policy update. In contrast, we propose a MO-RL algorithm for continuous-valued FB-MDPs (namely, FB-MOAC), without considering any initial preferences for the different objectives. Different from previous works (Abdolmaleki et al., 2020; Zhan & Cao, 2019; Chen et al., 2019a), we devise a *single-policy approach* to simplify the algorithm and avoid the need for an initial assumption on the reward preference. Moreover, a comprehensive analysis has been conducted to ensure the convergence of FB-MOAC to Pareto-front solutions at a certain rate. A remarkable result of such analysis is the ability of FB-MOAC to monotonically increase all expected objectives for any reward preference, thereby making the algorithm scale-insensitive.

**Convergence Analysis of RL algorithms.** A few recent works (Qiu et al., 2021; Xu et al., 2020; Fu et al., 2021; Yang et al., 2018; Khodadadian et al., 2022) have characterized RL algorithms with a stochastic policy, such as Actor-Critic (AC) and Policy Gradient (Sutton & Barto, 2018). Qiu et al. (2021) conduct a rigorous convergence analysis of the AC algorithm. Notably, their analysis is limited to a linear representation of the state-value function. Xu et al. (2020) provide a comprehensive characterization of the convergence rate and sample complexity of the Natural Actor-Critic (NAC) algorithm (Peters & Schaal, 2008). Their analysis requires that the considered MDP is ergodic. Fu et al. (2021) analyze the convergence of the AC algorithm under the assumption that the considered family of neural networks are closed under the Bellman operator. Lastly, Khodadadian et al. (2022) perform an accurate convergence analysis of the Natural Policy Gradient algorithm (Kakade, 2002). However, their investigation assumes that the initialization value of the state-value function is sufficiently close to the optimal value function. All the aforementioned works address the convergence of stochastic policies in single-objective RL algorithms for forward MDP problems. In contrast, this work targets multi-task problems involving a FB-MDP. We carry out a rigorous convergence analysis as a solid foundation to characterize multi-objective and forward-backward RL algorithms for FB-MDPs.

**Applications of RL to Network Systems.** RL algorithms have also been applied to network systems, particularly, to design dynamic caching and offloading policies (Zhang et al., 2021; Chen et al., 2021; Amidzadeh et al., 2021; Jiang et al., 2022; Chen et al., 2022; Zhou et al., 2023). Chen et al. (2021) devise a multi-agent reinforcement learning for ultra-dense networks, whereas Zhang et al. (2021) employ a deep RL algorithm to jointly optimize resource allocation and caching for Internet-of-Things scenarios. Amidzadeh et al. (2021) leverage a deep RL-based approach to develop an optimal cache policy for multicast-enabled cellular networks. Moreover, Jiang et al. (2022) develop an actor-critic RL algorithm for proactive caching in mobile edge networks. Finally, Chen et al. (2022) and Zhou et al. (2023) employ deep RL for joint caching and offloading problems in edge computing networks. All the works mentioned above only consider forward dynamics, whereas this work entails a more complex characterization that allows to obtain a better solution (see Section 5.2 for a detailed account).

## 3 Multi-Objective FB-MDPs

This section briefly describes multi-objective optimization and its associated Pareto-optimality as a basis to formally define FB-MDPs. The section concludes by characterizing the optimal solution of a multi-objective FB-MDP problem.

### 3.1 Pareto Optimality

Consider the following multi-objective optimization problem:

$$Q_1: \qquad \min_{\boldsymbol{x} \in \mathcal{X}} \ \big[f_1(\boldsymbol{x}), \ldots, f_r(\boldsymbol{x})\big],$$

where $f_j : \mathbb{R}^N \to \mathbb{R}$, $\mathcal{X}$ is the feasible set and $r$ the number of objectives.

**Definition 3.1.** We say that $\mathbf{y} \in \mathcal{X}$ Pareto-dominates $\mathbf{x} \in \mathcal{X}$, if $f_i(\boldsymbol{y}) \leq f_i(\boldsymbol{x})$ for all $i \in \{1, \ldots, r\}$ and there exists $j \in \{1, \ldots, r\}$ such that $f_j(\boldsymbol{y}) < f_j(\boldsymbol{x})$.

**Definition 3.2.** $\boldsymbol{x}^* \in \mathcal{X}$ is called a Pareto-optimal solution of $Q_1$, if there is no other solution $\boldsymbol{y} \in \mathcal{X}$ that dominates $\boldsymbol{x}^*$. Accordingly, $\left[f_1(\boldsymbol{x}^*), \ldots, f_r(\boldsymbol{x}^*)\right]$ is called a Pareto-optimal vector, and $\min_{\boldsymbol{x} \in \mathcal{X}} \left[f_1(\boldsymbol{x}), \ldots, f_r(\boldsymbol{x})\right]$ indicates to the set of Pareto-optimal solutions.

The following lemma (Schäffler et al., 2002; Ma et al., 2020) is instrumental to jointly minimize all objectives of $Q_1$.

**Lemma 3.3.** *Consider a vector-valued multivariate function $\boldsymbol{f} = [f_1, \ldots, f_r]$, $f_j : \mathbb{R}^n \to \mathbb{R}$ for $j \in \{1, \ldots, r\}$. Let $\boldsymbol{q}(\cdot) = \sum_{j=1}^r \alpha_j^* \nabla f_j(\cdot)$, then $-\boldsymbol{q}(\cdot)$ is a descent direction for all functions $\{f_j(\cdot)\}_1^r$, where $\{\alpha_j^*\}_1^r$ are the solutions of the following optimization problem:*

$$Q_2 : \qquad \min_{\{\alpha_j\}_{j=1}^r} \left\| \sum_{j=1}^r \alpha_j \nabla f_j(\cdot) \right\|^2, \qquad \text{s.t.} \quad \sum_{j=1}^r \alpha_j = 1, \quad \alpha_j \geq 0, \quad j \in \{1, \ldots, r\}.$$

**Remark 3.4.** The lemma above can be leveraged to develop a multi-objective gradient descent algorithm. To jointly decrease different objectives, it suffices to optimize $\boldsymbol{\alpha} = [\{\alpha_j\}_{j=1}^r]$ by using the quadratic program $Q_2$ and obtain $\mathbf{q}(\cdot)$, which is *nonlinear* as $\boldsymbol{\alpha}$ itself depends on $\{\nabla f_j(\cdot)\}_j$.

Accordingly, the optimal solution of problem $Q_2$ can be obtained as follows.

**Corollary 3.5.** *If $\nabla \boldsymbol{f}(\cdot)^\top \nabla \boldsymbol{f}(\cdot)$ is invertible and all $\alpha_j \geq 0$, the solution of $Q_2$ is given by:*

$$\boldsymbol{\alpha}^* = \left( \mathbf{1}_r^\top \left( \nabla \boldsymbol{f}(\cdot)^\top \nabla \boldsymbol{f}(\cdot) \right)^{-1} \mathbf{1}_r \right)^{-1} \left( \nabla \boldsymbol{f}(\cdot)^\top \nabla \boldsymbol{f}(\cdot) \right)^{-1} \mathbf{1}_r, \tag{1}$$

*where $\nabla \boldsymbol{f}(\cdot)$ is an $n \times r$ matrix with $\nabla \boldsymbol{f}(\cdot) = \left[\nabla f_1, \ldots, \nabla f_r\right](\cdot)$. For the case $\alpha_j < 0$ for $j \in \mathcal{S}_0 \subset \{1, \ldots, r\}$, we set $\nabla \boldsymbol{f}(\cdot) = [\nabla f_k(\cdot)]_{k \in \{1, \ldots, r\} \setminus \mathcal{S}_0}$.*

### 3.2 Forward-Backward Markov Decision Processes

We introduce a class of *multi-objective FB-MDPs*, expressed by a tuple $\left( \mathcal{S}, \mathcal{Y}, \mathcal{A}, P_f(\cdot), P_b(\cdot), \boldsymbol{r}^f(\cdot), \boldsymbol{r}^b(\cdot) \right)$, where: $\mathcal{S}$ and $\mathcal{Y}$ are the forward and backward state-spaces, respectively; $\mathcal{A}$ is the action space; $P_f : \mathcal{S} \times \mathcal{A} \times \mathcal{S} \to [0, 1]$ is the forward transition probability, which describes the forward dynamics; $P_b : \mathcal{Y} \times \mathcal{A} \times \mathcal{Y} \to [0, 1]$ is the backward transition probability, which expresses the backward dynamics; finally, $\boldsymbol{r}^f : \mathcal{S} \times \mathcal{A} \to \mathbb{R}^{|S_f|}$ and $\boldsymbol{r}^b : \mathcal{Y} \times \mathcal{A} \to \mathbb{R}^{|S_b|}$ are the forward and backward reward functions (respectively), where $S_f$ and $S_b$ are the sets of indices of the forward and backward rewards (respectively). The forward transition probability determines the next forward state of the system $\mathbf{s}_{t+1} \sim P_f(\cdot | \mathbf{s}_t, \mathbf{a}_t)$ starting from $\mathbf{s}_t \in \mathcal{S}$ and performing the action $\mathbf{a}_t \in \mathcal{A}$. Moreover, in an anti-causal way, the previous backward state of the system follows $\mathbf{y}_{t-1} \sim P_b(\cdot | \mathbf{y}_t, \mathbf{a}_t)$ from $\mathbf{y}_t \in \mathcal{Y}$ by performing action $\mathbf{a}_t \in \mathcal{A}$. The initial forward state $\mathbf{s}_1$ and the final backward state $\mathbf{y}_T$ are assumed to be known. Figure 1a on page 2 illustrates a FB-MDP.

**Assumption**. This work constrains the definition of FB-MDPs to the case where the forward (backward) dynamics does not depend on the backward (forward) state.

> **Remark 3.6 (FB-MDPs cannot be expressed as standard MDPs).** The backward dynamics **cannot** be represented based on a standard forward system *in presence of a forward dynamics*. We can consider the transformations $\mathbf{z}_{T-t} := \mathbf{y}_t$ and $t' := T - t$ to convert the backward MDP with transition probability $\mathbf{y}_{t-1} \sim P_b(\cdot | \mathbf{y}_t, \mathbf{a}_t)$ into a forward one. Consequently, we get a forward MDP over $\mathbf{z}_{t'}$ with transition probability $\mathbf{z}_{t'+1} \sim P_b(\cdot | \mathbf{z}_{t'}, \mathbf{a}_{T-t'})$. However, this is a **non-standard** MDP as the state $z_t$ becomes a function of actions that are scheduled for future time steps $\mathbf{a}_{T-t}$. Specifically, the state relies on future actions that are not available when progressing forward in time. This violation of the conventional causal structure prevents the use of standard RL algorithms.

The aim of a FB-MDP problem is thus to optimize the following discounted multi-objective cumulative reward from the Pareto-optimality perspective:

$$\max_{\{\mathbf{a}_t \in \mathcal{A}\}_{t \in \{1,T\}}} \mathbb{E}\left\{\sum_{t=1}^{T} \gamma^{t-1} \left[\boldsymbol{r}^f(\mathbf{s}_t, \mathbf{a}_t), \; \boldsymbol{r}^b(\mathbf{y}_{T-t+1}, \mathbf{a}_{T-t+1})\right]\right\}, \tag{2}$$

In Equation (2), $T \in \mathbb{N}$ is the finite horizon of the optimization, $\gamma \in [0,1]$ the discount factor, and the expectation refers to the different realizations of the forward-backward trajectory.

Remark 3.6 highlights that solving a FB-MDP problem of the type in Equation (2) requires developing novel theoretical foundations. To do so, we build on the following observation and the resulting optimal solutions.

> **Remark 3.7.** Both the forward and backward dynamics of a FB-MDP problem can be accurately learned through a $\boldsymbol{\theta}$-parametric stochastic policy $\mathbf{a}_t \sim \pi_{\boldsymbol{\theta}}(\cdot|\mathbf{s}_t)$, whereas employing the policy $\mathbf{a}_t \sim \pi_{\boldsymbol{\theta}}(\cdot|\mathbf{s}_t, \mathbf{y}_t)$ is unfeasible due to the anti-causal nature of the backward dynamics. Therefore, we need to optimize the policy $\pi_{\boldsymbol{\theta}}(\cdot|\mathbf{s}_t)$ based on the trajectories of *both* forward and backward dynamics.

The following section delves into this process.

### 3.3 Characterizing an Optimal Solution

We now analyze Remark 3.7 and provide a theoretical framework to characterize the optimal solution of a multi-objective FB-MDP problem. Accordingly, the probability of a forward-backward trajectory $\boldsymbol{\tau}$ is determined by:

$$P_{\boldsymbol{\theta}}(\boldsymbol{\tau}) := \mathbb{P}(\mathbf{s}_1, \mathbf{a}_1, \dots, \mathbf{s}_T, \mathbf{a}_T, \mathbf{y}_T, \dots, \mathbf{y}_1)$$
$$= \mathbb{P}(\mathbf{s}_1) \prod_{t=1}^{T-1} P_f(\mathbf{s}_{t+1}|\mathbf{s}_t, \mathbf{a}_t) \prod_{t=1}^{T-1} \pi_{\boldsymbol{\theta}}(\mathbf{a}_t|\mathbf{s}_t) \prod_{t=1}^{T-1} P_b(\mathbf{y}_{T-t}|\mathbf{y}_{T-t+1}, \mathbf{a}_{T-t+1}) \mathbb{P}(\mathbf{y}_T). \tag{3}$$

The problem in Equation (2) is then reformulated as the following *policy distribution optimization*:

$$\mathrm{O}_2: \qquad \max_{\boldsymbol{\theta}} \; \mathbb{E}_{\tau \sim P_{\boldsymbol{\theta}}(\boldsymbol{\tau})}\left\{\sum_{t=1}^{T} \gamma^{t-1} \left[\boldsymbol{r}^f(\mathbf{s}_t, \mathbf{a}_t), \; \boldsymbol{r}^b(\mathbf{y}_{T-t+1}, \mathbf{a}_{T-t+1})\right] \; \Big| \; \boldsymbol{\theta}\right\}$$
$$\text{s.t. } \left\{\mathbf{s}_{t+1} \sim P_f(\cdot|\mathbf{s}_t, \mathbf{a}_t), \qquad \mathbf{y}_{t-1} \sim P_b(\cdot|\mathbf{y}_t, \mathbf{a}_t), \qquad \mathbf{a}_t \sim \pi_{\boldsymbol{\theta}}(\cdot|\mathbf{s}_t)\right\}. \tag{4}$$

The multivariate objective of $\mathrm{O}_2$ can thus be expressed as:

$$\mathbf{J}(\boldsymbol{\theta}) := \Big[\underbrace{\mathbb{E}_{\tau \sim P_{\boldsymbol{\theta}}(\boldsymbol{\tau})}\sum_{k=1}^{T} \gamma^{k-1} \boldsymbol{r}^f(\mathbf{s}_k, \mathbf{a}_k)}_{\mathbf{J}^f(\boldsymbol{\theta})}, \quad \underbrace{\mathbb{E}_{\tau \sim P_{\boldsymbol{\theta}}(\boldsymbol{\tau})}\sum_{k=1}^{T} \gamma^{k-1} \boldsymbol{r}^b(\mathbf{y}_{T-k+1}, \mathbf{a}_{T-k+1})}_{\mathbf{J}^b(\boldsymbol{\theta})}\Big].$$

To Pareto optimize $\mathbf{J}(\boldsymbol{\theta})$, we need to first compute its component-wise gradient with respect to $\boldsymbol{\theta}$, i.e., $\nabla_{\boldsymbol{\theta}}\mathbf{J}(\boldsymbol{\theta}) = \frac{\partial \mathbf{J}(\boldsymbol{\theta})}{\partial P_{\boldsymbol{\theta}}(\boldsymbol{\tau})} \frac{\partial P_{\boldsymbol{\theta}}(\boldsymbol{\tau})}{\partial \boldsymbol{\theta}}$. For the forward cumulative rewards $\mathbf{J}^f(\boldsymbol{\theta})$, we have (Grondman et al., 2012):

$$\nabla_{\boldsymbol{\theta}}\mathbf{J}^f(\boldsymbol{\theta}) = \mathbb{E}\left\{\sum_{k=1}^{T} \nabla_{\boldsymbol{\theta}} \log \pi_{\boldsymbol{\theta}}(\mathbf{a}_k|\mathbf{s}_k) \mathbf{A}^f(\mathbf{s}_k, \mathbf{a}_k) \; \Big| \; \boldsymbol{\theta}\right\}, \tag{5}$$

where: $\mathbf{A}^f : \mathcal{S} \times \mathcal{A} \to \mathbb{R}^{|S_f|}$, $\mathbf{A}^f(\mathbf{s}_k, \mathbf{a}_k) := \boldsymbol{r}^f(\mathbf{s}_k, \mathbf{a}_k) + \gamma V^f(\mathbf{s}_{k+1}) - V^f(\mathbf{s}_k)$ is the forward advantage multivariate function; and $V^f : \mathcal{S} \to \mathbb{R}^{|S_f|}$ with

$$V^f(\mathbf{s}_k) := \mathbb{E}_{P_{\boldsymbol{\theta}}(\boldsymbol{\tau})}\left\{\sum_{k'=k}^{T} \gamma^{k'-k} \boldsymbol{r}^f(\mathbf{s}_{k'}, \mathbf{a}_{k'})|\mathbf{s}_k\right\} \tag{6}$$

is the forward state-value multivariate function. We finally obtain the following lemma to characterize the optimal backward trajectories and the Pareto-optimal solutions of FB-MDP problem $O_2$.

**Lemma 3.8.** *For the backward cumulative reward $\boldsymbol{J}^b(\boldsymbol{\theta})$, it is:*

$$\nabla_{\boldsymbol{\theta}} \boldsymbol{J}^b(\boldsymbol{\theta}) = \mathbb{E}_{P_{\boldsymbol{\theta}}(\boldsymbol{\tau})} \left\{ \sum_{k=0}^{T-1} \nabla_{\boldsymbol{\theta}} \log \pi_{\boldsymbol{\theta}}(\boldsymbol{a}_{T-k}|\boldsymbol{s}_{T-k}) \boldsymbol{A}^b(\boldsymbol{y}_{T-k}, \boldsymbol{s}_{T-k}, \boldsymbol{a}_{T-k}) \,\Big|\, \boldsymbol{\theta} \right\}, \tag{7}$$

*where $A^b : \mathcal{Y} \times \mathcal{S} \times \mathcal{A} \to \mathbb{R}^{|S_b|}$ is the bidirectional advantage multivariate function:*

$$A^b(\boldsymbol{y}_{T-k}, \boldsymbol{s}_{T-k}, \boldsymbol{a}_{T-k}) := \boldsymbol{r}^b(\boldsymbol{y}_{T-k}, \boldsymbol{a}_{T-k}) + \gamma V^b(\boldsymbol{y}_{T-k-1}, \boldsymbol{s}_{T-k-1}) - V^b(\boldsymbol{y}_{T-k}, \boldsymbol{s}_{T-k}),$$

*and $V^b : \mathcal{Y} \times \mathcal{S} \to \mathbb{R}^{|S_b|}$ is the bidirectional state-value multivariate function:*

$$V^b(\boldsymbol{y}_{T-k}, \boldsymbol{s}_{T-k}) := \mathbb{E}\left\{ \sum_{k'=T-k}^{1} \gamma^{T-k'-1} \boldsymbol{r}^b(\boldsymbol{y}_{k'}, \boldsymbol{a}_{k'}) \Big| \boldsymbol{y}_{T-k}, \boldsymbol{s}_{T-k} \right\} = \mathbb{E}\left\{ \sum_{k'=k}^{T-1} \gamma^{k'-k} \boldsymbol{r}^b(\boldsymbol{y}_{T-k'}, \boldsymbol{a}_{T-k'}) \Big| \boldsymbol{y}_{T-k}, \boldsymbol{s}_{T-k} \right\}, \tag{8}$$

*which adheres to the backward Bellman's equation:*

$$V^b(\boldsymbol{y}_{T-k}, \boldsymbol{s}_{T-k}) \quad = \quad \mathbb{E}_{\substack{\boldsymbol{a}_{T-k} \sim \pi_{\boldsymbol{\theta}}(\cdot|\boldsymbol{s}_{T-k}) \\ \boldsymbol{y}_{T-k-1} \sim P_b(\cdot|\boldsymbol{y}_{T-k}, \boldsymbol{a}_{T-k}) \\ \boldsymbol{s}_{T-k-1} \sim P(\cdot|\boldsymbol{s}_{T-k})}} \left\{ \boldsymbol{r}^b(\boldsymbol{y}_{T-k}, \boldsymbol{a}_{T-k}) + \gamma V^b(\boldsymbol{y}_{T-k-1}, \boldsymbol{s}_{T-k-1}) \Big| \boldsymbol{\theta} \right\}. \tag{9}$$

*For the stationary forward and backward transition probabilities, a Bellman Pareto-optimality equation is given by:*

$$\left[ V^{f^*}(\boldsymbol{s}), V^{b^*}(\boldsymbol{y}, \boldsymbol{s}) \right] \in \max_{\boldsymbol{a}} \left[ \mathbb{E}_{\boldsymbol{s}^+ \sim P_f(\cdot|\boldsymbol{s}, \boldsymbol{a})} \left\{ \boldsymbol{r}^f(\boldsymbol{s}, \boldsymbol{a}) + \gamma V^{f^*}(\boldsymbol{s}^+) \right\}, \mathbb{E}_{\substack{\boldsymbol{y}^- \sim P_b(\cdot|\boldsymbol{y}, \boldsymbol{a}) \\ \boldsymbol{s}^- \sim P(\cdot|\boldsymbol{s})}} \left\{ \boldsymbol{r}^b(\boldsymbol{y}, \boldsymbol{a}) + \gamma V^{b^*}(\boldsymbol{y}^-, \boldsymbol{s}^-) \right\} \right], \tag{10}$$

*for $(\boldsymbol{s}, \boldsymbol{y}, \boldsymbol{a}) \in \mathcal{S} \times \mathcal{Y} \times \mathcal{A}$, where $\left[ V^{f^*}(\boldsymbol{s}), V^{b^*}(\boldsymbol{y}, \boldsymbol{s}) \right]$ is a Pareto-optimal vector, $\boldsymbol{s}^+ \in \mathcal{S}$ is the forward state following $\boldsymbol{s}$, and $\boldsymbol{y}^- \in \mathcal{Y}$ is the backward state preceding $\boldsymbol{y}$.*

**Remark 3.9.** The formulation of Lemma 3.8 differs from its counterpart for forward MDPs. Specifically, the bidirectional state-value $V^b(\mathbf{y}_{T-k}, \mathbf{s}_{T-k})$ is defined in Equation (9) *so as to* have a backward Bellman's equation. Note that Equation (9) exhibits a forward dynamics with a dependency on the policy distribution that itself relies on the forward state rather than the backward state. Moreover, the *Bellman's Pareto-optimality* equation [i.e., Equation (10)] characterizes an optimal solution for FB-MDPs, which notably exhibits a bidirectional optimality dynamics, due to presence of $\mathbf{s}^-$ and $\mathbf{s}^+$ on the right-hand side. This requires that both dynamics should be jointly and *simultaneously* considered to obtain an optimal policy. We leverage these findings in devising our algorithm next.

## 4 Forward-Backward Multi-Objective RL

We now build upon the results in the previous section to develop an RL algorithm for multi-objective FB-MDP problems. Specifically, we devise a *Forward-Backward Step-Wise* (FB-SW) mechanism according to Remark 3.7 and Lemma 3.8. The mechanism comprises three steps: (i) *forward evaluation*, in which the forward dynamics is evaluated by generating actions using the policy $\mathbf{a}_t \sim \pi_{\boldsymbol{\theta}}(\cdot|\mathbf{s}_t)$; (ii) *backward evaluation*, in which the backward dynamics is evaluated in a time-reversed way by leveraging the actions generated in the previous step; and (iii) *bidirectional learning*, employing a multi-objective optimization mechanism *with a suitable chronological order* to optimize the policy $\pi_{\boldsymbol{\theta}}(\cdot|\mathbf{s}_t)$ based on the experiences obtained from both the forward and backward dynamics. In the next sections, we utilize this general mechanism as an adaptable framework to devise an RL algorithm suited to FB-MDP problems.

Figures 1b and 1c (on page 2) outline such an algorithm.

## 4.1 The Forward-Backward Algorithm

According to Equations (5) and (7), the gradient of $\mathbf{J}(\boldsymbol{\theta})$ depends on the policy distribution $\pi_{\boldsymbol{\theta}}(\cdot|\cdot)$ in addition to the state-value functions $V^f(\cdot)$ and $V^b(\cdot,\cdot)$. For the policy distribution $\pi_{\boldsymbol{\theta}}(\cdot|\cdot)$, we consider an *actor agent* represented by a $\boldsymbol{\theta}$-parametric neural network (NN). For the forward state-value function $V^f(\cdot)$, we set a *forward-critic network* represented by a $\boldsymbol{\phi}$-parametric NN, denoted by $V_{\boldsymbol{\phi}}^f(\cdot)$. Moreover, we use a *backward-critic network* with a $\boldsymbol{\psi}$-parametric NN for the bidirectional state-value function, indicated by $V_{\boldsymbol{\psi}}^b(\cdot,\cdot)$. We must now align the evaluation and update procedures for the actor and critic networks with the FB-SW mechanism. In this regard, $\pi_{\boldsymbol{\theta}}(\cdot|\cdot)$ and $V_{\boldsymbol{\phi}}^f(\cdot)$ are evaluated during the forward-evaluation step of the FB-SW mechanism, $V_{\boldsymbol{\psi}}^b(\cdot,\cdot)$ is evaluated during the backward-evaluation step, then their values are employed to compute $\nabla_{\boldsymbol{\theta}}\mathbf{J}(\boldsymbol{\theta})$ and update $\pi_{\boldsymbol{\theta}}(\cdot|\cdot)$ during the forward-backward optimization step.

The update mechanism of actor policy $\pi_{\boldsymbol{\theta}}(\cdot|\cdot)$ depends on the forward and bidirectional state-value functions, i.e., $V_{\boldsymbol{\phi}}^f(\cdot)$ and $V_{\boldsymbol{\psi}}^b(\cdot,\cdot)$. As a consequence, we need to set some losses to also update these state-value functions. In line with Bellman's equation $V^f(\mathbf{s}_k) = \mathbb{E}_{\mathbf{s}_{k+1},\mathbf{a}_k|\mathbf{s}_k}\{\boldsymbol{r}^f(\mathbf{s}_k,\mathbf{a}_k) + \gamma V^f(\mathbf{s}_{k+1})\}$ and Temporal Difference (TD)-learning (Grondman et al., 2012), the following *forward-critic losses* are considered to update $\boldsymbol{\phi}$:

$$\sum_{k=1}^{T} A_{\boldsymbol{\phi},i}^f(\mathbf{s}_k,\mathbf{a}_k)^2, \quad \text{for} \quad i \in S_f, \tag{11}$$

where $A_{\boldsymbol{\phi},i}^f(\mathbf{s}_k,\mathbf{a}_k) = V_{\boldsymbol{\phi},i}^f(\mathbf{s}_k) - r_i^f(\mathbf{s}_k,\mathbf{a}_k) - \gamma V_{\boldsymbol{\phi},i}^f(\mathbf{s}_{k+1})$ are parametric representations of the so-called forward advantage functions. Conversely, we set the following *backward-critic losses* to update the parameter $\boldsymbol{\psi}$ based on the derived backward Bellman's equation [i.e., Equation (9)]:

$$\sum_{k=0}^{T-1} A_{\boldsymbol{\psi},i}^b(\mathbf{y}_{T-k},\mathbf{s}_{T-k},\mathbf{a}_{T-k})^2, \quad \text{for} \quad i \in S_b, \tag{12}$$

where $A_{\boldsymbol{\psi},i}^b(\mathbf{y}_{T-k},\mathbf{s}_{T-k},\mathbf{a}_{T-k}) = V_{\boldsymbol{\psi},i}^b(\mathbf{y}_{T-k},\mathbf{s}_{T-k}) - r_i^b(\mathbf{y}_{T-k},\mathbf{a}_{T-k}) - \gamma V_{\boldsymbol{\psi},i}^b(\mathbf{y}_{T-k-1},\mathbf{s}_{T-k-1})$ are the parametric bidirectional advantage functions.

Equations (5) and (7) indicate multiple directions for optimizing the actor, while Equations (11) and (12) show multiple losses for optimizing the forward / bidirectional critic networks. A straightforward approach to carry out multi-objective optimization involves using the scalarization technique, namely, obtaining a single-objective loss through a preference function (or scales) for different rewards. However, Pareto solutions cannot be necessarily obtained via this method (Kirlik & Sayın, 2014). As a consequence, tuning the scalarization settings might require a trial-and-error approach, which is sensitive to the selected setup. Instead, we use a scale-insensitive multi-objective optimization method (Schäffler et al., 2002; Amidzadeh, 2023) to devise a forward-backward RL algorithm. Accordingly, we employ Lemma 3.3 to formulate forward / bidirectional critic networks and a multi-objective actor agent shared between the forward and bidirectional critics.

### 4.1.1 Forward / Bidirectional Critic Networks

Equation (11) [Equation (12)] provides multiple losses for the forward (backward) critic network. By recalling Lemma 3.3, we formulate the *multi-objective* loss $K^f(\boldsymbol{\phi})$ [$K^b(\boldsymbol{\psi})$] by using the coefficients $\boldsymbol{\beta_f}$ ($\boldsymbol{\beta_b}$), so that a common *descent* direction is formulated for all forward (backward) critic losses. Accordingly, we have:

$$K^f(\boldsymbol{\phi}) = \sum_{j \in S_f} \beta_{f,j}^* \sum_{k=1}^{T} A_{\boldsymbol{\phi},j}^f(\mathbf{s}_k,\mathbf{a}_k)^2, \qquad K^b(\boldsymbol{\psi}) = \sum_{j \in S_b} \beta_{b,j}^* \sum_{k=0}^{T-1} A_{\boldsymbol{\psi},j}^b(\mathbf{y}_{T-k},\mathbf{s}_{T-k},\mathbf{a}_{T-k})^2, \tag{13}$$

where $\boldsymbol{\beta}_f^*$ and $\boldsymbol{\beta}_b^*$ are tuned by the following problems (see $Q_2$ of Lemma 3.3):

$$\boldsymbol{\beta}_f^* = \underset{\substack{\beta_j \ge 0 \\ \sum_{j \in S_f} \beta_j = 1}}{\mathrm{argmin}} \left\| \sum_{j \in S_f} \beta_j \nabla_{\boldsymbol{\phi}} \sum_{k=1}^{T} A_{\boldsymbol{\phi},j}^f(\mathbf{s}_k, \mathbf{a}_k)^2 \right\|^2, \qquad \boldsymbol{\beta}_b^* = \underset{\substack{\beta_j \ge 0 \\ \sum_{j \in S_b} \beta_j = 1}}{\mathrm{argmin}} \left\| \sum_{j \in S_b} \beta_j \nabla_{\boldsymbol{\psi}} \sum_{k=0}^{T-1} A_{\boldsymbol{\psi},j}^b(\mathbf{y}_{T-k}, \mathbf{s}_{T-k}, \mathbf{a}_{T-k})^2 \right\|^2.$$

(14)

These critic networks are then updated via TD-learning with the following Stochastic Gradient Descent (SGD) rules (Grondman et al., 2012):

$$\boldsymbol{\phi} \leftarrow \boldsymbol{\phi} - \mu_f \nabla_{\boldsymbol{\phi}} K^f(\boldsymbol{\phi}), \qquad \boldsymbol{\psi} \leftarrow \boldsymbol{\psi} - \mu_b \nabla_{\boldsymbol{\psi}} K^b(\boldsymbol{\psi}),$$

(15)

where $\mu_f$ and $\mu_b$ are the learning rates of the forward and bidirectional critic networks, respectively.

### 4.1.2 Actor Agent

We follow the same strategy as in the previous section to devise a single-policy multi-objective actor agent shared between the forward and backward processes. The following forward and backward gradients follow from Equations (5) and (7):

$$\nabla_{\boldsymbol{\theta}} \hat{J}_i^f(\boldsymbol{\theta}, \boldsymbol{\phi}) = \sum_{k=1}^{T} \nabla_{\boldsymbol{\theta}} \log \pi_{\boldsymbol{\theta}}(\mathbf{a}_k | \mathbf{s}_k) A_{\boldsymbol{\phi},i}^f(\mathbf{s}_k, \mathbf{a}_k),$$

$$\nabla_{\boldsymbol{\theta}} \hat{J}_j^b(\boldsymbol{\theta}, \boldsymbol{\psi}) = \sum_{k=0}^{T-1} \nabla_{\boldsymbol{\theta}} \log \pi_{\boldsymbol{\theta}}(\mathbf{a}_{T-k} | \mathbf{s}_{T-k}) A_{\boldsymbol{\psi},j}^b(\mathbf{y}_{T-k}, \mathbf{s}_{T-k}, \mathbf{a}_{T-k})$$

(16)

for $i \in S_f$ and $j \in S_b$. We then employ Lemma 3.3 to provide a simultaneous *ascent* direction for all forward / backward rewards. Note that, in contrast to the critic network, an ascent direction is desired as the actor maximizes the rewards. Hence, the multi-objective actor agent is updated by the following multi-objective SGD:

$$\boldsymbol{\theta} \leftarrow \boldsymbol{\theta} + \mu \Big( \sum_{j \in S_f} \beta_{\mathrm{act,j}} \nabla_{\boldsymbol{\theta}} \hat{J}_j^f(\boldsymbol{\theta}, \boldsymbol{\phi}) + \sum_{j \in S_b} \beta_{\mathrm{act,j}} \nabla_{\boldsymbol{\theta}} \hat{J}_j^b(\boldsymbol{\theta}, \boldsymbol{\psi}) \Big),$$

(17)

where $\mu$ is the learning rate of the actor agent, and

$$\boldsymbol{\beta}_{\mathrm{act}} = \underset{\{\beta_j\}_j}{\mathrm{argmin}} \left\| \sum_{j \in S_f} \beta_j \nabla_{\boldsymbol{\theta}} \bar{J}_j^f(\boldsymbol{\theta}) + \sum_{j \in S_b} \beta_j \nabla_{\boldsymbol{\theta}} \bar{J}_j^b(\boldsymbol{\theta}) \right\|^2,$$

$$\text{s.t.} \quad \beta_j \ge 0, \quad \sum_{j \in S_f \cup S_b} \beta_j = 1,$$

(18)

with

$$\nabla_{\boldsymbol{\theta}} \bar{J}_j^f(\boldsymbol{\theta}) := \mathbb{E}_{\boldsymbol{\phi}} \mathbb{E} \left\{ \sum_{k=1}^{T} \nabla_{\boldsymbol{\theta}} \log \pi_{\boldsymbol{\theta}}(\mathbf{a}_k | \mathbf{s}_k) A_{\boldsymbol{\phi},j}^f(\mathbf{s}_k, \mathbf{a}_k) \,\Big|\, \boldsymbol{\theta}, \boldsymbol{\phi} \right\} = \mathbb{E} \left\{ \nabla_{\boldsymbol{\theta}} \hat{J}_i^f(\boldsymbol{\theta}, \boldsymbol{\phi}) \,|\, \boldsymbol{\theta} \right\},$$

$$\nabla_{\boldsymbol{\theta}} \bar{J}_j^b(\boldsymbol{\theta}) := \mathbb{E}_{\boldsymbol{\psi}} \mathbb{E} \left\{ \sum_{k=0}^{T-1} \nabla_{\boldsymbol{\theta}} \log \pi_{\boldsymbol{\theta}}(\mathbf{a}_{T-k} | \mathbf{s}_{T-k}) A_{\boldsymbol{\psi},j}^b(\mathbf{y}_{T-k}, \mathbf{s}_{T-k}, \mathbf{a}_{T-k}) \,\Big|\, \boldsymbol{\theta}, \boldsymbol{\psi} \right\} = \mathbb{E} \left\{ \nabla_{\boldsymbol{\theta}} \hat{J}_j^b(\boldsymbol{\theta}, \boldsymbol{\psi}) \,|\, \boldsymbol{\theta} \right\}, \quad (19)$$

for $j \in S_b$ and $i \in S_f$. Note that, as opposed to the critic losses, we theoretically leverage the expected gradients $\nabla_{\boldsymbol{\theta}} \bar{J}_j^f(\boldsymbol{\theta})$ and $\nabla_{\boldsymbol{\theta}} \bar{J}_j^b(\boldsymbol{\theta})$ to optimize $\boldsymbol{\beta}_{\mathrm{act}}$ in Equation (18) [compare with Equation (14)]. This approach interestingly ensures that all forward and backward cumulative rewards – namely, $\{J_j^f(\boldsymbol{\theta})\}_{j \in |S|_f}$ and $\{J_i^b(\boldsymbol{\theta})\}_{i \in |S|_b}$ – monotonically increase at each iteration and, more importantly, facilitates the convergence of the FB-MOAC algorithm (please refer to Theorem 4.3 in the appendix for more details).

To estimate the expected gradients $\nabla_{\boldsymbol{\theta}} \bar{J}_j^f(\boldsymbol{\theta})$ and $\nabla_{\boldsymbol{\theta}} \bar{J}_j^b(\boldsymbol{\theta})$, we employ *Monte Carlo Sampling* (MCS) together with an *exponential moving average*, applied to $\nabla_{\boldsymbol{\theta}} \hat{J}_j^f(\boldsymbol{\theta}, \boldsymbol{\phi})$ and $\nabla_{\boldsymbol{\theta}} \hat{J}_j^b(\boldsymbol{\theta}, \boldsymbol{\psi})$. Specifically, we first implement

---

**Algorithm 1**  Pseudo-code of the Forward-Backward Multi-Objective Actor-Critic (FB-MOAC) algorithm.

---

1:  **for** $episode = 1$ **to** $E_{\max}$ **do**
2:      **Input:** Initial forward-backward state $(\mathbf{s}_1, \mathbf{y}_T)$
3:              Actor, forward-critic and backward-critic parameters: $\boldsymbol{\theta}$, $\boldsymbol{\phi}$ and $\boldsymbol{\psi}$
4:  **Step 1: Forward Evaluation:**
5:      **for** $t = 1$ **to** $T$ **do**
6:          Select $\mathbf{a}_t \sim \pi_{\boldsymbol{\theta}}(\cdot|\mathbf{s}_t)$, interact with environment
7:          Observe forward state $\mathbf{s}_{t+1}$ and forward rewards $\{r_j^f(\mathbf{s}_t, \mathbf{a}_t)\}_{j \in S_f}$
8:          Compute $\{A_{\boldsymbol{\phi},j}^f(\mathbf{s}_t, \mathbf{a}_t)\}_{j \in S_f}$ by forward state-value $\{V_{\boldsymbol{\phi},j}^f(\mathbf{s}_t)\}_{j \in S_f}$, Equation (11)
9:          Compute $\log\left(\pi_{\boldsymbol{\theta}}(\mathbf{a}_t|\mathbf{s}_t)\right)$
10:     **end for**
11: **Step 2: Backward Evaluation:**
12:     **for** $t = 1$ **to** $T$ **do**
13:         Observe backward state $\mathbf{y}_{T-t}$ and backward rewards $\{r_j^b(\mathbf{y}_{T-t}, \mathbf{a}_{T-t})\}_{j \in S_b}$ depending on the action drawn in step *Forward-Evaluation*
14:         Compute $\{A_{\boldsymbol{\psi},j}^b(\mathbf{y}_{T-t}, \mathbf{s}_{T-t}, \mathbf{a}_{T-t})\}_{j \in S_b}$ by bidirectional state-value $\{V_{\boldsymbol{\psi},j}^b(\mathbf{y}_{T-t}, \mathbf{s}_{T-t})\}_{j \in S_b}$, Equation (12)
15:     **end for**
16: **Step 3: Bidirectional Learning:**
17:     **Forward / bidirectional Critic Update**:
18:     Obtain $\boldsymbol{\beta}_f^*$ and $\boldsymbol{\beta}_b^*$ by Equation (14)
19:     Compute multi-objective forward-critic loss $K^f(\boldsymbol{\phi})$ and backward-critic loss $K^b(\boldsymbol{\psi})$
20:     Apply the rules:

$$\boldsymbol{\phi} \leftarrow \boldsymbol{\phi} - \mu_f \nabla_{\boldsymbol{\phi}} K^f(\boldsymbol{\phi}), \qquad \boldsymbol{\psi} \leftarrow \boldsymbol{\psi} - \mu_b \nabla_{\boldsymbol{\psi}} K^b(\boldsymbol{\psi})$$

21:     **Forward-Backward Actor Update**:
22:     Obtain $\boldsymbol{\beta}^*$ through Equation (18) and the outcomes of *episodic MCS-average*
23:     Compute stochastic forward and backward gradients $\nabla_{\boldsymbol{\theta}} \hat{J}_j^f(\boldsymbol{\theta}, \boldsymbol{\phi})$ and $\nabla_{\boldsymbol{\theta}} \hat{J}_j^b(\boldsymbol{\theta}, \boldsymbol{\psi})$ through Equation (16)
24:     Apply the SGD rule:

$$\boldsymbol{\theta} \leftarrow \boldsymbol{\theta} - \mu\bigg(\sum_{j \in S_f} \beta_{\mathrm{act,j}} \nabla_{\boldsymbol{\theta}} \hat{J}_j^f(\boldsymbol{\theta}, \boldsymbol{\phi}) + \sum_{j \in S_b} \beta_{\mathrm{act,j}} \nabla_{\boldsymbol{\theta}} \hat{J}_j^b(\boldsymbol{\theta}, \boldsymbol{\psi})\bigg)$$

25: **end for**

---

$N_{\mathrm{mcs}}$ distinct backward and forward critic networks with learnable parameters $\{\boldsymbol{\psi}_l\}_{l=1}^{N_{\mathrm{mcs}}}$ and $\{\boldsymbol{\phi}_l\}_{l=1}^{N_{\mathrm{mcs}}}$, respectively, and use the approximations

$$\nabla_{\boldsymbol{\theta}} \bar{J}_j^f(\boldsymbol{\theta}) \approx \frac{1}{N_{\mathrm{MCS}}} \sum_{l=1}^{N_{\mathrm{MCS}}} \mathbb{E}\left\{\nabla_{\boldsymbol{\theta}} \hat{J}_j^f(\boldsymbol{\theta}, \boldsymbol{\phi}_l)|\boldsymbol{\theta}\right\}, \qquad \nabla_{\boldsymbol{\theta}} \bar{J}_i^b(\boldsymbol{\theta}) \approx \frac{1}{N_{\mathrm{MCS}}} \sum_{l=1}^{N_{\mathrm{MCS}}} \mathbb{E}\left\{\nabla_{\boldsymbol{\theta}} \hat{J}_i^b(\boldsymbol{\theta}, \boldsymbol{\psi}_l)|\boldsymbol{\theta}\right\}.$$

In addition, we consider different episodes to take an exponential average with a smoothing factor $\gamma_{\mathrm{mov}}$ to estimate $\mathbb{E}\left\{\nabla_{\boldsymbol{\theta}} \hat{J}_j^f(\boldsymbol{\theta}, \boldsymbol{\phi}_l)|\boldsymbol{\theta}\right\}$ and $\mathbb{E}\left\{\nabla_{\boldsymbol{\theta}} \hat{J}_i^b(\boldsymbol{\theta}, \boldsymbol{\psi}_l)|\boldsymbol{\theta}\right\}$. We name this approach **episodic MCS-average**.

Figure 1b overviews the proposed Forward-Backward Multi-Objective Actor-Critic (FB-MOAC) algorithm, whereas Algorithm 1 provides its pseudo-code.

## 4.2  Convergence Analysis

In this section, we analytically characterize the convergence of the FB-MOAC algorithm. Our investigation starts by establishing some foundational assumptions, followed by the presentation of important theorems and corollaries. We then carry out the convergence analysis for the scenario where the expected rewards are *Lipschitz-smooth* (please refer to Appendix B for more details).

We emphasize that stochastic nature of FB-MDP affects the values of $\boldsymbol{\phi}$, $\boldsymbol{\psi}$ and $\boldsymbol{\theta}$, based on the SGD rules [Equations (15) and (17)], so they are treated as random variables. We now make the following assumptions.

**Assumption 1:** The estimations of state-value functions are unbiased up to the residual terms, i.e.,

$$\mathbb{E}\big\{V_{\phi,i}^f(\mathbf{s}) \mid \mathbf{s}, \boldsymbol{\theta}\big\} = V_i^f(\mathbf{s}) + \delta_i^f, \qquad i \in S_f, \quad \mathbf{s} \in \mathcal{S}$$

$$\mathbb{E}\big\{V_{\psi,j}^b(\mathbf{y},\mathbf{s}) \mid \mathbf{y}, \mathbf{s}, \boldsymbol{\theta}\big\} = V_j^b(\mathbf{y},\mathbf{s}) + \delta_j^b, \qquad j \in S_b, \quad (\mathbf{y},\mathbf{s}) \in \mathcal{Y} \times \mathcal{S},$$

where $\{\delta_i^f\}_{i \in S_f}$ and $\{\delta_j^b\}_{j \in S_b}$ are the forward and backward residuals arising from the approximation of the true value functions $V_i^f(\mathbf{s})$ / $V_j^b(\mathbf{s},\mathbf{y})$ by the neural network parameterization and the stochastic gradient updates (15).

**Assumption 2:** The forward and backward expected rewards $(J_j^f(\boldsymbol{\theta}),\ J_j^b(\boldsymbol{\theta}))$ are Lipschitz-smooth functions with constants $L_f$ and $L_b$, respectively, w.r.t. $\boldsymbol{\theta}$:

$$\left\|\nabla_{\boldsymbol{\theta}} J_j^f(\boldsymbol{\theta}') - \nabla_{\boldsymbol{\theta}} J_j^f(\boldsymbol{\theta})\right\| \le L_f \|\boldsymbol{\theta}' - \boldsymbol{\theta}\|, \ j \in S_f, \qquad \left\|\nabla_{\boldsymbol{\theta}} J_j^b(\boldsymbol{\theta}') - \nabla_{\boldsymbol{\theta}} J_j^b(\boldsymbol{\theta})\right\| \le L_b \|\boldsymbol{\theta}' - \boldsymbol{\theta}\|, \ j \in S_b.$$

Assumption 2 can be mapped to a set of assumptions related to the architecture of neural networks.

**Proposition 4.1.** *Let the actor be represented by a $\boldsymbol{\theta}$-parametric neural network, where all activation functions are Lipschitz-continuous, Lipschitz-smooth, and bounded both above and below. Moreover, assume that either the action space $\mathcal{A}$ is compact or actions sampled from the policy distribution $\pi_{\boldsymbol{\theta}}(\cdot|\cdot)$ are clipped. Assumption 2 holds for any family of distributions that are bounded whenever their parameters and input are bounded.*

*Proof.* Please refer to Proposition B.2 in Appendix B. □

**Assumption 3:** The stochastic forward / backward gradient:

$$\nabla \hat{\boldsymbol{J}}^{\mathrm{fb}}(\boldsymbol{\theta}, \boldsymbol{\phi}, \boldsymbol{\psi}) = \left[\big[\nabla_{\boldsymbol{\theta}} \hat{J}_j^f(\boldsymbol{\theta}, \boldsymbol{\phi})\big]_{j \in S_f}, \big[\nabla_{\boldsymbol{\theta}} \hat{J}_j^b(\boldsymbol{\theta}, \boldsymbol{\psi})\big]_{j \in S_b}\right],$$

has a conditional covariance bounded by a positive semi-definite matrix $\boldsymbol{B}$:

$$\mathbb{E}\left\{\nabla \hat{\boldsymbol{J}}^{\mathrm{fb}}(\boldsymbol{\theta}, \boldsymbol{\phi}, \boldsymbol{\psi})^\top \nabla \hat{\boldsymbol{J}}^{\mathrm{fb}}(\boldsymbol{\theta}, \boldsymbol{\phi}, \boldsymbol{\psi}) \mid \boldsymbol{\theta}\right\} - \nabla \boldsymbol{J}^{\mathrm{fb}}(\boldsymbol{\theta})^\top \nabla \boldsymbol{J}^{\mathrm{fb}}(\boldsymbol{\theta}) \preceq \boldsymbol{B},$$

where $\nabla \boldsymbol{J}^{\mathrm{fb}}(\boldsymbol{\theta}) = \left[\big[\nabla_{\boldsymbol{\theta}} j^f(\boldsymbol{\theta})\big]_{j \in S_f}, \big[\nabla_{\boldsymbol{\theta}} J_j^b(\boldsymbol{\theta})\big]_{j \in S_b}\right]$.

Note that the assumptions outlined in this context align with the those adopted in the literature on convergence analysis (Tian et al., 2023; Xiong et al., 2022; Zhou et al., 2022; Qiu et al., 2021).

We now need to present a definition for the convergence to locally Pareto-optimal solutions.

**Definition 4.2.** The parameter sequence $\{\boldsymbol{\theta}_i\}_{i=1}^I$ is said to converge to locally Pareto-optimal solutions (Zhou et al., 2022) if

$$\lim_{i \to \infty} \mathbb{E}\left\{\min_{\substack{\beta_j \ge 0 \\ \sum_j \beta_j = 1}} \left\|\sum_{j \in |S_f \cup S_b|} \nabla_{\boldsymbol{\theta}} J_j^{\mathrm{fb}}(\boldsymbol{\theta}^i)\beta_j\right\|^2\right\} \to 0,$$

where $\nabla J_j^{\mathrm{fb}}(\boldsymbol{\theta})$ is the $j$-th element of $\nabla \boldsymbol{J}^{\mathrm{fb}}(\boldsymbol{\theta})$, with $\nabla \boldsymbol{J}^{\mathrm{fb}}(\boldsymbol{\theta}) = \left[\big[\nabla_{\boldsymbol{\theta}} J_j^f(\boldsymbol{\theta})\big]_{j \in S_f}, \big[\nabla_{\boldsymbol{\theta}} J_j^b(\boldsymbol{\theta})\big]_{j \in S_b}\right]$.

We are now ready to present the main theorem and the resulting consequences related to convergence.

**Theorem 4.3.** *Assume the forward/backward state-value estimations, i.e., $\{V_{j,\phi}^f(\cdot)\}_{j \in S_f}$ and $\{V_{j,\psi}^b(\cdot)\}_{j \in S_b}$, following Assumption 1. Moreover, consider the forward/backward expected rewards, i.e., $\{J_j^f(\cdot)\}_{j \in S_f}$ and $\{J_j^b(\cdot)\}_{j \in S_b}$, and the forward/backward stochastic rewards, i.e., $\{\hat{J}_j^f(\cdot, \cdot)\}_{j \in S_f}$ and $\{\hat{J}_j^b(\cdot, \cdot)\}_{j \in S_b}$, complying with Assumptions 2 and 3, and $\beta_{\text{act}}$ being the solution of Equation (18). Finally, consider the SGD as in Equations (14) and (17) characterized by iteration number $i$ and actor learning rate $\{\mu_i\}_{i=1}^I$ with*

$$\mu_i \leq \min \left\{ \frac{1}{\max\{L_f, L_b\}}, \frac{1}{\max\{L_f, L_b\}\|\boldsymbol{B}\|} \left( \mathbf{1}^\top \left( \nabla \boldsymbol{J}^{\text{fb}}(\boldsymbol{\theta}^i)^\top \nabla \boldsymbol{J}^{\text{fb}}(\boldsymbol{\theta}^i) \right)^{-1} \mathbf{1} \right)^{-1} \right\},$$

*and $0 < \mu_I \leq \ldots \leq \mu_i \leq \ldots \leq \mu_1$, which generate the sequences $\{\phi^i\}_{i=1}^I$, $\{\psi^i\}_{i=1}^I$ and $\{\theta^i\}_{i=1}^I$. Then, we get:*

$$\frac{1}{I} \sum_{i=1}^I \mathbb{E}\left\{ \|\nabla \boldsymbol{J}^{\text{fb}}(\boldsymbol{\theta}^i)\beta_{\text{act}}^i\|^2 \right\} \leq \frac{\max\{L_f, L_b\}\|\boldsymbol{B}\|}{I} \sum_{i=1}^I \frac{\mu_i}{2 - \mu_i \max\{L_f, L_b\}}$$

$$+ \frac{2}{I \, \mu_I \, |S_f \cup S_b|} \sum_{j \in S_f \cup S_b} \mathbb{E}\left\{ J_j^{\text{fb}}(\boldsymbol{\theta}^I) - J_j^{\text{fb}}(\boldsymbol{\theta}^1) \right\}. \tag{20}$$

*Proof (sketch).* It can be shown that the episodic MCS-average approach ensures that the forward and backward expected rewards $\{J_j^f(\boldsymbol{\theta})\}_{j \in |S_f|}$ and $\{J_i^b(\boldsymbol{\theta})\}_{i \in |S_b|}$ constantly increase at each iteration based on Corollary 3.5 (see Theorem B.3). We leverage this fact and the characteristic features of Lipschitz-smooth rewards to guarantee the convergence to a locally Pareto-optimal solution. Appendix B.2 provides a complete proof. □

**Remark 4.4 (Sublinear convergence).** Theorem 4.3 implies the convergence to a locally Pareto-optimal solution (Zhou et al., 2022) with convergence rate of $\mathcal{O}(1/\sqrt{I})$ under the learning-rate scheduling $\mu_i = \mathcal{O}(1/\sqrt{i})$, where $I$ is the number of algorithm iterations – equivalently, the number of policy updates in FB-MOAC. This rate is notably consistent with that of single-objective actor-critic methods for forward-MDPs, which exhibit a convergence rate of $\mathcal{O}(1/\sqrt{I})$ (Fu et al., 2021).

**Remark 4.5 (Complexity).** The overall architectural complexity of the FB-MOAC algorithm is comparable to that of standard actor-critic approaches. However, the episodic MCS-average add-on makes FB-MOAC computationally different from standard algorithms. The computational burden introduced by this add-on depends on the number of critic agents. We have empirically determined that as few as three agents is sufficient to achieve desirable performance (see Section 5 for more details). Given that the convergence rate of FB-MOAC algorithm is comparable to the standard RL algorithms, the computational complexity required to achieve convergence is of the same order as that of standard forward-only methods.

### 4.3 Deriving the Pareto-Front

Our FB-MOAC algorithm is designed as a multi-objective framework where a single-policy agent interacts with **multiple** reward-specific critic networks. Crucially, these critics are updated through a non-linear mechanism with respect to the reward functions, as described by Remark 3.4 in addition to Equations (13) and (15). Hence, we consider the critics for developing a preference policy with respect to different rewards. To systematically explore the Pareto front, we introduce forward and backward preference parameters, $\boldsymbol{\epsilon}^f \in (0, 1]^{|S_f|}$, $\boldsymbol{\epsilon}^b \in (0, 1]^{|S_b|}$, which are used to re-scale the corresponding advantage functions:

$$A_i^f(\mathbf{s}_k, \mathbf{a}_k) = \epsilon_i^f \, r_i^f(\mathbf{s}_k, \mathbf{a}_k) + \gamma V_i^f(\mathbf{s}_{k+1}) - V_i^f(\mathbf{s}_k), \quad \text{for } i \in S_f$$

$$A_j^b(\mathbf{y}_{T-k}, \mathbf{s}_{T-k}, \mathbf{a}_{T-k}) = \epsilon_j^b \, r_j^b(\mathbf{y}_{T-k}, \mathbf{a}_{T-k}) + \gamma V_j^b(\mathbf{y}_{T-k-1}, \mathbf{s}_{T-k-1}) - V_j^b(\mathbf{y}_{T-k}, \mathbf{s}_{T-k}), \quad \text{for } j \in S_b. \tag{21}$$

We then apply the FB-MOAC algorithm with the updated advantage functions. Note that this re-scaling does not lead to a linear preference due to Remark 3.4. We thus employ different preference parameters to steer the learning process toward different regions of the Pareto front. It is also important to highlight that

the scalarization technique cannot be applied on the forward and backward rewards to formulate a single reward, since the resulting reward would depend on both the forward and backward states. As a result, a state-coupled FB-MDP would occur and Lemma 3.8 would no longer hold. This further motivates using multi-objective optimization to find the Pareto-optimal solutions.

The non-linear re-scaling mechanism described above allows to characterize the Pareto-front of a problem. Note that Theorem 4.3 guarantees convergence to (locally) Pareto-optimal solutions. On the other hand, Theorem B.3 ensures that the expected rewards monotonically increase for any preference policy. Consequently, the convergence to a (locally) Pareto-optimal solution is preserved regardless of the different preferences. These considerations explain that the mechanism in Equation (21) allows to derive locally Pareto-optimal solutions for the case of Lipschitz-smooth rewards. An evaluation of the proposed mechanism is provided in Appendix E.

## 5 Evaluation

FB-MDPs find application in stochastic optimal control problems driven by forward-backward stochastic differential equations (FB-SDEs) and networked systems (Zabihi et al., 2023). Accordingly, we evaluate FB-MOAC against the state of the art through diverse representative problems in these domains: mathematical finance, as an example of how a FB-SDE-driven stochastic control problem can be solved by FB-MOAC; and cache-assisted content delivery in wireless networks. Appendix D provides an additional use case in the context of computation offloading through an edge (cloud) server. The code of FB-MOAC is available at: `https://github.com/amidimohsen/FBMOAC`.

### 5.1 Use Case: Mathematical Finance

We consider an investment-consumption problem (Ma & Yong, 1999; El Karoui, 1997). In particular, we address a stochastic optimal control problem driven by a forward-backward stochastic differential equation (FB-SDE), which is then discretized according to the method in Appendix F to find an optimal solution with the FB-MOAC algorithm.

#### 5.1.1 System Model

A financial market consists of $n$ risky assets whose prices follow the following F-SDEs:

$$dp_n(t) = p_n(t)\big(r_n^{\mathrm{app}}(t)dt + \langle \boldsymbol{\sigma}_n^{\mathrm{vol}}(t), d\boldsymbol{\beta}(t)\rangle\big), \quad p_n(0) > 0,$$

for $n \in \{1, \dots, N\}$, where $\boldsymbol{\beta}(t) \in \mathbb{R}^N$ is the Wiener process with an identity diffusion matrix, $r_n^{\mathrm{app}}(t)$ is the instantaneous appreciation rate, and $\boldsymbol{\sigma}_n^{\mathrm{vol}}(t) \in \mathbb{R}^N$ is the asset volatility. A trader invests in risky assets by fractional investments $\{0 \leq \phi_n(t) \leq 1\}_{n=1}^N$ or borrow / lend money with an interest rate $r^{\mathrm{int}}(t)$. Hence, the wealth $w(t)$ of the trader with consumption plan $c(t)$ can be obtained by a F-SDE:

$$dw(t) = f^{\mathrm{drf}}\big(w(t), c(t), \{\phi_n(t)\}_n\big)dt + \sum_{n=1}^N w(t)\phi_n(t)\langle \boldsymbol{\sigma}_n^{\mathrm{vol}}(t), d\boldsymbol{\beta}(t)\rangle, \tag{22}$$

where $f^{\mathrm{drf}}$ is the respective drift function obtained as:

$$f^{\mathrm{drf}}\big(w(t), c(t), \{\phi_n(t)\}_n\big) = r^{\mathrm{int}}(t)w(t) + \sum_{n=1}^N w(t)\phi_n(t)\big(r_n^{\mathrm{app}}(t) - r_n^{\mathrm{int}}(t) - c(t)\big)$$

with $w(0) = w_0$ and $w_0$ the initial wealth. Note that $\{\phi_n(t)\}_{n=1}^N$ is called the investment portfolio, with $\sum_{n=0}^N \phi_n(t) = 1$.

An utility process $u(t)$ of the investor is then taken into account. This process at time $t$ depends on the consumption plan $c(t)$ and the future utility, and is described by the following backward SDE (B-SDE):

$$du(t) = -f^{\mathrm{gen}}\big(c(t), u(t), z(t)\big)dt + \langle \mathbf{z}(t), d\boldsymbol{\beta}(t)\rangle, \ u(T) = f^{\mathrm{fin}}(w(T)), \tag{23}$$

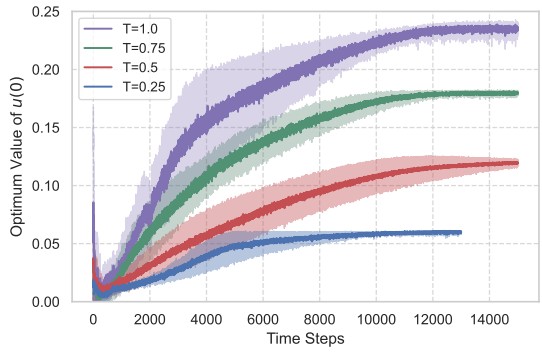

(a)

| Risky assets $N$ | FB-MOAC | | | Approach in Ji et al. (2022b; 2020) | | |
|---|---|---|---|---|---|---|
| | $T$=0.50 | $T$=0.75 | $T$=1.00 | $T$=0.50 | $T$=0.75 | $T$=1.00 |
| 10 | 0.121 | 0.182 | 0.241 | 0.122 | 0.182 | 0.242 |
| 20 | 0.122 | 0.181 | 0.240 | 0.122 | 0.178 | 0.242 |
| 50 | 0.120 | 0.180 | 0.235 | 0.121 | 0.181 | 0.237 |

(b)

Figure 2: Evaluation in a stochastic optimal control problem: optimal investor utility $u(0)$ (a) for $N = 50$ risky assets over different horizons and (b) comparison against the state of the art.

where $f^{\text{gen}}(\cdot)$ is the generator function, $\mathbf{z} \in \mathbb{R}^N$ is the control process of the backward dynamics, $T$ is the finite horizon, and $f^{\text{fin}}(\cdot)$ is the final utility function. The objective of this problem is to optimize the initial backward state $u(0)$ by designing an optimal portfolio and consumption plan. This is formulated based on the following stochastic optimal control problem:

$$\max_{\{\phi_n(t)\}_{n=0}^N,\, c(t)} \quad \mathbb{E}\big\{f^{\text{fin}}(w(T)) + \int_0^T f^{\text{gen}}(c(t), u(t), z(t))dt\big\},$$
$$\text{s.t.} \qquad \text{FB-SDE (22) and (23).} \tag{24}$$

To apply FB-MOAC, we discretize this stochastic optimal control problem with the Euler-Maruyama scheme (Kloeden & Platen, 1992) (see Appendix F for more details). The forward state is set to the wealth $w(t)$ [i.e., $s(t) = w(t)$], while the backward state to the utility process $u(t)$ [i.e., $y(t) = u(t)$]; the action is on the investment portfolio $\{\phi_i(t)\}_{i=1}^N$, the control process of the backward dynamics, and the consumption plan $c(t) \geq 0$, i.e., $\mathbf{a}(t) = [\{\phi_i(t)\}_{i=1}^N,\, c(t),\, z(t)]$; finally, the *cumulative* backward reward is the initial backward state $u(0)$ expressed by $\mathbb{E}\big\{f^{\text{fin}}(w(T)) + \int_0^T f^{\text{gen}}(c(t), u(t), z(t))dt\big\}$. Note that the discretization of the B-SDE in Equation (24) leads to a backward-MDP due to the backward flow of action information (see Remark 3.6). This motivates the use the of FB-MOAC algorithm. We thus partition the time interval $[0, T]$ into $N$ sub-intervals $[t_{k-1}, t_k)$ for $k \in \{1, \dots, N\}$, each sub-interval with length $\Delta t = \frac{T}{N^{\text{dis}}}$, where $t_0 = 0$ and $t_{N^{\text{dis}}} = T$. By applying this discretization, the backward cumulative reward leads to:

$$R^b(\mathbf{a}, \boldsymbol{y}) = \mathbb{E}\Big\{f^{\text{fin}}(w(t_{N^{\text{dis}}})) + \sum_{i=0}^{N^{\text{dis}}} f^{\text{gen}}\big(c(i\Delta t), u(i\Delta t), z(i\Delta t)\big)\Delta t\Big\}, \tag{25}$$

where $\mathbf{a} = \{\mathbf{a}(t_k)\}_{k=1}^{N^{\text{dis}}}$ and $\boldsymbol{y} = \{y(t_k)\}_{k=1}^{N^{\text{dis}}}$. Further, the F-SDE in Equation (22) maps to the following forward MDP:

$$w(t_{k+1}) - w(t_k) = f^{\text{drf}}\big(w(t_k), c(t_k), \{\phi_n(t_k)\}_n\big)\Delta t + \sum_{n=1}^N w(t_k)\phi_n(t_k)\langle \boldsymbol{\sigma}_n^{\text{vol}}(t_k), \Delta\boldsymbol{\beta}(t_k)\rangle, \tag{26}$$

where $\Delta\boldsymbol{\beta}(t_k)$ is a normal random variable with zero mean and variance $\Delta t$, and the B-SDE in Equation (23) maps to the following backward MDP:

$$u(t_k) = u(t_{k+1}) + f^{\mathrm{gen}}(c(t_{k+1}), u(t_{k+1}), z(t_{k+1}))\Delta t, \quad u(t_{N^{\mathrm{dis}}}) = f^{\mathrm{fin}}(w(t_{N^{\mathrm{dis}}})), \tag{27}$$

The respective sequential decision-making problem is then expressed as follows:

$$\max_{\boldsymbol{\theta}} \quad R^b(\mathbf{a}, \boldsymbol{y}),$$
$$\text{s.t.} \quad \left\{ \mathbf{a}(t_k) \sim \pi_{\boldsymbol{\theta}}(\cdot | s(t_k)), \quad \text{FB-MDP of Equations (26) − (27)} \right\}.$$

### 5.1.2 Experiment Setup and Hyper-parameters

We use the same settings as those in (Ji et al., 2022a). The number of assets is $N \in \{10, 20, 50\}$, the generator function $f^{\mathrm{gen}}(c(t), u(t), z(t)) = -0.05u(t) + c(t) - c(t)^2$, the final utility function $f^{\mathrm{fin}}(x) = \exp(-x)$, the interest rate $r^{\mathrm{int}}(t) = 0.03$, the appreciation rate $r^{\mathrm{app}}(t) = 0.05$, the volatility $\boldsymbol{\sigma}_n^{\mathrm{vol}} = 0.1\mathbf{I}_n$ for $n \in \{1, \ldots, N\}$, the finite horizon $T \in \{0.5, 0.75, 1.0\}$, and the initial wealth $x_0 = 100$.

As this problem only entails a backward reward, we only establish the backward-critic network; moreover, we set $N_{\mathrm{MCS}} = 1$, the number of neurons in the hidden layer for the actor and critics to 8, the actor and bidirectional critic learning rates $2 \times 10^{-2}$, and the smoothing factor $\gamma_{\mathrm{mov}} = 1$. We use the Dirichlet distribution for $\{\phi_n(t)\}_{n=0}^N$ to jointly motivate the exploration and to satisfy $\sum_{n=0}^N \phi_n(t) = 1$. Finally, the rectified linear unit (ReLU) activation function is used for the neuron connections, the number of neurons in the hidden layer for the actor and critics is 100, the actor and forward / bidirectional critic learning rates are $3 \times 10^{-4}$, and the smoothing factor is $\gamma_{\mathrm{mov}} = 0.95$.

### 5.1.3 Performance Evaluation

Figure 2a shows the performance of FB-MOAC as a function of time steps for different values of finite horizon $T \in \{0.5, 0.75, 1.0\}$. For comparison purposes, we consider the approaches in (Ji et al., 2022b; 2020), which develop deep learning methods by focusing on stochastic control theory and incorporate the system dynamics a priori for the optimization. In contrast, FB-MOAC learns multivariate rewards for FB-MDPs *without* knowing the transition probability of the underlying dynamics. Table 2b compares FB-MOAC against state of the art in terms of the optimal initial investor utility $u(0)$. The solution by FB-MOAC is close to the values obtained by (Ji et al., 2022b; 2020) for different values of $T$ *despite* treating the system dynamics as a black-box during the learning process. This demonstrates the ability of the proposed algorithm to find an optimal solution for environments characterized as FB-MDPs, thereby broadening its application to a variety of stochastic optimal control problems described by FB-SDEs.

## 5.2 Case Study: Edge Caching

We now consider a real-world forward-backward multi-task problem in the context of edge caching (Nomikos et al., 2022). For conciseness, the rest of the section omits details that can be found in Appendix C.

### 5.2.1 System Model

The environment of this experiment is a wireless network with cache-equipped Base-Stations (BSs) The environment also includes a library containing $N$ different content items as well as fixed mobile users requesting these items from the cellular network. The network operates over time slots with a *discrete* index $t \in \{1, \ldots, T\}$, where $T$ is the total duration of the operation. The network thus addresses that user requests at the beginning of each time slot. Content items have different *popularity* $\{p_n^{\mathrm{pop}}(t)\}_{n=1}^N$, where $p_n^{\mathrm{pop}}(t)$ is the probability that content $n$ is requested by a randomly selected user at time $t$. The goal is to satisfy as many users as possible during the network operation. At the beginning of each time-slot, the BSs cache the most popular content items with probability $\{p_n^{\mathrm{cach}}(t)\}_{n=1}^N$ and simultaneously *multicast* them toward users by consuming content-specific radio resources $\{w_n(t)\}_{n=1}^N$. The transmission at time-slot $t$ completes within $d(t)$ seconds. We thus denote the system action parameters by the vector $\mathbf{a}(t)$, which depends on

the content-specific bandwidth allocation and cache placement at BSs, i.e., $\mathbf{a}(t) = [\{p_n^{\text{cach}}(t)\}_{n=1}^N, \{w_n(t)\}_{n=1}^N]$. A multicast outage may occur with probability $\{O_n(\mathbf{a}(t),t)\}_{n=1}^N$. As a result, certain users fail to receive the requested content in a given time slot and their request is deferred to the subsequent one. Hence, each time-slot sees a distribution of users accounting for the repeated requests and a distribution describing the new preferences toward content items. This leads to a time-varying model for the request probability $p_n^{\text{req}}(t)$ of content $n$:

$$p_n^{\text{req}}(t) = \underbrace{p_n^{\text{req}}(t-1)O_n\big(\mathbf{a}(t-1),t-1\big)}_{\text{repeated request}} + \underbrace{p_n^{\text{pop}}(t)\sum_{m=1}^N \big(1 - O_m\big(\mathbf{a}(t-1),t-1\big)\big)p_m^{\text{req}}(t-1)}_{\text{new request based on the popularity}}. \quad (28)$$

Note that $p_n^{\text{req}}(t)$ indicates the request probability of content $n$ averaged over all users. Then, it can be simply verified that $\sum_{n=1}^N p_n^{\text{req}}(t) = 1$, considering $\sum_{n=1}^N p_n^{\text{pop}}(t) = 1$. Equation (28) therefore represents a **forward dynamics**, with the forward state vector $\mathbf{s}(t) = \mathbf{p}^{\text{req}}(t)$ and the action vector $\mathbf{a}(t) = [\{p_n^{\text{cach}}(t)\}_{n=1}^N, \{w_n(t)\}_{n=1}^N]$.

A request for a content item is repeated across several time-slots until successfully fulfilled, resulting in an expected latency $L_n(t)$ for successful delivery of content $n$. For this quantity, a time-varying dynamics can be derived by the law of total expectation as follows:

$$L_n(\mathbf{a}(t),t) = \Big(d(t) + L_n(\mathbf{a}(t+1),t+1)\Big)O_n(\mathbf{a}(t),t) + \frac{d(t)}{2}\big(1 - O_n(\mathbf{a}(t),t)\big), \quad L_n(\mathbf{a}(T),T) = 0, \quad (29)$$

where $d(t)$ is the duration of time-slot $t$ in seconds, and we have $L_n(\mathbf{a}(T),T) = 0$ since system operations finish at time $t = T$ and the users do not need to wait any longer. Equation (29) represents a **backward dynamics**, with the backward state vector $\mathbf{y}(t) = \mathbf{L}(\mathbf{a}(t),t)$ and the action vector $\mathbf{a}(t)$. Note that this model fully captures the trade-offs involved in the delay dynamics and differs from the conventional formulation that does not provide a comprehensive model when accounting for successive slots; for the delivery without outage, the expected latency simply becomes $L_n(t) = \frac{d(t)}{2}$, as its realizations follow a uniform distribution with values between 0 and $d(t)$.

Equation (29) may suggest that it is possible to convert it to a standard forward dynamics. For this purpose, we can consider a variable transformation $K_n(T-t) := L_n(\mathbf{a}(t),t)$ as well as a time transformation $t' := T - t$. We can then obtain the following forward dynamics on $K_n(t')$:

$$K_n(t') = \big(d(T-t') + K_n(t'-1)\big)O_n(\mathbf{a}(T-t'),T-t') + \frac{d(T-t')}{2}\big(1 - O_n(\mathbf{a}(T-t'),T-t')\big), \quad \text{for } t' \geq 1,$$

with $K_n'(0) = 0$. However, this shows a non-standard MDP, as the state $K_n(t')$ depends on the far future of action $\mathbf{a}_n(T-t')$ that cannot be revealed by moving forward in time. This argument aligns with Remark 3.6.

Equation (29) also shows that for a full-error transmission scheme (i.e., with the outage equal to one) $L_n(t) = d(t) + L_n(t+1)$ holds, which means that the expected latency maximally accumulates as one goes backwards in time. This is expected, as no successful receptions take place. Moreover, it is worth stressing that minimizing the expected latency in Equation (29) enables to *optimally* keep track of the *precise* time slot at which requests are finally fulfilled. Alternatively, one could track the service time of requests to prioritize those that have waited longer, or track for the failed / succeeded content transmissions. However, these policies do not completely map to the tracking of overall latency, and oversimplify the problem. Consequently, they fail to account for the complex interactions within the system, leading to a sub-optimal solution. The evaluation in Section 5.2.3 empirically proves this. Therefore, the problem is instead modeled as a FB-MDP, coupling forward and backward dynamics through system actions, where the action space is $[0,1]^n \times [0,\infty)^n$ with $n$ as the number of content items.

Three widely-used *network performance metrics* (Li et al., 2018b) are considered as reward functions to design an optimal policy: the quality of service $r_{\text{QoS}}(\cdot)$; the total bandwidth consumption $r_{\text{BW}}(\cdot)$; and the overall expected latency $r_{\text{Lat}}(\cdot)$. The QoS determines the overall probability of unsatisfied UEs and is given by $r_{\text{QoS}}(t) = -\sum_{n=1}^N p_n^{\text{req}}(t)O_n\big(\mathbf{a}(t),t\big)$, $0 \leq -r_{\text{QoS}}(t) \leq 1$, namely, the likelihood of a UE request

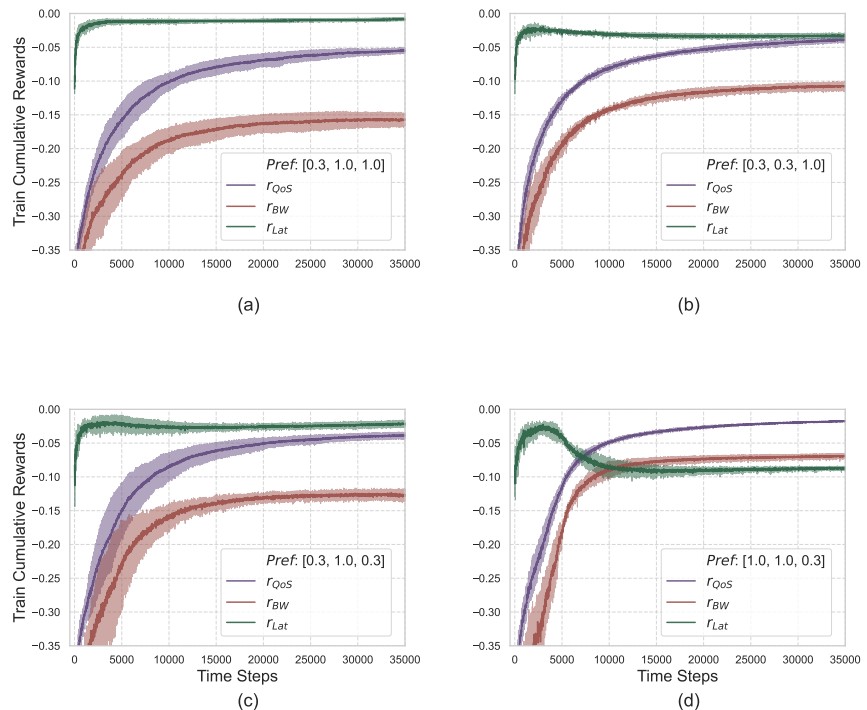

Figure 3: Pareto-optimal solutions in the edge caching use case for different settings of the preference parameters $\epsilon = [\epsilon^f, \epsilon^b]$: (a) $[0.3, 1.0, 1.0]$, (b) $[0.3, 0.3, 1.0]$ , (c) $[0.3, 1.0, 0.3]$ , and (d) $[1.0, 1.0, 0.3]$ . Note that these solutions are Pareto-optimal as none of them dominates the others.

remaining unfulfilled during the multicast transmission at time-slot $t$. The total bandwidth consumption is $r_{\mathrm{BW}}(t) = -W\big(\mathbf{a}(t), t\big) = -\sum_{n=1}^{N} w_n(t)$, where $W(\cdot)$ represents the total bandwidth consumption for the network. Finally, the overall expected latency is $r_{\mathrm{Lat}}(t) = -\sum_{n=1}^{N} p_n^{\mathrm{req}}(t) L_n(\mathbf{a}(t), t)$, with $L_n(t)$ obtained from Equation (29). Note that these rewards compete with each other. For instance, reducing $r_{\mathrm{BW}}$ requires increasing $w_n$ which, in turn, decreases the outage $O_n$. Furthermore, a decrease in $O_n$ makes $r_{\mathrm{QoS}}$ grow but reduces the latency $L_n$ which, in turn, increases $r_{\mathrm{lat}}$.

Clearly, $r_{\mathrm{QoS}}(t)$ and $r_{\mathrm{BW}}(t)$ relate to the forward state, and constitute the forward bivariate reward function $\boldsymbol{r}^f(t) = [r_{\mathrm{QoS}}(t), r_{\mathrm{BW}}(t)]$. Instead, $r_{\mathrm{Lat}}(t)$ relates to the backward state, and constitutes a backward univariate reward function $r^b(t) = r_{\mathrm{Lat}}(t)$. The respective sequential decision-making problem is then expressed as follows:

$$\max_{\boldsymbol{\theta}} \quad \mathbb{E}\left\{\sum_{t=1}^{T} \gamma^{t-1} \left[\boldsymbol{r}^f(t),\ r^b(t)\right]\right\},$$

$$\text{s.t.} \quad \left\{\mathbf{a}_t \sim \pi_{\boldsymbol{\theta}}(\cdot | \boldsymbol{s}_t), \quad \text{FB-MDP } 28 - 29\right\}.$$

### 5.2.2  Experiment Setup and Hyper-parameters

We select the following parameters for the considered environment. The number of content items is set to $N = 200$, the spatial intensity of the BSs to $\lambda_{\mathrm{bs}} = 100$ points / km$^2$, and the transmission rate to 1 Mbps. The total number of time slots is $T = 256$. For the content popularity, we use time-varying Zipf distributions (Li et al., 2018a).

As for FB-MOAC, three separate sets of NNs representing the multi-objective actor in addition to the forward-critic and the backward-critic networks. We use $N_{\mathrm{MCS}} = 4$ many NNs for the forward critic as well as for the backward critic. The forward critic outputs two values representing the reward-specific state-value

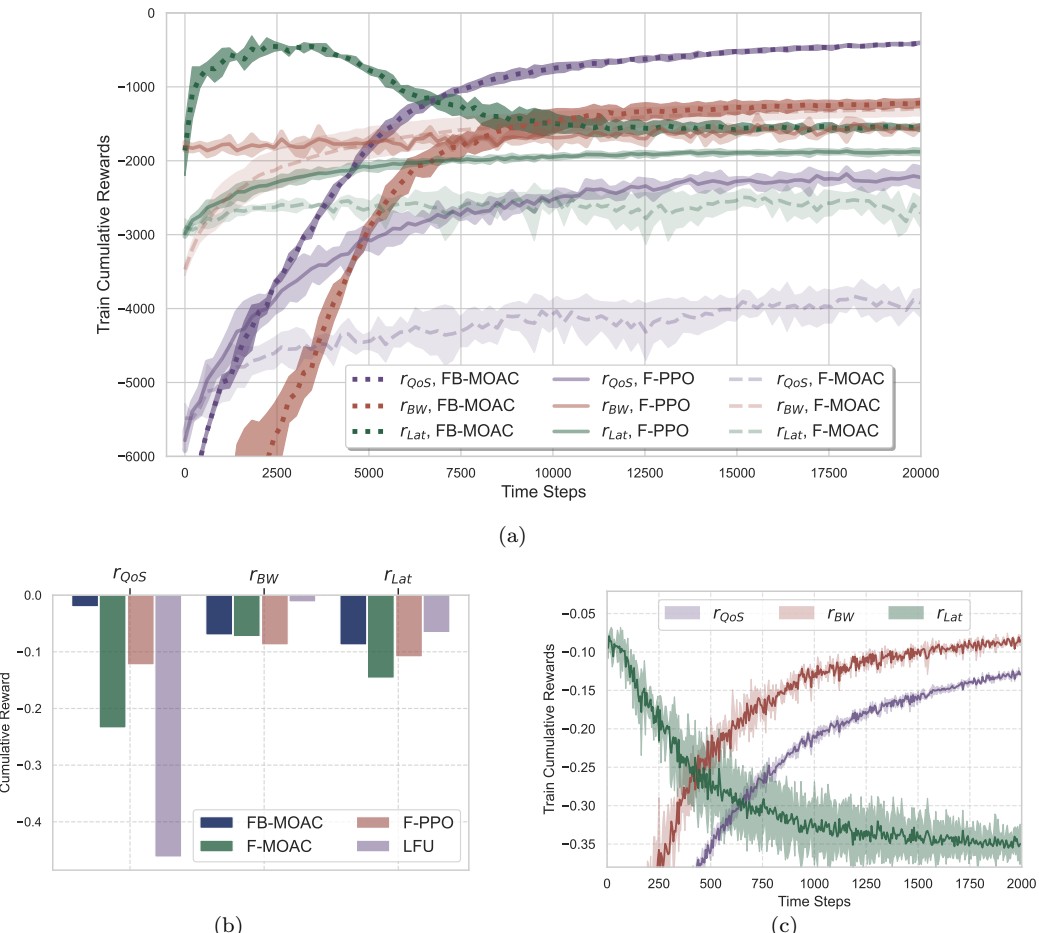

Figure 4: Performance of FB-MOAC for the edge caching use case: (a) performance comparison against forward learning with F-PPO, F-MOAC, and LFU; (b) comparison between the test solution of FB-MOAC with those of F-PPO, F-MOAC, and LFU; (c) learning in absence of backward optimization.

functions $V_{\phi, j}^{f}(\cdot)$, related to $r_{\text{QoS}}(\cdot)$ and $r_{\text{BW}}(\cdot)$. On the other hand, the backward critic outputs one value representing the state-value functions $V_{\psi}^{b}(\cdot)$, related to $r_{\text{Lat}}(\cdot)$. We set the actor and critic learning rates to $4 \times 10^{-4}$ and the discount factor to $\gamma = 0.92$.

To leverage the convergence result of Theorem 4.3, we assess whether a neural network can be used for this problem so as to give Lipschitz-smooth expected rewards. For this purpose, we resort to Proposition 4.1. The actions $\{p_n^{\text{cach}}(t), w_n(t)\}_{n=1}^{N}$ can be parameterized by a neural network with Sigmoid activation functions and can be sampled from LogNormal (for $w_n$) and Dirichlet (for $p_n^{\text{cach}}$) distributions. They are also bounded, as $0 < p_n < 1$ and $w_n$ is clipped (though it is not initially bounded) to avoid a large latency and bandwidth consumption. Therefore, Proposition 4.1 can be considered for the neural network of this problem, and $\log \pi_{\theta}(\mathbf{a}|\mathbf{s})$ inherits Lipschitz-smoothness in $\theta$.

### 5.2.3 Performance Evaluation

Figure 3 illustrates the learning results of FB-MOAC in deriving most of the Pareto-optimal solutions, i.e., the resulting solution in each plot does not dominate the others. Recall that Section 4.3 describes a mechanism for obtaining a Pareto-front; Appendix E further characterizes such a front for the use case considered here. For clarity, the performance metrics are *normalized* based on the value of $r_{\text{QoS}}$, so that they can be clearly shown in a single plot; more importantly, the value of $r_{\text{QoS}}$ shows the *average* percentage of failed requests.

As the results of forward and backward rewards eventually evolve into a stable solution, the actor and the forward / bidirectional critics are effectively learned.

We consider three baselines for comparison purposes: a widely-used rule-based approach for caching, the Least Frequently Used (LFU) strategy (Ahmed et al., 2013); and two learning-based algorithms by replacing the backward reward with a related one (for fairness) so that the backward MDP can be safely removed. Specifically, we manage the time slot during which requests are served by optimizing $d(t)$. We further leverage the fact that maximizing $r_{\mathrm{QoS}}$ reduces $r_{\mathrm{Lat}}$ based on Equation (29). Hence, we consider $r_{\mathrm{QoS}}$ and $r_{\mathrm{BW}}$ as forward rewards, replace the backward reward with optimizing $d(t)$, then use the baseline proximal policy optimization (PPO) algorithm (Schulman et al., 2017a) as well as a multi-objective extension of the baseline advantage actor critic (A2C) algorithm (Grondman et al., 2012). We term the solutions obtained with these strategies as F-PPO and F-MOAC (respectively), since they are developed for forward MDPs. Figure 4a compares the training performance of FB-MOAC against the F-PPO and F-MOAC baselines in terms of normalized rewards, while Figure 4b shows the solutions of these algorithms during test. We select a solution for FB-MOAC among different ones by prioritizing $r_{\mathrm{QoS}}$. Instead, we learn forward rewards and additionally optimize $d(t)$ for the two baselines (F-PPO and F-MOAC) to achieve a $r_{\mathrm{Lat}}$ comparable to that of FB-MOAC. The results show that FB-MOAC remarkably outperforms both F-MOAC and F-PPO in all rewards. This means that the FB-MOAC strategy can fulfill the content requests considerably better than forward-only strategies. Specifically, more than 15% of the content items are lost due to the quality of service in both F-PPO and F-MOAC, whereas the failure rate of FB-MOAC is only **2%**. Moreover, FB-MOAC gives a comparable or better policy than those obtained by F-PPO and F-MOAC. Finally, the solution of FB-MOAC Pareto-dominates those of F-PPO and F-MOAC. The LFU policy does not benefit from any preference settings with respect to the rewards, even though it is better than FB-MOAC from the bandwidth consumption $r_{\mathrm{BW}}$ perspective; unfortunately, it is very unreliable because 45% of the requests fail.

These findings show that minimizing the failure of transmissions (i.e., minimizing the outage) leads to a sub-optimal solution for the overall latency, also highlighting the importance of explicitly incorporating the backward MDP instead of trying to remove it through adjustments to the backward rewards. The results also demonstrate the effectiveness of the proposed FB-MOAC algorithm in solving the considered FB-MDP problem.

### 5.2.4 Ablation Study

We now conduct an ablation study to assess the benefit of the backward evaluation / optimization in FB-MOAC. For this purpose, we first disable the backward evaluation of the algorithm and only consider the forward actor and critic updates. Figure 4c shows the resulting rewards as a function of time steps, highlighting that $r_{\mathrm{lat}}$ does not improve over time. As a consequence, the results demonstrate the necessity of the backward evaluation / optimization in FB-MOAC.

We further carry out another ablation study to evaluate the impact of the multi-objective procedure in Equations (17) to (19) on the performance. For this purpose, we replace the proposed multi-objective optimization with a single-objective one through a linear scalarization technique. Specifically, we update the actor parameter $\boldsymbol{\theta}$ with the following rule:

$$\boldsymbol{\theta} \leftarrow \boldsymbol{\theta} - \mu \big( \sum_{j \in S_f} s_j^f \nabla_{\boldsymbol{\theta}} \hat{J}_j^f(\boldsymbol{\theta}, \boldsymbol{\phi}) + \sum_{j \in S_b} s_j^b \nabla_{\boldsymbol{\theta}} \hat{J}_j^b(\boldsymbol{\theta}, \boldsymbol{\psi}) \big),$$

where $\mathbf{s}^f = [\{s_j^f\}_{j \in S_f}] \in [0.1, 1]^{|S_f|}$ and $\mathbf{s}^f = [\{s_j^b\}_{j \in S_b}] \in [0.1, 1]^{|S_b|}$ are the scalarization settings. Figure 5 shows the train performance of this approach for different scalarization settings $[\mathbf{s}^f, s^b]$: $[0.3, 0.3, 1.0]$, $[1.0, 1.0, 0.3]$ and $[0.3, 1.0, 0.3]$. Clearly, the single-objective mechanism fails to provide stable solutions, in contrast with the proposed multi-objective approach (see also Figure 3).

## 6 Conclusion and Limitations

We introduced the notion of forward-backward Markov decision processes (FB-MDPs), a class of MDPs that cannot be expressed as standard MDPs. We then obtained an optimality condition for the solution

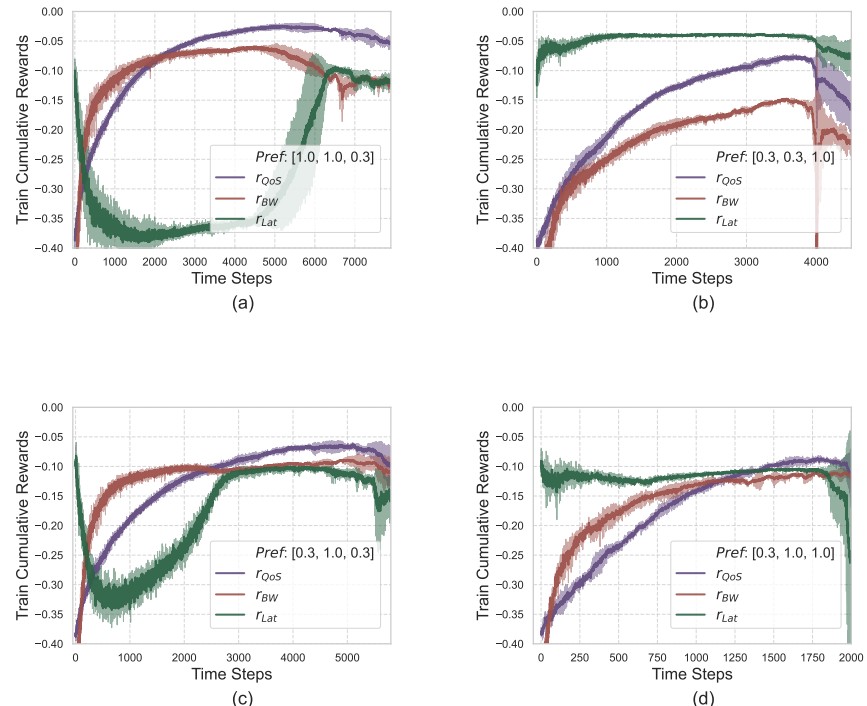

Figure 5: Train performance when single-objective optimization is used with different values of the scalarization settings $\mathbf{s} = [\mathbf{s}^f, s^b]$: (a) $[0.3, 0.3, 1.0]$, (b) $[1.0, 1.0, 0.3]$, (c) $[0.3, 1.0, 0.3]$ and (d) $[0.3, 1.0, 1.0]$.

of FB-MDPs and devised a multi-objective RL algorithm called FB-MOAC accordingly. We analytically characterized the optimality and convergence of FB-MOAC, then conducted an extensive evaluation in diverse use cases to demonstrate its effectiveness.

As a limitation, our mechanism targeted FB-MDP problems wherein forward and backward dynamics are purely coupled within the action space. Addressing fully-coupled FB-MDPs is an interesting direction for future work.

### Acknowledgments

This work was partially funded by the Research Council of Finland under grants number 354322 and 357533.

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

# A   Proof of Lemma 3.8

*Proof.* We simply denote the conditional expectation $\mathbb{E}\{\cdot|\boldsymbol{\theta}\}$ by $\mathbb{E}\{\cdot\}$ for convenience. Accordingly, we have:

$$\nabla_{\boldsymbol{\theta}}\mathbf{J}^b(\boldsymbol{\theta}) = \nabla_{\boldsymbol{\theta}}\,\mathbb{E}_{\mathrm{P}_{\boldsymbol{\theta}}(\boldsymbol{\tau})}\left\{\sum_{k=0}^{T-1}\gamma^k\boldsymbol{r}^b(\mathbf{y}_{T-k},\mathbf{a}_{T-k})\right\} \tag{30}$$

$$\overset{a}{=}\mathbb{E}_{\mathrm{P}_{\boldsymbol{\theta}}(\boldsymbol{\tau})}\left\{\sum_{k=0}^{T-1}\gamma^k\boldsymbol{r}^b(\mathbf{y}_{T-k},\mathbf{a}_{T-k})\sum_{k=0}^{T-1}\nabla_{\boldsymbol{\theta}}\log\pi_{\boldsymbol{\theta}}(\mathbf{a}_{T-k}|\mathbf{s}_{T-k})\right\}$$

$$\overset{b}{=}\mathbb{E}_{\mathrm{P}_{\boldsymbol{\theta}}(\boldsymbol{\tau})}\left\{\sum_{k=0}^{T-1}\nabla_{\boldsymbol{\theta}}\log\pi_{\boldsymbol{\theta}}(\mathbf{a}_{T-k}|\mathbf{s}_{T-k})\sum_{k'=k}^{T-1}\gamma^{k'-k}\boldsymbol{r}^b(\mathbf{y}_{T-k'},\mathbf{a}_{T-k'})\right\}$$

$$\overset{c}{=}\sum_{k=0}^{T-1}\underset{\substack{\mathbf{y}_{T-k}\\\mathbf{a}_{T-k}\\\mathbf{s}_{T-k}}}{\mathbb{E}}\left\{\nabla_{\boldsymbol{\theta}}\log\pi_{\boldsymbol{\theta}}(\mathbf{a}_{T-k}|\mathbf{s}_{T-k})\underbrace{\mathbb{E}\left\{\sum_{k'=k}^{T-1}\gamma^{k'-k}\boldsymbol{r}^b(\mathbf{y}_{T-k'},\mathbf{a}_{T-k'})\big|\mathbf{y}_{T-k},\mathbf{s}_{T-k},\mathbf{a}_{T-k}\right\}}_{Q^b(\mathbf{y}_{T-k},\mathbf{s}_{T-k},\mathbf{a}_{T-k})}\right\}$$

$$\overset{d}{=}\sum_{k=0}^{T-1}\underset{\substack{\mathbf{y}_{T-k}\\\mathbf{a}_{T-k}\\\mathbf{s}_{T-k}}}{\mathbb{E}}\left\{\nabla_{\boldsymbol{\theta}}\log\pi_{\boldsymbol{\theta}}(\mathbf{a}_{T-k}|\mathbf{s}_{T-k})\Big(Q^b(\mathbf{y}_{T-k},\mathbf{s}_{T-k},\mathbf{a}_{T-k})-V^b(\mathbf{y}_{T-k},\mathbf{s}_{T-k})\Big)\right\}$$

$$\overset{e}{=}\sum_{k=0}^{T-1}\underset{\substack{\mathbf{y}_{T-k}\\\mathbf{a}_{T-k}\\\mathbf{s}_{T-k}}}{\mathbb{E}}\left\{\nabla_{\boldsymbol{\theta}}\log\pi_{\boldsymbol{\theta}}(\mathbf{a}_{T-k}|\mathbf{s}_{T-k})\Big(\underset{\substack{\mathbf{y}_{T-k-1}\sim P_b(\cdot|\mathbf{y}_{T-k},\mathbf{a}_{T-k})\\\mathbf{s}_{T-k-1}\sim P(\cdot|\mathbf{s}_{T-k})}}{\mathbb{E}}\{\boldsymbol{r}^b(\mathbf{y}_{T-k},\mathbf{a}_{T-k})+\gamma V^b(\mathbf{y}_{T-k-1},\mathbf{s}_{T-k-1})\}-V^b(\mathbf{y}_{T-k},\mathbf{s}_{T-k})\Big)\right\}$$

$$=\sum_{k=0}^{T-1}\mathbb{E}\left\{\nabla_{\boldsymbol{\theta}}\log\pi_{\boldsymbol{\theta}}(\mathbf{a}_{T-k}|\mathbf{s}_{T-k})\big(\boldsymbol{r}^b(\mathbf{y}_{T-k},\mathbf{a}_{T-k})+\gamma V^b(\mathbf{y}_{T-k-1},\mathbf{s}_{T-k-1})-V^b(\mathbf{y}_{T-k},\mathbf{s}_{T-k})\big)\right\}$$

$$=\mathbb{E}_{\mathrm{P}_{\boldsymbol{\theta}}(\boldsymbol{\tau})}\left\{\sum_{k=0}^{T-1}\nabla_{\boldsymbol{\theta}}\log\pi_{\boldsymbol{\theta}}(\mathbf{a}_{T-k}|\mathbf{s}_{T-k})\mathbf{A}^b(\mathbf{y}_{T-k},\mathbf{s}_{T-k},\mathbf{a}_{T-k})\right\}, \tag{31}$$

where $V^b(\cdot,\cdot):\mathcal{Y}\times\mathcal{S}\to\mathbb{R}^{|S_b|}$, $Q^b(\cdot,\cdot,\cdot):\mathcal{Y}\times\mathcal{S}\times\mathcal{A}\to\mathbb{R}^{|S_b|}$ and $A^b(\cdot,\cdot,\cdot):\mathcal{Y}\times\mathcal{S}\times\mathcal{A}\to\mathbb{R}^{|S_b|}$ are the bidirectional state-value, backward action-value, and bidirectional advantage multivariate functions (respectively). We also have:

$$V^b(\mathbf{y}_{T-k},\mathbf{s}_{T-k}):=\mathbb{E}\left\{\sum_{k'=k}^{T-1}\gamma^{k'-k}\boldsymbol{r}^b(\mathbf{y}_{T-k'},\mathbf{a}_{T-k'})\big|\mathbf{y}_{T-k},\mathbf{s}_{T-k}\right\}. \tag{32}$$

For (a), we used $\nabla_{\boldsymbol{\theta}}\mathrm{P}_{\boldsymbol{\theta}}(\boldsymbol{\tau})=\mathrm{P}_{\boldsymbol{\theta}}(\boldsymbol{\tau})\,\nabla_{\boldsymbol{\theta}}\log\mathrm{P}_{\boldsymbol{\theta}}(\boldsymbol{\tau})$, and $\nabla_{\boldsymbol{\theta}}\log\mathrm{P}_{\boldsymbol{\theta}}(\boldsymbol{\tau})=\sum_{k=0}^{T-1}\nabla_{\boldsymbol{\theta}}\log\pi_{\boldsymbol{\theta}}(\mathbf{a}_{T-k}|\mathbf{s}_{T-k})$ based on Equation (3). For (b), we considered anti-causality, namely, that the current action does not affect the future

of backward rewards. For (c), we applied the definition of backward action-value functions is applied. For (d), $V^b(\mathbf{y}_{T-k}, \mathbf{s}_{T-k})$, does not affect the result since $\mathbb{E}_{\mathbf{a}_{T-k} \sim \pi_\theta(\cdot|\mathbf{s}_{T-k})}\{\nabla_\theta \log \pi_\theta(\mathbf{a}_{T-k}|\mathbf{s}_{T-k})\} = \mathbf{0}$. For (e), we derive the Bellman's equation for the backward action-value function as follows:

$$
\begin{aligned}
&Q^b(\mathbf{y}_{T-k}, \mathbf{s}_{T-k}, \mathbf{a}_{T-k}) \\
&= \int \mathbb{E}\left\{ \sum_{k'=k}^{T-1} \gamma^{k'-k} \boldsymbol{r}^b(\mathbf{y}_{T-k'}, \mathbf{a}_{T-k'}) \Big| \mathbf{y}_{T-k}, \mathbf{a}_{T-k}, \mathbf{y}_{T-k-1}, \mathbf{s}_{T-k}, \mathbf{s}_{T-k-1} \right\} P_b(\mathbf{y}_{T-k-1}|\mathbf{y}_{T-k}, \mathbf{a}_{T-k}) \times \\
&\hspace{8cm} P(\mathbf{s}_{T-k-1}|\mathbf{s}_{T-k}) d\mathbf{y}_{T-k-1} d\mathbf{s}_{T-k-1} \\
&\overset{\alpha}{=} \int \left( \boldsymbol{r}^b(\mathbf{y}_{T-k}, \mathbf{a}_{T-k}) + \gamma \mathbb{E}\left\{ \sum_{k'=k+1}^{T-1} \gamma^{k'-k} \boldsymbol{r}^b(\mathbf{y}_{T-k'}, \mathbf{a}_{T-k'}) \Big| \mathbf{y}_{T-k-1}, \mathbf{s}_{T-k-1} \right\} \right) P_b(\mathbf{y}_{T-k-1}|\mathbf{y}_{T-k}, \mathbf{a}_{T-k}) \times \\
&\hspace{8cm} P(\mathbf{s}_{T-k-1}|\mathbf{s}_{T-k}) d\mathbf{y}_{T-k-1} d\mathbf{s}_{T-k-1} \\
&= \mathbb{E}_{\substack{\mathbf{y}_{T-k-1} \sim P_b(\cdot|\mathbf{y}_{T-k}, \mathbf{a}_{T-k}) \\ \mathbf{s}_{T-k-1} \sim P(\cdot|\mathbf{s}_{T-k})}} \left\{ \boldsymbol{r}^b(\mathbf{y}_{T-k}, \mathbf{a}_{T-k}) + \gamma V^b(\mathbf{y}_{T-k-1}) \right\},
\end{aligned}
\tag{33}
$$

where for $(\alpha)$ we considered $(\mathbf{y}_{T-k-1}, \mathbf{s}_{T-k-1})$ as the only relevant information to compute the expectation $\mathbb{E}\left\{ \sum_{k'=k+1}^{T-1} \gamma^{k'-k} \boldsymbol{r}^b(\mathbf{y}_{T-k'}, \mathbf{a}_{T-k'}) \right\}$. The same strategy can be applied to obtain the Bellman's equation for the bidirectional state-value function as follows:

$$
V^b(\mathbf{y}_{T-k}, \mathbf{s}_{T-k}) = \mathbb{E}_{\substack{\mathbf{a}_{T-k} \sim \pi_\theta(\cdot|\mathbf{s}_{T-k}) \\ \mathbf{y}_{T-k-1} \sim P_b(\cdot|\mathbf{y}_{T-k}, \mathbf{a}_{T-k}) \\ \mathbf{s}_{T-k-1} \sim P(\cdot|\mathbf{s}_{T-k})}} \left\{ \boldsymbol{r}^b(\mathbf{y}_{T-k}, \mathbf{a}_{T-k}) + \gamma V^b(\mathbf{y}_{T-k-1}, \mathbf{s}_{T-k-1}) \,\big|\, \boldsymbol{\theta} \right\}.
$$

Note that no distinct forward and backward Bellman optimality equations exist for FB-MDPs. However, a Bellman Pareto-optimality equation can be found instead. For this purpose, we consider the forward and backward value functions to become stationary when the forward and backward transition probabilities are stationary. By recalling the notion of Pareto-optimality and Pareto front, we then define the Pareto-optimal forward and backward value functions as follows:

$$
\left[Q^{f^*}(\mathbf{s}, \mathbf{a}), Q^{b^*}(\mathbf{y}, \mathbf{s}, \mathbf{a})\right] \in \max_{\pi(\cdot|\cdot)} \left[Q^f(\mathbf{s}, \mathbf{a}), Q^b(\mathbf{y}, \mathbf{s}, \mathbf{a})\right], \qquad \left[V^{f^*}(\mathbf{s}), V^{b^*}(\mathbf{y})\right] \in \max_{\pi(\cdot|\cdot)} \left[V^f(\mathbf{s}), V^b(\mathbf{y}, \mathbf{s})\right],
$$

for all $(\mathbf{s}, \mathbf{y}, \mathbf{a}) \in \mathcal{S} \times \mathcal{Y} \times \mathcal{A}$, where $\left[Q^{f^*}(\mathbf{s}, \mathbf{a}), Q^{b^*}(\mathbf{y}, \mathbf{s}, \mathbf{a})\right]$ and $\left[V^{f^*}(\mathbf{s}), V^{b^*}(\mathbf{y}, \mathbf{s})\right]$ are the Pareto-optimal vector for the multi-objective optimization above. Now, we consider the following policy:

$$
\pi^*(\mathbf{a}|\mathbf{s}) = \begin{cases} 1, & \mathbf{a} \in \operatorname{argmax}_{\mathbf{a}}\left[Q^{f^*}(\mathbf{s}, \mathbf{a}), Q^{b^*}(\mathbf{y}, \mathbf{s}, \mathbf{a})\right] \\ 0, & \text{otherwise} \end{cases}.
$$

Note that here $\operatorname{argmax}_{\mathbf{a}}\left[Q^{f^*}(\mathbf{s}, \mathbf{a}), Q^{b^*}(\mathbf{y}, \mathbf{s}, \mathbf{a})\right]$ returns a set of vectors. We then have:

$$
\left[V^{f^*}(\mathbf{s}), V^{b^*}(\mathbf{y}, \mathbf{s})\right] \in \max_{\mathbf{a}} \left[Q^{f^*}(\mathbf{s}, \mathbf{a}), Q^{b^*}(\mathbf{y}, \mathbf{s}, \mathbf{a})\right].
$$

On the other hand, based on Equation (33) and the Bellman's equation for the forward action-value function, we can get:

$$
Q^{f^*}(\mathbf{s}, \mathbf{a}) = \mathbb{E}_{\mathbf{s}^+ \sim P_f(\cdot|\mathbf{s}, \mathbf{a})} \left\{ \boldsymbol{r}^f(\mathbf{s}, \mathbf{a}) + \gamma V^{f^*}(\mathbf{s}^+) \right\}
$$

$$
Q^{b^*}(\mathbf{y}, \mathbf{s}, \mathbf{a}) = \mathbb{E}_{\substack{\mathbf{y}^- \sim P_b(\cdot|\mathbf{y}, \mathbf{a}) \\ \mathbf{s}^- \sim P(\cdot|\mathbf{s})}} \left\{ \boldsymbol{r}^b(\mathbf{y}, \mathbf{a}) + \gamma V^{b^*}(\mathbf{y}^-, \mathbf{s}^-) \right\},
$$

where $\mathbf{y}^-$ is the backward state preceding $\mathbf{y}$ and $\mathbf{s}^+$ is the forward state following $\mathbf{s}$. We therefore obtain the following:

$$
\left[V^{f^*}(\mathbf{s}), V^{b^*}(\mathbf{y}, \mathbf{s})\right] \in \max_{\mathbf{a}} \left[ \mathbb{E}_{\mathbf{s}^+ \sim P_f(\cdot|\mathbf{s}, \mathbf{a})} \left\{ \boldsymbol{r}^f(\mathbf{s}, \mathbf{a}) + \gamma V^{f^*}(\mathbf{s}^+) \right\}, \mathbb{E}_{\substack{\mathbf{y}^- \sim P_b(\cdot|\mathbf{y}, \mathbf{a}) \\ \mathbf{s}^- \sim P(\cdot|\mathbf{s})}} \left\{ \boldsymbol{r}^b(\mathbf{y}, \mathbf{a}) + \gamma V^{b^*}(\mathbf{y}^-, \mathbf{s}^-) \right\} \right],
$$

Such an equation, referred to as the *Bellman Pareto-optimality equation*, provides a base to formulate dynamic programming algorithms for multi-objective FB-MOAC problems and motivates employing a multi-objective optimization framework.

$\square$

## B   Convergence of the FB-MOAC Algorithm

This section presents a comprehensive study on the convergence of the FB-MOAC algorithm. Our investigation starts by establishing of some foundational assumptions, followed by introducing some preliminaries. Subsequently, we analyze convergence for the scenario where the expected rewards are *Lipschitz-smooth*.

For the analysis, we need to emphasize that the stochastic nature of an FB-MDP affects the values of $\phi$, $\psi$ and $\theta$, based on the SGD rules [Equations (15) and (17)], so they are treated as random variables.

We now make the following assumptions.

**Assumption 1:** The estimations of the state-value functions are unbiased up to residual terms, i.e.,

$$\mathbb{E}\big\{V_{\phi,i}^f(\mathbf{s}) \mid \mathbf{s}, \boldsymbol{\theta}\big\} = V_i^f(\mathbf{s}) + \delta_i^f, \qquad i \in S_f, \quad \mathbf{s} \in \mathcal{S}$$

$$\mathbb{E}\big\{V_{\psi,j}^b(\mathbf{y}, \mathbf{s}) \mid \mathbf{y}, \mathbf{s}, \boldsymbol{\theta}\big\} = V_j^b(\mathbf{y}, \mathbf{s}) + \delta_j^b, \qquad j \in S_b, \quad (\mathbf{y}, \mathbf{s}) \in \mathcal{Y} \times \mathcal{S},$$

where $\{\delta_i^f\}_{i \in S_f}$ and $\{\delta_j^b\}_{j \in S_b}$ are the forward and backward residuals. These terms arise from approximating the true value functions $V_i^f(\mathbf{s}) \,/\, V_j^b(\mathbf{s}, \mathbf{y})$ by the neural network parameterization and the stochastic gradient updates in Equation (15).

The following corollary is a consequence of Assumption 1.

**Corollary B.1.** *Under Assumption 1, the expected forward / backward gradients in Equation (19) coincide with the corresponding reward gradients given in Equations (5) and (7), respectively.*

$$\nabla_{\boldsymbol{\theta}} \bar{J}_j^f(\boldsymbol{\theta}) = \nabla_{\boldsymbol{\theta}} J_j^f(\boldsymbol{\theta}), \;\; j \in S_f, \qquad \nabla_{\boldsymbol{\theta}} \bar{J}_j^b(\boldsymbol{\theta}) = \nabla_{\boldsymbol{\theta}} J_j^b(\boldsymbol{\theta}), \;\; j \in S_b$$

*Proof.* Based on Equations (16) and (19) we get:

$$\nabla_{\boldsymbol{\theta}} \bar{J}_j^f(\boldsymbol{\theta}) = \mathbb{E}\left\{\nabla_{\boldsymbol{\theta}} \hat{J}_j^f(\boldsymbol{\theta}, \phi) \,\big|\, \boldsymbol{\theta}\right\} = \sum_{k=1}^{T} \mathbb{E}\left\{\nabla_{\boldsymbol{\theta}} \log \pi_{\boldsymbol{\theta}}(\mathbf{a}_k|\mathbf{s}_k) A_{\phi,j}^f(\mathbf{s}_k, \mathbf{a}_k) \,\big|\, \boldsymbol{\theta}\right\}$$

$$= \sum_{k=1}^{T} \mathbb{E}_{\mathbf{s}_k, \mathbf{a}_k, \mathbf{s}_{k+1}|\boldsymbol{\theta}} \, \mathbb{E}\left\{\nabla_{\boldsymbol{\theta}} \log \pi_{\boldsymbol{\theta}}(\mathbf{a}_k|\mathbf{s}_k) A_{\phi,j}^f(\mathbf{s}_k, \mathbf{a}_k) \,\big|\, \boldsymbol{\theta}, \mathbf{s}_k, \mathbf{a}_k, \mathbf{s}_{k+1}\right\}$$

$$= \sum_{k=1}^{T} \mathbb{E}_{\mathbf{s}_k, \mathbf{a}_k, \mathbf{s}_{k+1}|\boldsymbol{\theta}} \left\{\nabla_{\boldsymbol{\theta}} \log \pi_{\boldsymbol{\theta}}(\mathbf{a}_k|\mathbf{s}_k) \mathbb{E}\left\{A_{\phi,j}^f(\mathbf{s}_k, \mathbf{a}_k) \,\big|\, \boldsymbol{\theta}, \mathbf{s}_k, \mathbf{a}_k, \mathbf{s}_{k+1}\right\}\right\}$$

$$\overset{(a)}{=} \sum_{k=1}^{T} \mathbb{E}_{\mathbf{s}_k, \mathbf{a}_k, \mathbf{s}_{k+1}|\boldsymbol{\theta}} \left\{\nabla_{\boldsymbol{\theta}} \log \pi_{\boldsymbol{\theta}}(\mathbf{a}_k|\mathbf{s}_k) \left(A_j^f(\mathbf{s}_k, \mathbf{a}_k) + (\gamma - 1)\delta_j^f\right)\right\}$$

$$= \mathbb{E}\left\{\sum_{k=1}^{T} \nabla_{\boldsymbol{\theta}} \log \pi_{\boldsymbol{\theta}}(\mathbf{a}_k|\mathbf{s}_k)\left(A_j^f(\mathbf{s}_k, \mathbf{a}_k) + (\gamma - 1)\delta_j^f(\boldsymbol{\theta})\right) \,\Big|\, \boldsymbol{\theta}\right\} \overset{(b)}{=} \nabla_{\boldsymbol{\theta}} J_j^f(\boldsymbol{\theta}) \qquad (34)$$

where (a) follows from Assumption 1 and the definition of advantage function $A_{\phi,j}^f(\mathbf{s}_k, \mathbf{a}_k) = r(\mathbf{s}_k, \mathbf{a}_k) + \gamma V_{\phi,j}^f(\mathbf{s}_{k+1}) - V_{\phi,j}^f(\mathbf{s}_k)$, while (b) stems from $\mathbb{E}\left\{\sum_{k=1}^{T} \nabla_{\boldsymbol{\theta}} \log \pi_{\boldsymbol{\theta}}(\mathbf{a}_k|\mathbf{s}_k)\delta_j^f(\boldsymbol{\theta}) \,\big|\, \boldsymbol{\theta}\right\} = 0$. Likewise, it can be shown that:

$$\nabla_{\boldsymbol{\theta}} \bar{J}_j^b(\boldsymbol{\theta}) = \mathbb{E}\left\{\sum_{k=1}^{T} \nabla_{\boldsymbol{\theta}} \log \pi_{\boldsymbol{\theta}}(\mathbf{a}_k|\mathbf{s}_k) A_j^b(\mathbf{y}_k, \mathbf{s}_k, \mathbf{a}_k) \,\Big|\, \boldsymbol{\theta}\right\} = \nabla_{\boldsymbol{\theta}} J_j^b(\boldsymbol{\theta}) \qquad (35)$$

$\square$

**Assumption 2:** The forward and backward expected rewards are Lipschitz-smooth functions with constants $L_f$ and $L_b$, respectively, w.r.t. $\boldsymbol{\theta}$:

$$\left\|\nabla_{\boldsymbol{\theta}} J_j^f(\boldsymbol{\theta}') - \nabla_{\boldsymbol{\theta}} J_j^f(\boldsymbol{\theta})\right\| \leq L_f \|\boldsymbol{\theta}' - \boldsymbol{\theta}\|, \qquad j \in S_f.$$

$$\left\|\nabla_{\boldsymbol{\theta}} J_j^b(\boldsymbol{\theta}') - \nabla_{\boldsymbol{\theta}} J_j^b(\boldsymbol{\theta})\right\| \leq L_b \|\boldsymbol{\theta}' - \boldsymbol{\theta}\|, \qquad j \in S_b.$$

Assumption 2 can be related to the assumptions on the architecture of the considered neural networks.

**Proposition B.2.** *Let the actor be represented by a neural network parameterized by $\boldsymbol{\theta} \in \Theta$, where all activation functions are* regular*, i.e., Lipschitz-smooth with constant $L_{\text{act}}^s$, Lipschitz-continuous with constant $L_{\text{act}}^c$, and bounded both above and below. Moreover, assume that either the action space $\mathcal{A}$ is compact or actions sampled from the policy distribution $\pi_{\boldsymbol{\theta}}(\cdot|\cdot)$ are clipped. Then Assumption 2 holds for any family of distributions that are bounded whenever its parameters and input are bounded.*

*Proof.* With respect to Equations (34) and (35), it suffices to evaluate Lipschitz-smoothness of the log-policy $\log \pi_{\boldsymbol{\theta}}(\mathbf{a}|\mathbf{s})$ since the sum of Lipschitz-smooth functions is itself Lipschitz-smooth. Let denote such a policy by a bi-variate function $f(\mathbf{a}, \mathbf{g}(\boldsymbol{\theta}, \mathbf{s})) : \mathcal{A} \times \mathcal{G} \to \mathbb{R}$, wherein $\mathbf{a}$ is the sampled actions being detached (so they do not depend on $\boldsymbol{\theta}$), $\mathbf{g} : \Theta \times \mathcal{S} \to \mathcal{G}$ is the output of the $\boldsymbol{\theta}$-parametric neural network, and $\mathcal{G}$ is the space of neural network output. Based on the chain rule, we have:

$$\left\|\nabla_\theta f(\mathbf{a}, \mathbf{g}(\theta_1, \mathbf{s})) - \nabla_\theta f(\mathbf{a}, \mathbf{g}(\theta_2, \mathbf{s}))\right\|$$
$$\leq \left\|\nabla_g f(\mathbf{a}, \mathbf{g}(\theta_1, \mathbf{s}))\right\| . \left\|\mathcal{J}_\theta \mathbf{g}(\theta_1, \mathbf{s}) - \mathcal{J}_\theta \mathbf{g}(\theta_2, \mathbf{s})\right\| + \left\|\mathcal{J}_\theta \mathbf{g}(\theta_2, \mathbf{s})\right\| . \left\|\nabla_g f(\mathbf{a}, \mathbf{g}(\theta_1, \mathbf{s})) - \nabla_g f(\mathbf{a}, \mathbf{g}(\theta_2, \mathbf{s}))\right\|$$
$$\overset{(a)}{\leq} \left\|\nabla_g f(\mathbf{a}, \mathbf{g}(\theta_1, \mathbf{s}))\right\| f_{\mathcal{N}}(L_{\text{act}}^s, L_{\text{act}}^c) \|\theta_1 - \theta_2\| + h_{\mathcal{N}}(L_{\text{act}}^c) \left\|\nabla_g f(\mathbf{a}, \mathbf{g}(\theta_1, \mathbf{s})) - \nabla_g f(\mathbf{a}, \mathbf{g}(\theta_2, \mathbf{s}))\right\|,$$

where $\mathcal{J}_\theta \mathbf{g}$ is the Jacobian of $\mathbf{g}$ w.r.t. $\boldsymbol{\theta}$, $\nabla_g f$ is the gradient of $f$ w.r.t. $\mathbf{g}$, and $f_{\mathcal{N}}$ and $h_{\mathcal{N}}$ are two functions depending on the architecture of the neural network. For (a), we leverage the fact that the composition of bounded and Lipschitz-smooth (continuous) functions remains Lipschitz-smooth (continuous) as all the activation functions are regular. Therefore, the output of neural network $\mathbf{g}$ is Lipschitz-smooth with constant $f_{\mathcal{N}}(L_{\text{act}}^s, L_{\text{act}}^c)$ and Lipschitz-continuous with constant $h_{\mathcal{N}}(L_{\text{act}}^c)$. Now, as $\mathbf{g}$ and $\mathbf{a}$ are bounded (if $\mathbf{a}$ is inherently unbounded, it can be clipped whenever the problem allows doing such), the norms of the first and second derivatives of the log-policy $f(\mathbf{a}, \mathbf{g}(\boldsymbol{\theta}, \mathbf{s}))$ with respect to the network output $\mathbf{g}(\boldsymbol{\theta}, \mathbf{s})$ are bounded for typical policy distributions such as Normal, LogNormal, Dirichlet and Beta. Therefore, there exists a parameter $A_1$ such that $\left\|f_g(\mathbf{a}, \mathbf{g}(\theta, \mathbf{s}))\right\| \leq A_1$. The boundedness of the norm of the second derivative is equivalent to Lipschitz-smoothness, thus, there also exists $A_2$ such that $\left\|\nabla_g f(\mathbf{a}, \mathbf{g}(\theta_1, \mathbf{s})) - \nabla_g f(\mathbf{a}, \mathbf{g}(\theta_2, \mathbf{s}))\right\| \leq A_2 \|\mathbf{g}(\theta_2, \mathbf{s}) - \mathbf{g}(\theta_1, \mathbf{s})\|$. Therefore, we get:

$$\left\|\nabla_\theta f(\mathbf{a}, \mathbf{g}(\theta_1, \mathbf{a})) - \nabla_\theta f(\mathbf{a}, \mathbf{g}(\theta_2, \mathbf{a}))\right\| \leq A_1 f_{\mathcal{N}}(L_{\text{act}}^s, L_{\text{act}}^c) \|\theta_1 - \theta_2\| + h_{\mathcal{N}}(L_{\text{act}}^c) A_2 \|\mathbf{g}(\theta_1, \mathbf{s}) - \mathbf{g}(\theta_2, \mathbf{s})\|$$
$$\leq A_1 f_{\mathcal{N}}(L_{\text{act}}^s, L_{\text{act}}^c) \|\theta_1 - \theta_2\| + h_{\mathcal{N}}(L_{\text{act}}^c)^2 A_2 \|\theta_1 - \theta_2\|$$
$$= \left(A_1 f_{\mathcal{N}}(L_{\text{act}}^s, L_{\text{act}}^c) + h_{\mathcal{N}}(L_{\text{act}}^c)^2 A_2\right) \|\theta_1 - \theta_2\|$$

Hence, the log-policy is Lipschitz-smooth with parameter $A_1 f_{\mathcal{N}}(L_{\text{act}}^s, L_{\text{act}}^c) + h_{\mathcal{N}}(L_{\text{act}}^c)^2 A_2$ and the statement follows. $\square$

**Assumption 3:** Consider the following stochastic forward / backward gradient:

$$\nabla \hat{\boldsymbol{J}}^{\text{fb}}(\boldsymbol{\theta}, \boldsymbol{\phi}, \boldsymbol{\psi}) = \left[\left[\nabla_{\boldsymbol{\theta}} \hat{J}_j^f(\boldsymbol{\theta}, \boldsymbol{\phi})\right]_{j \in S_f}, \left[\nabla_{\boldsymbol{\theta}} \hat{J}_j^b(\boldsymbol{\theta}, \boldsymbol{\psi})\right]_{j \in S_b}\right],$$

then, its conditional covariance is bounded by a positive semi-definite matrix $\boldsymbol{B}$:

$$\mathbb{E}\left\{\nabla \hat{\boldsymbol{J}}^{\text{fb}}(\boldsymbol{\theta}, \boldsymbol{\phi}, \boldsymbol{\psi})^\top \nabla \hat{\boldsymbol{J}}^{\text{fb}}(\boldsymbol{\theta}, \boldsymbol{\phi}, \boldsymbol{\psi}) \,\middle|\, \boldsymbol{\theta}\right\} - \nabla \boldsymbol{J}^{\text{fb}}(\boldsymbol{\theta})^\top \nabla \boldsymbol{J}^{\text{fb}}(\boldsymbol{\theta}) \preceq \boldsymbol{B},$$

where

$$\nabla \boldsymbol{J}^{\mathrm{fb}}(\boldsymbol{\theta}) = \left[ \left[ \nabla_{\boldsymbol{\theta}} J_j^f(\boldsymbol{\theta}) \right]_{j \in S_f}, \left[ \nabla_{\boldsymbol{\theta}} J_j^b(\boldsymbol{\theta}) \right]_{j \in S_b} \right],$$

Note that the assumptions outlined in this context align with the conventions in the literature related to convergence analysis (Tian et al., 2023; Qiu et al., 2021; Zhou et al., 2022; Xiong et al., 2022).

## B.1 Preliminaries

The following theorem proves that forward / backward expected rewards monotonically increase with each update iteration.

**Theorem B.3.** *Consider the forward / backward expected rewards, i.e., $\{J_j^f(\cdot)\}_{j \in S_f}$ and $\{J_j^b(\cdot)\}_{j \in S_b}$, and the forward / backward stochastic rewards, i.e., $\{\hat{J}_j^f(\cdot, \cdot)\}_{j \in S_f}$ and $\{\hat{J}_j^b(\cdot, \cdot)\}_{j \in S_b}$, complying with Assumptions 2 and 3, and $\boldsymbol{\beta}_{\mathrm{act}}$ as the solution of Equation (18). Moreover, consider the SGD in Equations (14) and (17) characterized by iteration number $i$ and actor learning rate $\{\mu_i\}_{i \in \mathcal{I}}$ with*

$$\mu_i \leq \min \left\{ \frac{1}{\max\{L_f, L_b\}}, \frac{1}{\max\{L_f, L_b\} \|\mathbf{B}\|} \mathbb{E} \left\{ \frac{1}{\mathbf{1}^{\top} \left( \nabla \boldsymbol{J}^{\mathrm{fb}}(\boldsymbol{\theta}^i)^{\top} \nabla \boldsymbol{J}^{\mathrm{fb}}(\boldsymbol{\theta}^i) \right)^{-1} \mathbf{1}} \right\} \right\},$$

*which generate the sequences $\{\boldsymbol{\phi}^i\}_{i \in \mathcal{I}}$, $\{\boldsymbol{\psi}^i\}_{i \in \mathcal{I}}$ and $\{\boldsymbol{\theta}^i\}_{i \in \mathcal{I}}$. We then get:*

$$\mathbb{E}\left\{ J_j^f(\boldsymbol{\theta}^{i+1}) \right\} \geq \mathbb{E}\left\{ J_j^f(\boldsymbol{\theta}^i) \right\}, \qquad j \in S_f$$

*and*

$$\mathbb{E}\left\{ J_k^b(\boldsymbol{\theta}^{i+1}) \right\} \geq \mathbb{E}\left\{ J_k^b(\boldsymbol{\theta}^i) \right\}, \qquad k \in S_b.$$

*Proof.* Based on Assumption 2, we obtain:

$$J_j^f(\boldsymbol{\theta}^{i+1}) - J_j^f(\boldsymbol{\theta}^i) \geq \nabla J_j^f(\boldsymbol{\theta}^i)^{\top} (\boldsymbol{\theta}^{i+1} - \boldsymbol{\theta}^i) - \frac{L_f}{2} \|\boldsymbol{\theta}^{i+1} - \boldsymbol{\theta}^i\|^2. \tag{36}$$

On the other hand, the update rule in Equation (17) gives:

$$\boldsymbol{\theta}^{i+1} = \boldsymbol{\theta}^i + \mu_i \left[ \nabla \hat{\boldsymbol{J}}^{\mathrm{f}}(\boldsymbol{\theta}^i, \boldsymbol{\phi}^i), \nabla \hat{\boldsymbol{J}}^{\mathrm{b}}(\boldsymbol{\theta}^i, \boldsymbol{\psi}^i) \right] \boldsymbol{\beta}_{\mathrm{act}}^i = \boldsymbol{\theta}^i + \mu_i \nabla \hat{\boldsymbol{J}}^{\mathrm{fb}}(\boldsymbol{\theta}^i, \boldsymbol{\phi}^i, \boldsymbol{\psi}^i) \boldsymbol{\beta}_{\mathrm{act}}^i. \tag{37}$$

Plugging Equation (37) into Equation (36) yields:

$$J_j^f(\boldsymbol{\theta}^{i+1}) - J_j^f(\boldsymbol{\theta}^i) \geq \mu_i \nabla J_j^f(\boldsymbol{\theta}^i)^{\top} \nabla \hat{\boldsymbol{J}}^{\mathrm{fb}}(\boldsymbol{\theta}^i, \boldsymbol{\phi}^i, \boldsymbol{\psi}^i) \boldsymbol{\beta}_{\mathrm{act}}^i - \frac{\mu_i^2 L_f}{2} \boldsymbol{\beta}_{\mathrm{act}}^{i}{}^{\top} \nabla \hat{\boldsymbol{J}}^{\mathrm{fb}}(\boldsymbol{\theta}^i, \boldsymbol{\phi}^i, \boldsymbol{\psi}^i)^{\top} \nabla \hat{\boldsymbol{J}}^{\mathrm{fb}}(\boldsymbol{\theta}^i, \boldsymbol{\phi}^i, \boldsymbol{\psi}^i) \boldsymbol{\beta}_{\mathrm{act}}^i.$$

Taking the expectation on both sides of the equation above results in:

$$\mathbb{E}\left\{ J_j^f(\boldsymbol{\theta}^{i+1}) - J_j^f(\boldsymbol{\theta}^i) \right\} \overset{(a)}{\geq} \mu_i \mathbb{E}\left\{ \nabla J_j^f(\boldsymbol{\theta}^i)^{\top} \nabla \hat{\boldsymbol{J}}^{\mathrm{fb}}(\boldsymbol{\theta}^i, \boldsymbol{\phi}^i, \boldsymbol{\psi}^i) \boldsymbol{\beta}_{\mathrm{act}}^i \right\} - \frac{\mu_i^2 L_f}{2} \boldsymbol{\beta}_{\mathrm{act}}^{i}{}^{\top} \left( \boldsymbol{B} + \nabla \boldsymbol{J}^{\mathrm{fb}}(\boldsymbol{\theta}^i)^{\top} \nabla \boldsymbol{J}^{\mathrm{fb}}(\boldsymbol{\theta}^i) \right) \boldsymbol{\beta}_{\mathrm{act}}^i$$

$$\overset{(b)}{=} \mu_i \mathbb{E}\left\{ \left( \boldsymbol{e}_j - \frac{\mu_i L_f}{2} \boldsymbol{\beta}_{\mathrm{act}}^i \right)^{\top} \nabla \boldsymbol{J}^{\mathrm{fb}}(\boldsymbol{\theta}^i)^{\top} \nabla \boldsymbol{J}^{\mathrm{fb}}(\boldsymbol{\theta}^i) \boldsymbol{\beta}_{\mathrm{act}}^i \right\} - \frac{\mu_i^2 L_f}{2} \boldsymbol{\beta}_{\mathrm{act}}^{i}{}^{\top} \boldsymbol{B} \boldsymbol{\beta}_{\mathrm{act}}^i$$

$$\overset{(c)}{\geq} \mu_i \mathbb{E}\left\{ \left( \boldsymbol{e}_j - \frac{\mu_i L_f}{2} \boldsymbol{\beta}_{\mathrm{act}}^i \right)^{\top} \nabla \boldsymbol{J}^{\mathrm{fb}}(\boldsymbol{\theta}^i)^{\top} \nabla \boldsymbol{J}^{\mathrm{fb}}(\boldsymbol{\theta}^i) \boldsymbol{\beta}_{\mathrm{act}}^i \right\} - \frac{\mu_i^2 L_f}{2} \|\boldsymbol{B}\|, \tag{38}$$

where: for (a), we used Assumption 3; for (b), we leveraged $\nabla J_j^f = \boldsymbol{e}_j^{\top} \nabla \boldsymbol{J}_j^{\mathrm{fb}}$ and the fact that

$$\mathbb{E}\left\{ \boldsymbol{\beta}_{\mathrm{act}}^{i}{}^{\top} \nabla \boldsymbol{J}^f(\boldsymbol{\theta}^i)^{\top} \nabla \hat{\boldsymbol{J}}^{\mathrm{fb}}(\boldsymbol{\theta}^i, \boldsymbol{\phi}^i, \boldsymbol{\psi}^i) \boldsymbol{\beta}_{\mathrm{act}}^i \mid \boldsymbol{\theta}^i \right\} = \boldsymbol{\beta}_{\mathrm{act}}^{i}{}^{\top} \nabla \boldsymbol{J}^f(\boldsymbol{\theta}^i)^{\top} \mathbb{E}\left\{ \nabla \hat{\boldsymbol{J}}^{\mathrm{fb}}(\boldsymbol{\theta}^i, \boldsymbol{\phi}^i, \boldsymbol{\psi}^i) \mid \boldsymbol{\theta}^i \right\} \boldsymbol{\beta}_{\mathrm{act}}^i$$

$$= \boldsymbol{\beta}_{\mathrm{act}}^{i}{}^{\top} \nabla \boldsymbol{J}^f(\boldsymbol{\theta}^i)^{\top} \nabla \boldsymbol{J}^{\mathrm{fb}}(\boldsymbol{\theta}^i) \boldsymbol{\beta}_{\mathrm{act}}^i$$

according to Corollary (B.1); for (c), we employed $\boldsymbol{\beta}_{\mathrm{act}}^{i}{}^{\top}\boldsymbol{B}\boldsymbol{\beta}_{\mathrm{act}}^{i} \leq \|\boldsymbol{B}\| \|\boldsymbol{\beta}_{\mathrm{act}}^{i}\|^{2} \leq \|\boldsymbol{B}\|$. On the other hand, we can derive from Equation (18) that for all $\beta_{\mathrm{act,j}}^{i} \geq 0$:

$$\boldsymbol{\beta}_{\mathrm{act}}^{i} = \left(\mathbf{1}^{\top}\left(\nabla\boldsymbol{J}^{\mathrm{fb}}(\boldsymbol{\theta}^{i})^{\top}\nabla\boldsymbol{J}^{\mathrm{fb}}(\boldsymbol{\theta}^{i})\right)^{-1}\mathbf{1}\right)^{-1}\left(\nabla\boldsymbol{J}^{\mathrm{fb}}(\boldsymbol{\theta}^{i})^{\top}\nabla\boldsymbol{J}^{\mathrm{fb}}(\boldsymbol{\theta}^{i})\right)^{-1}\mathbf{1}. \tag{39}$$

By substituting this into Equation (38), we finally get:

$$\mathbb{E}\left\{J_{j}^{f}(\boldsymbol{\theta}^{i+1}) - J_{j}^{f}(\boldsymbol{\theta}^{i})\right\} \geq \mu_{i}\left(1 - \frac{\mu_{i}L_{f}}{2}\right)\mathbb{E}\left\{\frac{1}{\mathbf{1}^{\top}\left(\nabla\boldsymbol{J}^{\mathrm{fb}}(\boldsymbol{\theta}^{i})^{\top}\nabla\boldsymbol{J}^{\mathrm{fb}}(\boldsymbol{\theta}^{i})\right)^{-1}\mathbf{1}}\right\} - \frac{\mu_{i}^{2}L_{f}}{2}\|\boldsymbol{B}\|$$

$$\overset{a}{\geq} \frac{\mu_{i}}{2}\mathbb{E}\left\{\frac{1}{\mathbf{1}^{\top}\left(\nabla\boldsymbol{J}^{\mathrm{fb}}(\boldsymbol{\theta}^{i})^{\top}\nabla\boldsymbol{J}^{\mathrm{fb}}(\boldsymbol{\theta}^{i})\right)^{-1}\mathbf{1}}\right\} - \frac{\mu_{i}^{2}L_{f}}{2}\|\boldsymbol{B}\| \geq 0,$$

where we used $\mu_{i}\max\{L_{f}, L_{b}\} \leq 1$ for (a). The statement follows by considering that the denominator in the right-hand side of the equation above is positive due to the positive-definiteness of $\left(\nabla\boldsymbol{J}^{\mathrm{fb}}(\boldsymbol{\theta}^{i})^{\top}\nabla\boldsymbol{J}^{\mathrm{fb}}(\boldsymbol{\theta}^{i})\right)^{-1}$. The same analysis can be applied to infer $\mathbb{E}\left\{J_{j}^{b}(\boldsymbol{\theta}^{i+1}) - J_{j}^{b}(\boldsymbol{\theta}^{i})\right\} \geq 0$ □

**Remark B.4.** Theorem B.3 guarantees all the forward and backward expected rewards $\left\{\mathbb{E}\,J_{j}^{f}(\boldsymbol{\theta})\right\}_{j\in S_{f}}$ and $\left\{\mathbb{E}\,J_{j}^{b}(\boldsymbol{\theta})\right\}_{j\in S_{b}}$ constantly increase with the number of algorithm iterations. Consequently, this enables us to jointly improve all of the cumulative rewards, either forward or backward, on average with each iteration.

**Corollary B.5.** *Consider the framework of Lemma B.3, we then get:*

$$\mathbb{E}\left\{\boldsymbol{\beta}_{\mathrm{act}}^{i}{}^{\top}\nabla\boldsymbol{J}^{\mathrm{fb}}(\boldsymbol{\theta}^{i})^{\top}\nabla\boldsymbol{J}^{\mathrm{fb}}(\boldsymbol{\theta}^{i})\boldsymbol{\beta}_{\mathrm{act}}^{i}\right\} \leq \frac{2}{\mu_{i}}\mathbb{E}\left\{\sum_{j\in S_{f}\cup S_{b}}\beta_{\mathrm{act,j}}^{i}\left(J_{j}^{\mathrm{fb}}(\boldsymbol{\theta}^{i+1}) - J_{j}^{\mathrm{fb}}(\boldsymbol{\theta}^{i})\right)\right\} + \mu_{i}\max\{L_{f}, L_{b}\}\|\boldsymbol{B}\|.$$

*Proof.* The statement follows based on Equation (38) and $\mu_{i}\max\{L_{f}, L_{b}\} \leq 1$. □

## B.2 Analysis for Lipschitz-smooth Rewards

We perform a convergence analysis by focusing on the Lipschitz-smoothness condition detailed in Assumption 2. However, we first need to present a definition for the convergence to locally Pareto-optimal solutions.

**Definition B.6.** The parameter sequence $\{\boldsymbol{\theta}_{i}\}_{i=1}^{I}$ is said to converge to locally Pareto-optimal solutions (Zhou et al., 2022) if

$$\lim_{i\to\infty}\mathbb{E}\left\{\min_{\substack{\beta_{j}\,\geq\,0\\\sum_{j}\beta_{j}=1}}\left\|\sum_{j\in|S_{f}\cup S_{b}|}\nabla_{\boldsymbol{\theta}}J_{j}^{\mathrm{fb}}(\boldsymbol{\theta}^{i})\beta_{j}\right\|^{2}\right\} \to 0,$$

where $\nabla J_{j}^{\mathrm{fb}}(\boldsymbol{\theta})$ is the $j$-th element of $\nabla\boldsymbol{J}^{\mathrm{fb}}(\boldsymbol{\theta})$ with $\nabla\boldsymbol{J}^{\mathrm{fb}}(\boldsymbol{\theta}) = \left[\left[\nabla_{\boldsymbol{\theta}}J_{j}^{f}(\boldsymbol{\theta})\right]_{j\in S_{f}}, \left[\nabla_{\boldsymbol{\theta}}J_{j}^{b}(\boldsymbol{\theta})\right]_{j\in S_{b}}\right]$.

We thus have the following theorem.

**Theorem B.7.** *Consider the forward/backward state-value estimations, i.e., $\{V_{j,\phi}^{f}(\cdot)\}_{j\in S_{f}}$ and $\{V_{j,\psi}^{b}(\cdot)\}_{j\in S_{b}}$, following Assumption 1. Moreover, assume that the forward/backward expected rewards, i.e., $\{J_{j}^{f}(\cdot)\}_{j\in S_{f}}$ and $\{J_{j}^{b}(\cdot)\}_{j\in S_{b}}$, and the forward/backward stochastic rewards, i.e., $\{\hat{J}_{j}^{f}(\cdot,\cdot)\}_{j\in S_{f}}$ and $\{\hat{J}_{j}^{b}(\cdot,\cdot)\}_{j\in S_{b}}$, comply with Assumptions 2 and 3, and that $\boldsymbol{\beta}_{\mathrm{act}}$ is the solution of Equation (18). Finally, consider the SGD in Equations (14) and (17) characterized by iteration number $i$ and actor learning rate $\{\mu_{i}\}_{i\in\mathcal{I}}$ with*

$$\mu_{i} \leq \min\left\{\frac{1}{\max\{L_{f}, L_{b}\}}, \frac{1}{\max\{L_{f}, L_{b}\}\|\boldsymbol{B}\|}\left(\mathbf{1}^{\top}\left(\nabla\boldsymbol{J}^{\mathrm{fb}}(\boldsymbol{\theta}^{i})^{\top}\nabla\boldsymbol{J}^{\mathrm{fb}}(\boldsymbol{\theta}^{i})\right)^{-1}\mathbf{1}\right)^{-1}\right\},$$

*and* $0 < \mu_I \le \ldots \le \mu_i \le \ldots \le \mu_1$, *which generate the sequences* $\{\phi^i\}_{i=1}^I$, $\{\psi^i\}_{i=1}^I$ *and* $\{\theta^i\}_{i\in\mathcal{I}}$. *Then, we get:*

$$
\frac{1}{I} \sum_{i=1}^I \mathbb{E}\Big\{ \|\nabla \boldsymbol{J}^{\text{fb}}(\boldsymbol{\theta}^i)\boldsymbol{\beta}_{\text{act}}^i\|^2 \Big\} \le \frac{\max\{L_f, L_b\}\|\boldsymbol{B}\|}{I} \sum_{i=1}^I \frac{\mu_i}{2 - \mu_i \max\{L_f, L_b\}}
$$
$$
+ \frac{2}{I\,\mu_I\,|S_f \cup S_b|} \sum_{j\in S_f \cup S_b} \mathbb{E}\big\{ J_j^{\text{fb}}(\boldsymbol{\theta}^I) - J_j^{\text{fb}}(\boldsymbol{\theta}^1) \big\}. \tag{40}
$$

*Proof.* Based on Equation (38) an its counterpart for $\{J_j^b(\boldsymbol{\theta})\}_{j\in S_b}$, we have:

$$
\mathbb{E}\bigg\{ \sum_{j\in S_f \cup S_b} J_j^f(\boldsymbol{\theta}^{i+1}) - J_j^f(\boldsymbol{\theta}^i) \bigg\} \ge \mu_i \mathbb{E}\bigg\{ \Big(\mathbf{1} - \frac{\mu_i L^{\max}}{2}\boldsymbol{\beta}_{\text{act}}^i|S_f \cup S_b|\Big)^\top \nabla \boldsymbol{J}^{\text{fb}}(\boldsymbol{\theta}^i)^\top \nabla \boldsymbol{J}^{\text{fb}}(\boldsymbol{\theta}^i)\boldsymbol{\beta}_{\text{act}}^i \bigg\} - \frac{\mu_i^2 L^{\max}}{2}\|\boldsymbol{B}\||S_f \cup S_b|
$$
$$
\overset{(a)}{\ge} |S_f \cup S_b|\mu_i\bigg( \mathbb{E}\Big\{ \Big(1 - \frac{\mu_i L^{\max}}{2}\Big)\boldsymbol{\beta}_{\text{act}}^i{}^\top \nabla \boldsymbol{J}^{\text{fb}}(\boldsymbol{\theta}^i)^\top \nabla \boldsymbol{J}^{\text{fb}}(\boldsymbol{\theta}^i)\boldsymbol{\beta}_{\text{act}}^i \Big\} - \frac{\mu_i L^{\max}}{2}\|\boldsymbol{B}\| \bigg)
$$

where $L^{\max} := \max\{L_f, L_b\}$, and we used the identity $\mathbf{1}^\top \nabla \boldsymbol{J}^{\text{fb}}{}^\top \nabla \boldsymbol{J}^{\text{fb}}\boldsymbol{\beta}_{\text{act}} = |S_f \cup S_b|\boldsymbol{\beta}_{\text{act}}{}^\top \nabla \boldsymbol{J}^{\text{fb}}{}^\top \nabla \boldsymbol{J}^{\text{fb}}\boldsymbol{\beta}_{\text{act}}$ based on Equation (39) for (a). Since $\mu_i L^{\max} \le 1$, it is:

$$
\mathbb{E}\Big\{ \|\nabla \boldsymbol{J}^{\text{fb}}(\boldsymbol{\theta}^i)\boldsymbol{\beta}_{\text{act}}^i\|^2 \Big\} \le \frac{1}{|S_f \cup S_b|\mu_i(1 - \frac{\mu_i}{2}L^{\max})} \mathbb{E}\bigg\{ \sum_{j\in S_f \cup S_b} \big( J_j^{\text{fb}}(\boldsymbol{\theta}^{i+1}) - J_j^{\text{fb}}(\boldsymbol{\theta}^i) \big) \bigg\} + \frac{\mu_i L^{\max}}{2 - \mu_i L^{\max}}\|\boldsymbol{B}\|
$$
$$
\overset{(a)}{\le} \frac{2}{\mu_I|S_f \cup S_b|} \mathbb{E}\bigg\{ \sum_{j\in S_f \cup S_b} \big( J_j^{\text{fb}}(\boldsymbol{\theta}^{i+1}) - J_j^{\text{fb}}(\boldsymbol{\theta}^i) \big) \bigg\} + \frac{\mu_i L^{\max}}{2 - \mu_i L^{\max}}\|\boldsymbol{B}\|, \tag{41}
$$

where (a) follows from $\mu_i L^{\max} \le 1$. We then take the summation on both side of Equation (41), and apply telescopic cancellation. The statement follows by considering that $\boldsymbol{\beta}_{\text{act}}$ is the solution of Equation (18) for $\boldsymbol{\theta} = \boldsymbol{\theta}^i$. $\qquad\square$

**Remark B.8.** Under the learning-rate scheduling $\mu_i = \mathcal{O}(1/\sqrt{i})$, both terms $\frac{2}{I\mu_I}$ and $\frac{1}{I}\sum_{i\in I} \mu_i/(2 - \mu_i \max\{L_f, L_b\})$ in Equation (40) decay at the rate of $\mathcal{O}(1/\sqrt{I})$, where $I$ is the number of algorithm iterations – equivalently, the number of policy updates in FB-MOAC. Consequently, Theorem B.7 guarantees convergence to a locally Pareto-optimal solution (Zhou et al., 2022) with a convergence rate of $\mathcal{O}(1/\sqrt{I})$. This rate is notably consistent with that of single-objective actor-critic methods for forward-MDPs, which exhibit a convergence rate of $\mathcal{O}(1/\sqrt{I})$ (Fu et al., 2021).

**Remark B.9.** In contrast, if the learning rate is chosen as $\mu_i = \mathcal{O}(1/i)$, the term $\frac{2}{I\mu_I}$ in Equation (40) does not vanish as iteration becomes large. This prevents a general convergence guarantee in this setting.

## C   Additional Details on the Edge Caching Use Case

The environment of this experiment is a cellular network with cache-equipped Base-Stations (BSs) similar to that in (Amidzadeh et al., 2023). The BSs are spatially distributed across the network with intensities $\lambda_{\text{bs}}$. The environment also includes a library of $N$ different content items as well as fixed mobile users requesting them from the cellular network. The network operates over time slots with index $t \in \{1, \ldots, T\}$, where $T$ is the total duration of the operation. The network handles user requests at the beginning of each time slot. Content items have different *popularity* $\{p_n^{\text{pop}}(t)\}_{n=1}^N$, where $p_n^{\text{pop}}(t)$ is the probability that content $n$ is requested by a randomly selected user at time $t$. The goal is to satisfy as many users as possible during the network operation. At the beginning of each time-slot, the BSs cache the most popular content items with probability $\{p_n^{\text{cach}}(t)\}_{n=1}^N$ and simultaneously *multicast* them toward users by consuming content-specific radio resources $\{w_n(t)\}_{n=1}^N$. We denote the system action parameters by the vector $\mathbf{a}(t)$, which depends on the content-specific bandwidth allocation and cache placement of BSs, i.e., $\mathbf{a}(t) = [\{p_n^{\text{cach}}(t)\}_{n=1}^N, \{w_n(t)\}_{n=1}^N]$. A multicast outage may occur with probability $\{O_n(\mathbf{a}(t), t)\}_{n=1}^N$:

$$
O_n(\mathbf{a}(t), t) = \text{erfc}\left( \frac{\pi^2 \lambda_{bs} p_n^{\text{cach}}(t)}{4\sqrt{\eta_n(t)}} \right), \quad \eta_n(t) = 2^{1/w_n(t)} - 1, \tag{42}
$$

which is obtained by averaging over users. As a result, certain users fail to receive the requested content in the current timeslot and their request is deferred to the subsequent one. Hence, each time-slot sees a distribution of users accounting for the repeated requests and a distribution describing the new preferences toward content items. This leads to a time-varying model for the request probability of content $n$, $p_n^{\text{req}}(t)$:

$$p_n^{\text{req}}(t) = \underbrace{p_n^{\text{req}}(t-1)O_n\big(\mathbf{a}(t-1),t-1\big)}_{\text{repeated request}} + \underbrace{p_n^{\text{pop}}(t)\sum_{m=1}^{N}\big(1-O_m\big(\mathbf{a}(t-1),t-1\big)\big)p_m^{\text{req}}(t-1)}_{\text{new request based on the popularity}}. \tag{43}$$

Note that $p_n^{\text{req}}(t)$ indicates the request probability of content $n$ averaged over all users. Then, it can be simply verified that $\sum_{n=1}^{N} p_n^{\text{req}}(t) = 1$, considering $\sum_{n=1}^{N} p_n^{\text{pop}}(t) = 1$. Equation (43) therefore represents a **forward dynamics**, with the forward state vector $\mathbf{s}(t) = \mathbf{p}^{\text{req}}(t)$ and the action vector $\mathbf{a}(t)$.

A request for a content item is repeated across several time-slots until successfully fulfilled, resulting in an expected latency $L_n(t)$ for the successful delivery of content $n$. Its time-varying dynamics can be derived by the law of total expectation as follows:

$$L_n(\mathbf{a}(t),t) = \mathbb{E}\{\text{latency} \mid \text{outage}\}\mathbb{P}\{\text{outage}\} + \mathbb{E}\{\text{latency}\mid \text{no outage}\}\mathbb{P}\{\text{no outage}\}$$

$$= \Big(d(t) + L_n(t+1)\Big)O_n(\mathbf{a}(t),t) + \frac{d(t)}{2}\big(1-O_n(\mathbf{a}(t),t)\big), \tag{44}$$

where $d(t)$ is the duration of time-slot $t$ in seconds, and $L_n(\mathbf{a}(T),T) = 0$ since system operations finish and the users do not need to wait any longer. Equation (44) represents a **backward dynamics**, with the backward state vector $\mathbf{y}(t) = \mathbf{L}(t)$ and the action vector $\mathbf{a}(t)$. Note that this model fully captures the trade-offs involved in the delay dynamics and differs from the conventional formalism that does not provide a comprehensive model when accounting for successive slots; for the delivery without outage, the expected latency simply becomes $L_n(t) = \frac{d(t)}{2}$, as its realizations follow a uniform distribution with values between 0 and $d(t)$. Notice that the backward dynamics in Equation (44) based on Theorem 3.6 cannot be expressed as a standard MDP. Thus, Equations (43) and (44) together model a FB-MDP and are coupled through the action $\mathbf{a}(t)$. Hence, they should be jointly considered to obtain an optimal cache policy.

# D  Case Study: Computation Offloading

We now present an additional use case in the context of computation offloading (Zabihi et al., 2023).

## D.1  System Model

There are $N_{\text{dev}}$ mobile devices and $N$ computational intensive tasks with diverse sizes $\{s_n\}_{n=1}^{N}$. A typical mobile device prefers task $n$ with probability $p_n^{\text{prf}}(t)$, and it *offloads* the preferred tasks with probability $p_n^{\text{off}|\text{prf}}(t)$ to an edge server. The server operates in time slots with duration $\tau$ indexed by $t$ and leverages a task-specific parallelism mechanism to process the offloaded tasks. Specifically, it employs $N$ buffers and $N$ computational resources with $B_n^{\text{edg}}(t)$ signifying the buffer capacity and $C_n^{\text{edg}}(t) \geq C_{\min}^{\text{edg}}$ denoting the computational resources allocated to file $n$. $C_{\min}^{\text{edg}}$ is the minimum extent of allocated resource, while $\sum_{n=1}^{N} B_n^{\text{edg}}(t) = B^{\text{edg}}$ and $\sum_{n=1}^{N} C_n^{\text{edg}}(t) = C^{\text{edg}}$ represent the total buffer limit $B^{\text{edg}}$ and computational capacity $C^{\text{edg}}$, respectively. The control parameters for this problem are thus $\{(p_n^{\text{off}|\text{prf}}(\cdot), B_n^{\text{edg}}(\cdot), C_n^{\text{edg}}(\cdot))\}_{n=1}^{N}$. A typical device offloading task $n$ encounters a failure if the corresponding buffer overflows, resulting in the need to re-offload the task. The queue length of the $n$-th buffer $S_n$ is thus described by the following expression:

$$L_n(t+1) = \max\Big\{L_n(t) + \underbrace{S_n p_n^{\text{off}}(t+1)\alpha_n(t+1)}_{\text{new data buffered}} - \underbrace{C_n^{\text{edg}}(t+1)}_{\text{computed data de-buffered}}, 0\Big\}, \tag{45}$$

for $n \in \{1, \ldots, N\}$, where $S_n = s_n N_{\text{dev}}$, $p_n^{\text{off}}(t)$ is the offloading probability for file $n$, $S_n p_n^{\text{off}}(t)$ is the total amount of data offloaded for the $n$-th buffer, and

$$\alpha_n(t+1) = \min \left\{ \frac{B_n^{\text{edg}}(t+1) - L_n(t)}{S_n p_n^{\text{off}}(t+1)}, 1 \right\},$$

denotes the fraction of data that can be buffered due to the buffer capacity $B_n^{\text{edg}}$. Therefore, it can be easily verified that $L_n(t+1) \leq B_n^{\text{edg}}(t+1)$. The overflow probability $\mathcal{O}_n$ for $n$-th buffer is thus obtained as $\mathcal{O}_n(t) = 1 - \alpha_n(t)$. This equation exhibits a controlled forward dynamics, based on which we constitute the forward state $[\mathbf{p}^{\text{prf}}(t), \mathbf{L}(t)]$ for this dynamics.

We now compute the *average computation time* needed for a typical device preferring task $n$, i.e., $t_n^{\text{prf}}(t)$. If the task is preferred and locally computed, $t_n^{\text{prf}}(t)$ depends on the computation capacity of the device itself; if it is offloaded, $t_n^{\text{prf}}(t)$ depends on the computation offloading time $t_n^{\text{off}}(t)$, i.e., the average computation time needed for a device to offload task $n$. Applying the law of total expectation yields:

$$t_n^{\text{prf}}(t) = \mathbb{E}\{\text{computation time} \mid \text{task preferred}\} = p_n^{\text{off}|\text{prf}}(t)\, t_n^{\text{off}}(t) + (1 - p_n^{\text{off}|\text{prf}}(t)) \frac{s_n}{C^{\text{dev}}}. \tag{46}$$

As a consequence, we need to compute $t_n^{\text{off}}(t)$. If the task being offloaded faces an overflow, it will be re-offloaded in the next time-slot. However, the computation time depends on the queue length and the computation resource allocated to the task if no overflow takes place. Therefore, $t_n^{\text{off}}(t)$ is derived by the total expectation law as follows:

$$t_n^{\text{off}}(t) = \mathbb{E}\{\text{computation time} \mid \text{task offloaded}\} = \mathcal{O}_n(t+1)(\tau + t_n^{\text{prf}}(t+1)) + (1 - \mathcal{O}_n(t+1))\, t_n, \tag{47}$$

for $n \in \{1, \ldots, N\}$, where $t_n$ stands for the needed time to compute task $n$ with size $s_n$ if it is successfully buffered. We thus have:

$$t_n = \frac{L_n(t) + \frac{1}{2}\alpha_n(t+1)p_n^{\text{off}}(t+1)S_n + s_n}{C_n^{\text{edg}}(t+1)}\tau,$$

obtained considering that $n$-th buffer has already a queue with length $L_n(t)$ and an additional buffer $S_n$ with probability $\frac{1}{2}\alpha_n(t+1)p_n^{\text{off}}(t+1)$. Equations (46) and (47) together provide a continuous model for the average time required to successfully compute task $n$ within different slots. Additionally, they represent a controlled backward dynamics with the backward state $\mathbf{t}^{\text{prf}}(t) = [t_1^{\text{prf}}, \ldots, t_n^{\text{prf}}](t)$. We now consider two action-coupled conflicting rewards. The forward reward is related to the overall overflow probability as:

$$r_{\text{OP}}(t) = -\sum_{n=1}^{N} p_n^{\text{prf}}(t)\mathcal{O}_n(t),$$

while the backward reward is related to the expected computation time:

$$r_{\text{CT}}(t) = -\sum_{n=1}^{N} p_n^{\text{prf}}(t)t_n^{\text{prf}}(t),$$

This problem thus represents a FB-MDP with an action $\left[\{C_n^{\text{edg}}\}_n, \{B_n^{\text{edg}}\}_n, \{p_n^{\text{off}|\text{prf}}\}_n\right]$ and an action space $[0,1]^N \times [0,1]^N \times [0,1]^N$.

## D.2 Experiment Setup and Hyper-parameters

We set the number of devices $N_{\text{dev}} = 100$, the number of tasks $N = 20$, the file size $s_n = 10 + n$ Kbits, the computational capacity of the devices $C^{\text{dev}} = 10$ Kbits / slot, the minimum extent of allocated resource $C_{\min}^{\text{edg}} = 10^{-5}$, the edge computational capacity $C^{\text{edg}} = 100$ Kbits/slot, the edge buffer capacity $B = 100$ Kbits and the slot duration $\tau = 60$ seconds. The hyper-parameters of FB-MOAC are the same as in the previous experiment excluding the learning rates of actors and critics, which are set to $3 \times 10^{-3}$.

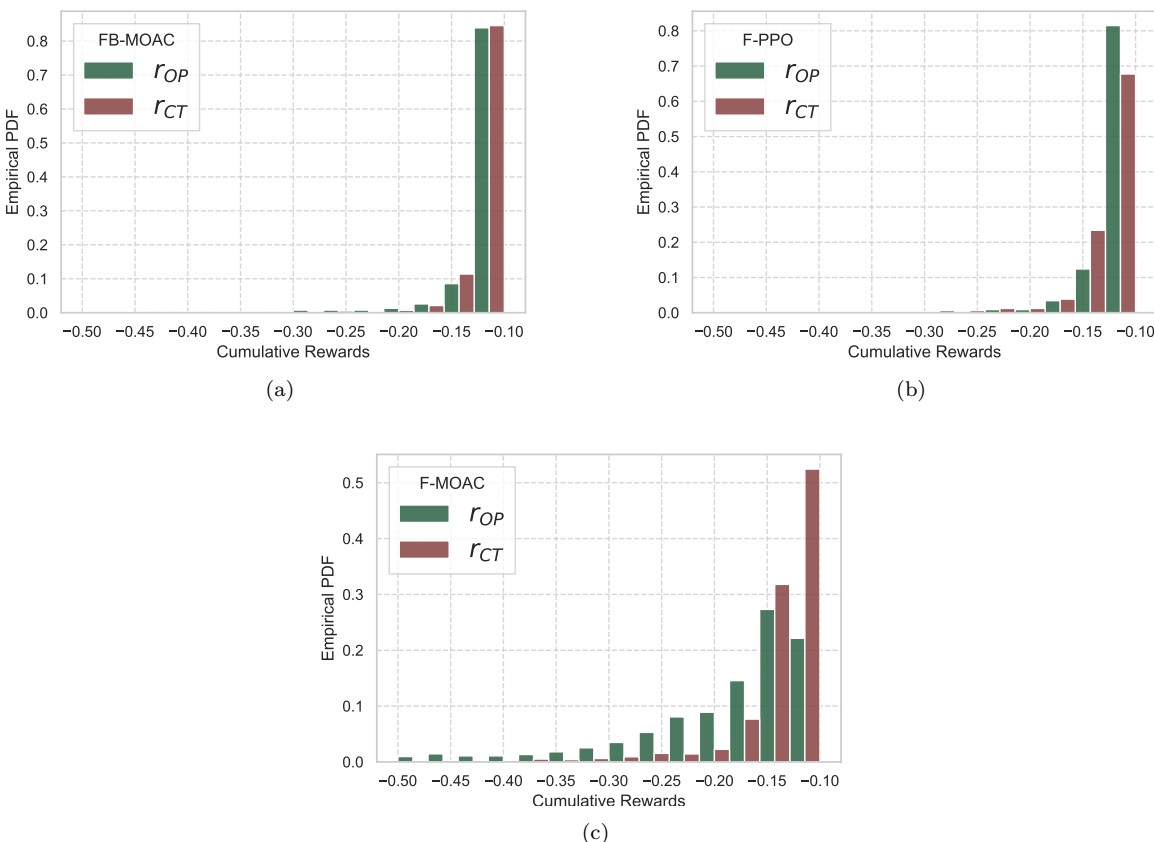

Figure 6: Histograms depicting the empirical distribution function of the computation time $r_{\mathrm{CT}}$ and the overflow probability $r_{\mathrm{OP}}$ for: (a) FB-MOAC, (b) F-PPO, and (c) F-MOAC.

## D.3    Performance Evaluation

We consider a learning-based strategy to evaluate FB-MOAC in this experiment. Accordingly, we leverage the fact that optimizing the overflow probability decreases the expected latency according to Equation (47). Consequently, we apply the baseline RL algorithms PPO (Schulman et al., 2017a) and A2C (Grondman et al., 2012) to obtain an offloading policy. Similar to Section 5.2.3, we call the resulting algorithms F-PPO and F-MOAC, as they manage the backward reward using a forward mechanism.

Figure 6 reports the histograms (i.e., the empirical probability density function) for the performance of FB-MOAC, F-PPO and F-MOAC algorithms (respectively) in terms of $r_{\mathrm{CT}}$ and $r_{OP}$. Clearly, both the forward and backward rewards of FB-MOAC are higher than those of F-PPO and F-MOAC. Consequently, FB-MOAC outperforms F-PPO and F-MOAC from the perspectives of both expected computation time $r_{\mathrm{CT}}$ and overflow probability $r_{\mathrm{OP}}$. In other words, the resulting policy of FB-MOAC Pareto-dominates the strategies of F-MOAC and F-PPO on average. As expected, FB-MOAC shows a more significant improvement in the computation time compared to the other two algorithms. Nonetheless, FB-MOAC even gives a better overflow probability performance than that of F-MOAC and F-PPO as it obtains a more favorable learning mechanism even for the forward dynamics. These results indicate the importance of explicitly incorporating the backward MDP rather than eliminating it through adjustments to the backward rewards. They also demonstrate the effectiveness of the proposed FB-MOAC algorithm in addressing the corresponding FB-MDP problem.

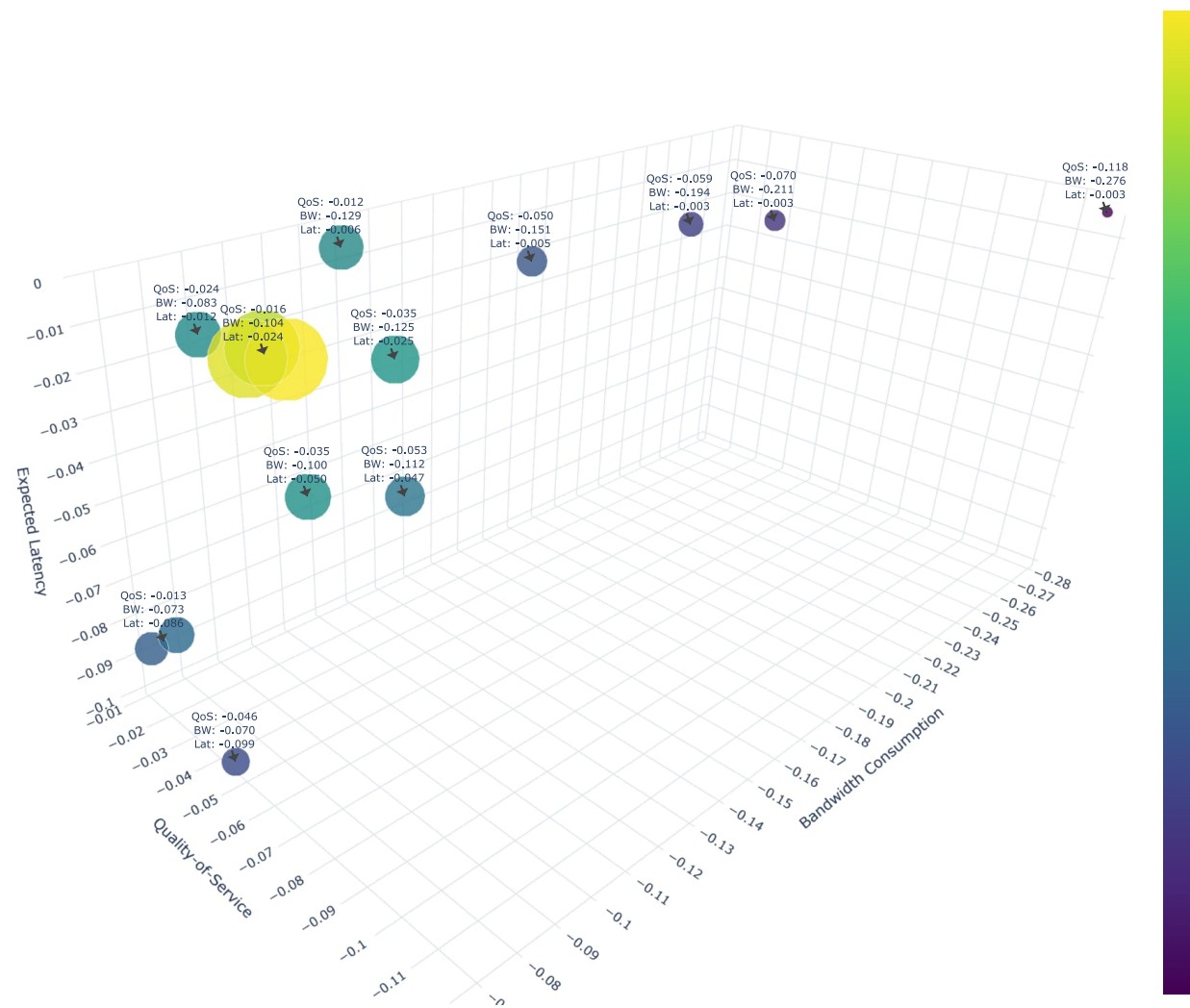

Figure 7: The collection of Pareto-optimal solutions for the use case of edge caching, obtained by applying a discrete policy over different preference settings $\epsilon_i^f \in [0.1, 1]$ and $\epsilon_i^b \in [0.1, 1]$. The radius of the points denotes the probability of occurrence. Given that many of these solutions cannot be dominated by others, the FB-MOAC algorithm can efficiently provide the Pareto-front solutions.

## E Deriving Pareto-optimal Solutions

Here, we employ the mechanism explained in Section 4.3 to obtain the Pareto-optimal solutions of the use case considered in Section 5.2, namely, edge caching in wireless networks. Figure 7 illustrates the collection of Pareto-optimal solutions with diverse preference parameters $\epsilon_i^f \in [0.1, 1]$ and $\epsilon_i^b \in [0.1, 1]$. Note that most of the solutions cannot be dominated by others, therefore, FB-MOAC can provide most of the Pareto-optimal solutions with the proposed preference policy.

## F Transforming a FB-SDE into a FB-MDP

This section shows the wider applicability of FB-MDPs and FB-MOAC to problems described by FB-SDEs. Specifically, it explains a general method to transform a FB-SDE into a FB-MDP, which can then be solved with FB-MOAC (as described in Section 5.1). In particular, the Euler-Maruyama method (Kloeden & Platen,

1992) can be used to discretize the space of control problems. More specifically, the forward and backward SDEs can be transformed to forward and backward MDPs, respectively.

Consider the following controlled FB-SDE:

$$
\begin{cases}
d\mathbf{x}(t) = \boldsymbol{f}(\mathbf{x}(t), \mathbf{u}(t), t)\, dt + \sum_{i=1}^{l} \boldsymbol{\sigma}^i(\mathbf{x}(t), \mathbf{u}(t), t)\, dw^i(t), \\
d\mathbf{y}(t) = \boldsymbol{g}(\mathbf{y}(t), \mathbf{u}(t), \{\mathbf{z}^i(t)\}_i, t)\, dt + \sum_{i=1}^{l} \mathbf{z}^i(t)\, dw^i(t), \\
\mathbf{x}(0) = x_0, \quad \mathbf{y}(T) = y_T,
\end{cases}
\tag{48}
$$

where $\mathbf{w}(t) = \{w^i(t)\}_{i=1}^{l}$ is an $l$-dimensional Wiener process, $\mathbf{x}(t) \in \mathbb{R}^n$ is the forward process, $\boldsymbol{f}(\cdot, \cdot, \cdot) : \mathbb{R}^n \times U \times [0, T] \to \mathbb{R}^n$ is the drift function which describes the dynamics of the forward SDE and is governed by an optimization control process $\mathbf{u}(t) \in U \subset \mathbb{R}^k$, and $\boldsymbol{\sigma}^i(\cdot, \cdot, \cdot) : \mathbb{R}^n \times U \times [0, T] \to \mathbb{R}^n$ is the diffusion coefficient determining the extent of noise added to the forward dynamics. Conversely, $\mathbf{y}(t) \in \mathbb{R}^m$ is the backward process, $\boldsymbol{g}(\cdot, \cdot, \cdot, \cdot) : \mathbb{R}^m \times U \times \mathbb{R}^{m \times n} \times [0, T] \to \mathbb{R}^m$ is the generator function and $\mathbf{z}^i(t) \in \mathbb{R}^m$ is the dynamics of the control process.

Note that the solution of the backward SDE is determined by the pair $(\mathbf{y}(\cdot), \{\mathbf{z}^i(\cdot)\}_i)$ where $\{\mathbf{z}^i(\cdot)\}_i$ should be found to guarantee that the backward process $\mathbf{y}(t)$ is properly adapted. The solution of the FB-SDE in Equation (48) is thus denoted by the tuple $(\mathbf{x}(\cdot), \mathbf{y}(\cdot), \{\mathbf{z}^i(\cdot)\}_i)$.

In this section, we resort to Euler-type numerical approaches that discretize the evolution of dynamics to numerically solve the considered FB-SDE. In this regard, we partition the time interval $[0, T]$ into $N$ sub-intervals $[t_{k-1}, t_k)$ for $k \in \{1, \dots, N\}$, each sub-interval with length $\Delta t = \frac{T}{N}$, where $t_0 = 0$ and $t_N = T$. By applying the Euler–Maruyama method (Kloeden & Platen, 1992), we then obtain:

$$
\mathbf{x}(t_k + \Delta t) \approx \mathbf{x}(t_k) + \boldsymbol{f}(\mathbf{x}(t_k), \mathbf{u}(t_k), t_k)\Delta t + \sum_{i=1}^{l} \boldsymbol{\sigma}^i(\mathbf{x}(t_k), \mathbf{u}(t_k), t_k)\Delta w^i(t_k), \quad \text{for } k \in \{0, \dots, N-1\} \tag{49}
$$

where $\Delta w^i(t_k) = w^i(t_k + \Delta t) - w^i(t_k)$ is a Gaussian random variable with variance $\Delta t$. Equation (49) can be interpreted as a forward MDP with action $\mathbf{u}(\cdot)$ and forward state $\mathbf{x}(\cdot)$.

For the backward SDE, we employ a *semi-stochastic approach* (Archibald et al., 2020) as it is considerably more efficient from the complexity perspective, even though it might require more iterations for the algorithm to converge to an accurate solution. This can be compared with the stochastic gradient descent that alleviates extreme complexity by estimating the expectation with a single sample. Accordingly, we get:

$$
\mathbf{y}(t_k) \approx \mathbf{y}(t_k + \Delta t) - \mathbf{g}\Big(\mathbf{y}(t_k + \Delta t), \mathbf{u}(t_k + \Delta t), \mathbf{Z}(t_k + \Delta t), t_k + \Delta t\Big)\Delta t. \tag{50}
$$

where $\mathbf{Z}(\cdot) = \{\mathbf{z}^i(\cdot)\}_{i=1}^{l}$, and $\mathbf{z}^i(t)$ is obtained as:

$$
\mathbf{z}^i(t_k) \approx \frac{1}{\Delta t}\mathbf{y}(t_k + \Delta t)\, \Delta w^i(t), \qquad \text{for } i \in \{1, \dots, l\}. \tag{51}
$$

Equation (50), together with Equation (51), represents a backward MDP with action $(\mathbf{u}(\cdot), \mathbf{Z}(\cdot))$ and backward state $\mathbf{y}(\cdot)$.

