# OpenReview forum: "FB-MOAC: A Reinforcement Learning Algorithm for Forward-Backward Markov Decision Processes"
_TMLR — Accepted by TMLR_

### Review · Reviewer_A32U · 2025-03-28

**Summary Of Contributions:**

This work introduces forward-backward Markov decision processes (FB-MDPs), a new class of problems that extend beyond the expressiveness of standard MDPs. Building on a derived optimality condition specific to FB-MDPs, the authors developed a multi-objective reinforcement learning algorithm, FB-MOAC. They provided theoretical guarantees on the algorithm’s optimality and convergence, and validated its effectiveness through comprehensive experiments in two distinct application domains.

**Audience:**

Yes

**Claims And Evidence:**

Yes

**Requested Changes:**

- Could you elaborate on how restrctive this assumption can be: This work constrains the definition of FB-MDPs to the case where the forward (backward) dynamics does not depend on the backward (forward) state.

- Where is Assumption 4 mentioned in Theorem B.8? Remarks B.9 and B.10 are very inform.

- Please provide a formal and complete version of Theorem 4.1 with all necessary assumptions. Do you assume the optimization problem at each iteration is well solved?

**Strengths And Weaknesses:**

### Strengths:
This paper introduces a novel approach to solving Forward-Backward MDPs using multi-objective reinforcement learning. The authors support their method with both theoretical analysis and empirical evaluations across multiple benchmark tasks.

### Weaknesses:
- The positioning of this paper within the field of Forward-Backward MDPs is not clearly articulated. Prior work, including key statements and theoretical results, should be explicitly cited and clearly distinguished. Additionally, the paper's novel contributions—both in terms of theoretical insights and algorithmic design—should be more prominently emphasized.
- The theoretical results rely on strong assumptions, such as strong convexity and Lipschitz smoothness. However, most assumptions have been put into the appendix and it's not clear how the assumptions would hold when using NN-based functions.
- The learning-based baselines used for the Edge Caching benchmark are more like ablations of the proposed algorithm, while the rule-based one is relatively too simple. The athors can consider using more advanced baselines (not a requirement).

---

> ### Author Response · Authors · 2025-05-04
>
> “_positioning of this paper…is not articulated. Prior work...should be cited and distinguished…the paper's contributions…should be emphasized_”
>
> Thank you for the opportunity to clarify this important aspect. To the best of our knowledge, this is the first paper that introduces the forward-backward Markov decision process (FB-MDP), extends the theories for this new class of MDPs, and develops an RL algorithm to find the solution. The most relevant related work has tried to better solve a forward-only MDP problem by augmenting the training dataset through an artificial backward trajectory (Edwards et al., 2018; Goyal et al., 2019; Wang et al., 2021; Lai et al., 2020). However, these works do not consider an independent controlled backward MDP (or dynamics) for their problems. In contrast, our methodology considers the actual controlled backward and forward dynamics jointly competing within the action space in both directions of time. We have revised Section 2 and more crisply stated our contribution accordingly as suggested.
>
> “_Theoretical results rely on strong assumptions, strong convexity and Lipschitz smoothness.Assumptions have been put into appendix…not clear how assumptions would hold when using NN-based functions_”
>
> Thank you for bringing this issue. First of all, Lemma 3.8 *does not* depend on those assumptions, therefore, the characteristic analysis of optimal solutions as well as the  solution of FB-MDPs are not affected. The convergence rate with order of $\mathcal{O}(1/\sqrt{K})$ is provided in Theorem B.8 for the case the expected losses are *only* Lipschitz-smooth. As a consequence, strong-convexity is not required. In the revised manuscript, the main assumptions are Lipschitz-smoothness (L-smooth) and bounded gradient variance, which align with those conventionally used in the literature about  convergence analysis (Tian et al., 2023; Xiong et al., 2022; Zhou et al., 2022; Qiu et al., 2021).
> Proposition 4.1 in the revised manuscript explicitly relates the impact of the Lipschitz-smooth assumption on neural network architectures. Specifically, the L-smoothness of a log-policy is guaranteed by (i) L-smoothness and Lipschitz-continuity of activation functions of the neural network (e.g. sigmoid, tanh), and (ii) boundedness of actions after sampling.
> We now need to check whether this proposition can hold for the considered experiments. This holds for Problem 5.2 and Problem of Appendix D. Specifically, in the Edge Caching experiment, the actions $\{w_n(t),p_n^{\rm cach}(t)\}$ can be parameterized by Sigmoid activations and can be sampled from LogNormal (for $w_n$) and Dirichlet distributions (for $p^{\rm cach}_n$).
>
> They are also limited ($0 < p_n <1$ and $w_n$ is clipped to avoid a large latency and bandwidth consumption, though it is not inherently bounded). Therefore, a neural network can be exploited to ensure L-smoothness of $log\pi_\theta(a|s)$ in *$\theta$*. In the Computation Offloading experiment, all the action elements are between 0 and 1, and as such the Sigmoid can be leveraged, and log-policy again remains L-smooth. Please refer to the added exposition in Section 5.2.2 and Proposition 4.1.
>
>  “_how restrictive this assumption: This work constrains FB-MDPs to the case where forward (backward) dynamics does not depend on backward (forward)_”
>
> Thank you for the insightful comment. The current theoretical results are obtained based on such an assumption. This assumption allows us to provide a theoretical framework and analytical results for addressing important problems such as those in the three considered use cases. Further, the Forward-Backward Stepwise mechanism (explained in the first paragraph of section 4) provides an adaptable mechanism that can be used for the case the dynamics depend on each other.
>
> “_Where is Assumption 4 mentioned in Theorem B.8? Remarks B.9 and B.10 are informal_”
>
> Thank you for the precise feedback. We intended to refer to _Assumption 3_ instead of _Assumption 4_ in the statement of that Theorem. This has now been corrected in the revised manuscript (please refer to Theorem B.7 in the revised manuscript).The statements of Remarks B.9 and B.10 have been also modified (please refer to Remarks B.8 and B.9 in the revised manuscript).
>
> “_Provide a complete version of Theorem 4.1 with all assumptions. Do you assume optimization problem at each iteration is well solved?_”
>
> We have revised Theorem 4.1 (Theorem 4.3 in the revised manuscript) to include exposition for the assumptions. These assumptions state that the value-function estimations are unbiased up to residual terms due to neural network parameterization, as well as Lipschitz-smoothness and bounded gradient variance of expected rewards that align with the literature of convergence analysis (Tian et al., 2023;Xiong et al., 2022;Zhou et al., 2022;Qiu et al., 2021). They ensure the monotonic increase of expected rewards (Theorem B.3) as well as the convergence. Apart from those, we make no other assumption.

---

### Review · Reviewer_tjxc · 2025-04-17

**Summary Of Contributions:**

The authors introduce Forward-Backward Markov Decision Process (FB‑MDPs) model, an extension of traditional MDPs that incorporates dual dynamics and rewards. In an FB‑MDP, each action is associated with two transition functions—one over the forward state space and one over the backward state space—and two corresponding reward signals. The learning objective is reformulated as finding Pareto‑optimal value functions that balance forward and backward rewards.

To tackle this multi‑objective optimization, the paper presents the Forward‑Backward Multi‑Objective Actor‑Critic (FB‑MOAC) algorithm. FB‑MOAC leverages actor‑critic architecture to simultaneously learn policies that navigate both dynamics and optimize the trade‑off between rewards.

Empirical evaluations on three benchmark tasks showcase the effectiveness of FB‑MOAC: in each case, the proposed algorithm outperforms prior methods in converging to higher Pareto‑optimal frontiers, thereby demonstrating its practical advantages for problems with coupled forward and backward dynamics.

**Audience:**

No

**Claims And Evidence:**

Yes

**Requested Changes:**

Since the authors define an FB‑MDP as a tuple $(S, Y, A, P_f, P_b, r_f, r_b)$, I recommend that each case study be presented by explicitly specifying: the forward state set $S$, the backward state set $Y$, the action set $A$, the forward transition function $P_f$ and backward transition function $P_b$ induced by an action, the forward reward $r_f$ and backward reward $r_b$, and the formulation of the resulting multi‑objective optimization. For each example, I would also recommend to include a brief discussion explaining why the problem cannot be equivalently modeled and solved as a classical MDP.

Additionally, I have a few minor points:

-Since Problems 5.1 and 5.2 are treated via discretization, please describe them as discrete‐state problems rather than as the original ODE formulations.

-Include the full name “Forward‑Backward Multi‑Objective Actor‑Critic (FB‑MOAC)” in the abstract.

-At the start of Section 4, the text refers to “mechanism according to Theorem 3.7 and Theorem 3.8,” but Theorem 3.7 is actually a remark, and Theorem 3.8 is a lemma—please correct these references.

**Strengths And Weaknesses:**

The main strength of the paper lies in its technical rigor and the soundness of its mathematical analysis and proofs. Although the authors employ established proof techniques rather than novel strategies, their derivations are clear, logically coherent, and meticulously presented.

However, the case studies are unclear and fail to demonstrate why FB‑MDPs are necessary compared to standard MDPs. In particular, in the motivating example (Section 2.1) it remains ambiguous why the process cannot be modeled as a regular MDP with stochastic dynamics, where agents move to different states and receive different rewards depending on if content transmission was completed or failed. Therefore, the paper does not convincingly illustrate the practical advantages of FB‑MDPs over classical models. For these reasons, I recommend rejecting the manuscript in its current form, although I would be willing to reconsider my evaluation if the authors substantially clarify their empirical evaluation to show how FB‑MDPs yield benefits that standard MDPs cannot capture in the proposed scenarios. I give detailed suggestions in the next section.

---

> ### Author Response · Authors · 2025-05-04
>
> “_case studies … fail to demonstrate why FB‑MDPs are necessary compared to standard MDPs…why process cannot be…a regular MDP with…different rewards depending on…transmissions completed or failed..._”
>
> Thanks for this opportunity to clarify. Regarding the motivating example, the following explains why it cannot be converted to a standard MDP. The average latency $l_n(t)=d(t)(1–e_n(t))+(\tau(t)+l_n(t+1))e_n(t)$ may suggest converting to a standard forward dynamics. In such a case, one could consider the transformations $K_n(T−t)=l_n(t)$ and $t’=T −t$. The following forward dynamics will then emerge:
> $k_n(t’)=d(T-t’)(1–e_n(T-t’))+(\tau(T-t’)+k_n(t’))e_n(T-t’)$ with $k_n(0)=l_n(T)$.
> However, this is a *non-standard* MDP, as state $k_n(t′)$ depends on action $e_n(T− t′)$ scheduled for future time steps. Specifically, the state relies on future actions that cannot be revealed by moving forward in time.
>
> More importantly the problem aims to decrease overall latency, as opposed to merely increasing the number of successful transmissions. Although a higher success rate over $[0,T]$ correlates with a lower latency, these two goals are not identical; one could maintain the same success percentage yet obtain a different overall latency $\sum_{n=1}^N p_n(t)l_n(t)$, and *the policy with less overall latency is more desirable as it can faster deliver the same contents*. For example, both the number of successful transmissions and average latency $l_n(t)$ can be determined by fixing the error probability $e_n(t)$ over [0,T]. If we then permute $e_n(t)$ in time, the total count of successful transmissions remains unchanged, but overall latency $\sum_{n=1}^N p_n(t) l_n(t)$ will differ due to $p_n(t)$. This is even more challenging for the Edge Caching as there is also a dynamics on $p_n^{ref}(t)$ Eq. (27). Similar arguments thus hold for scenarios of Edge Caching in Section 5.2 and Computation Offloading in Appendix D. Success/failure of content transmissions alone does not completely map to the overall latency.
>
> Nonetheless, it is possible to *approximate* the Edge Caching experiment using a forward-only MDP and apply the standard algorithms. As explained in Section 5.2.3 of the revised manuscript, this approximation strategy replaces the backward reward with optimizing $r_{QoS}$ and d(t), which in turn decrease the number of failed transmissions. We have already evaluated this strategy as a benchmark and the resulting comparison is plotted in Fig. 4.(a)-(b), which *clearly shows the benefit of modelling considered problems using FB-MDP* instead of being replaced by a standard MDP (refer to the comparison between FB-MOAC and F-PPO / F-MOAC); FB-MOAC gives less latency $r_{Lat}$ than forward-only RL algorithms. For the Mathematical Finance experiment expressed by a FB-SDE, the problem can be discretized merely using FB-MDP, not a standard MDP. Fig. 6 shows again the benefit of FB-MDP against standard MDP, as FB-MOAC gives less computation time ($r_{CT}$) than forward-only RL algorithms.
>
> “_authors define an FB‑MDP as a tuple (S,Y,A,Pf,Pb,rf,rb), I recommend that each case study be presented by specifying: forward state S,..._”
>
> Based on your suggestion, we have accordingly modified Sections 5.1 and 5.2 to better describe the considered experiments and introduce the forward/backward states, actions, respective problems, etc.
>
> Remark 3.6 and Section 5.2.1 explain why the problems in Section 5.2 and Appendix D cannot be modeled as a classical MDP. Problem 5.1 inherently follows a FB-SDE and can be only discretized into a FB-MDP, not a forward-only MDP. However, we have shown that these problems can either be described as a *non-standard* MDP problems with the state relying on future actions that are not available when progressing forward in time, or can be *approximated* by using a standard MDP leading to *suboptimal* solutions (see Figures 4b and 6).
>
> “_Problems 5.1 and 5.2 are treated via discretization, describe them as discrete‐state problems than as original ODE_”
>
> Thank you for the opportunity to clarify. Problem 5.1 is a continuous problem as it follows an FB-SDE. We have converted this problem to a FB-MDP formulation (i.e., a discrete‐state formulation) based on the mechanism described in Appendix E.1. We have revised the manuscript to explain the discretized model of this problem according to the reviewer’s suggestion. Instead, Problem 5.2 is inherently characterized by a discrete-state formulation. The reason is that the transmissions of the network are done in a time-slotted fashion indexed by a discrete quantity $t$. Therefore, the environment of this problem is described based on a MDP rather than an SDE. We have improved the explanation of the related text for clarity.
>
> “_Include full name “Forward‑Backward … Actor‑Critic” in abstract…Theorem 3.7 is a remark, Theorem 3.8 is a lemma please correct these references_”
>
> Thanks for the comments. We updated the abstract and the references in the revised manuscript.

---

### Review · Reviewer_mypH · 2025-04-20

**Summary Of Contributions:**

The paper studies forward-backward multi-objective reinforcement learning for multi-task control problems and proposes an actor-critic-based algorithm. The proposed method is demonstrated to be effective through convergence analysis and empirical results. The method is also demonstrated to be effective in several real world scenarios, such as mathematical finance and wireless caching.

**Audience:**

Yes

**Broader Impact Concerns:**

I do not have concerns of the ethical implications of the work.

**Claims And Evidence:**

Yes

**Requested Changes:**

1. Can you elaborate more on "The convergence results of Theorem 4.1 are aligned with those of single-optimization algorithms for forward-MDPs"? What do we want to align with single optimization algorithms for forward MDP, is that the best results available for forward MDP?
2. Please add a formal definition of locally Pareto-optimal. When there are multiple Pareto optimal points, which one does it converge to?
3. Remark 4.3 is unclear. Why is the computational complexity primarily described by the convergence rate?
4. As the theoretical guarantee is only based on Lipschitz and smooth loss, can you comment on whether the losses of the examples used in the experiments are Lipschitz and smooth? If they are not, I suggest adding an additional experiment on Lipschitz and smooth loss to better understand the performance of the algorithm on Lipschitz/smooth vs non-Lipschitz/smooth, especially when there are limited baseline algorithms.
5. Although the trained algorithm seems to be converged, I would still suggest specifying the number of random seeds used and report the standard deviation/confidence intervals of the results.

**Strengths And Weaknesses:**

The paper proposed an algorithm for multi-objective optimization in forward-backward MDP that addresses real-world applications. The proposed approach is supported by mathematical proofs and empirical evidence.

The paper could be better structured, and some of the discussions could be extended for readability. See the requested changes for a detailed list.

---

> ### Author Response · Authors · 2025-05-04
>
> “_Can you elaborate more on "The convergence results of Theorem 4.1 are aligned with those of single-optimization algorithms for forward-MDPs"? What do we want to align with single optimization algorithms…, is that the best results available for forward MDP?_”
>
> Thank you for raising this important point. Our algorithm lies within the class of *single-timescale* RL algorithms, where the actor and critic are updated simultaneously. Accordingly, Remark 4.4 in the revised manuscript compares the convergence rate of our multi-objective algorithm with that of the single-objective actor-critic algorithm within the same class. For the latter, an optimal solution can be found for forward MDP at a sublinear rate $O(1/\sqrt{K} )$, with K the number of iterations. This is comparable with our algorithm that can find a locally Pareto-optimal solution with the same rate for FB-MDPs. To the best knowledge of the authors, that is the best result for the *single-timescale actor-critic* that guarantees a globally optimal solution in solving general F-MDP problems. We have modified this remark in the revised manuscript.
>
> “add a definition of locally Pareto-optimal. When there are multiple Pareto optimal points, which one does it converge to?_”
>
> A sequence of ${(\theta_i)}$ is said to converges to a locally-Pareto solution if $E\{\ |\sum_{j \in |S_f \cup S_b|} \nabla_\theta J_j^{fb}(\theta^i) \beta_{j,act}^i \|^2\}$ tends to zeros as $i$ tends to infinity (Zhou et al., 2022) as explained in Definition 4.2 of the revised manuscript. Multi-objective problems have multiple Pareto optimal solutions, which together form the Pareto front. The analytical results in Section 4.2. show the convergence to one of those (local) Pareto optimal solutions. Finding a Pareto front is vital for many problems, therefore, Section 4.3 introduces a mechanism to empirically show that most of the Pareto solutions can be found.
>
> “_Remark 4.3 is unclear. Why is the complexity primarily described by the convergence rate?_”
>
> This remark argues that the computational complexity for FB-MOAC to reach convergence is comparable to those of standard RL algorithms. The reason is that the overall architectural complexity of FB-MOAC is comparable to that of standard actor-critic approaches, resulting in a similar convergence rate. We have revised the corresponding discussion (in Remark 4.5 of the revised manuscript) to better explain this aspect.
>
> “_As theoretical guarantee is only based on Lipschitz and smooth loss, can you comment on whether the losses of the examples used in the experiments are Lipschitz and smooth? If they are not, I suggest adding an additional experiment on Lipschitz and smooth loss …._”
>
> Proposition 4.1 in the revised manuscript explicitly relates the impact of the Lipschitz-smooth assumption on neural network architectures. Specifically, the L-smoothness of a log-policy is guaranteed by (i) L-smoothness and Lipschitz-continuity of activation functions of the neural network (e.g. sigmoid, tanh), and (ii) boundedness of actions after sampling. Not to mention that Lipschitz-smoothness assumption aligns with those commonly used in the literature about convergence analysis (Tian et al., 2023; Xiong et al., 2022; Zhou et al., 2022; Qiu et al., 2021).
>
> We now need to check whether this proposition can hold for the considered experiments. This holds for Problem 5.2 and Problem of Appendix D. Specifically, in the Edge Caching experiment, the actions $\{w_n(t),p_n^{cach}(t)\}$ can be parameterized by Sigmoid activations and can be sampled from LogNormal (for $w_n$) and Dirichlet distributions (for $p^{cach}_n$ ).
>
> They are also limited ($0 < p_n <1$ and $w_n$ is clipped to avoid a large latency and bandwidth consumption, though it is not inherently bounded). Therefore, a neural network can be exploited to ensure L-smoothness of $log\pi_\theta(a|s)$ in *$\theta$*. In the Computation Offloading experiment, all the action elements are between 0 and 1, and as such the Sigmoid can be leveraged, and log-policy again remains L-smooth. Please refer to the added exposition in Section 5.2.2 and Proposition 4.1.
>
> “_Although the trained algorithm seems to be converged, I would still suggest specifying the number of random seeds used and report the standard deviation/confidence intervals of the results_”
>
> The results have been obtained by using at least five different realizations. We have already plotted the statistical significance (i.e., the mean of the solutions in addition to their standard deviations) for both considered experiments using the method of shaded colors. This includes the main results, and the ablation studies. Specifically, Figure 2 shows the mean and the confidence interval of main results of Problem 5.1; Figures 3, 4-a and 4-c also show these for the main performance result, the comparison against the state of the art, and the ablation study (respectively).

---

### Decision · Action_Editor_2YAs · 2025-06-05

**Recommendation:** Accept as is

**Audience:**

Yes

**Audience Explanation:**

Although some reviewers have found the problem too niche and the proposed approaches incremental, they agree that the contribution is of interest for the community.

**Claims And Evidence:**

Yes

**Claims Explanation:**

The claims presented in the paper are sufficiently accurate and convincing. The revision significantly improved clarity.